# Selective induction Heads: How Transformers Select Causal Structures in Context

**Francesco D'Angelo**,* **Francesco Croce & Nicolas Flammarion**
Theory of Machine Learning Lab, EPFL, Lausanne, Switzerland

## Abstract

Transformers have exhibited exceptional capabilities in sequence modelling tasks, leveraging self-attention and in-context learning. Critical to this success are induction heads, attention circuits that enable copying tokens based on their previous occurrences. In this work, we introduce a novel synthetic framework designed to enable the theoretical analysis of transformers' ability to dynamically handle causal structures. Existing works rely on Markov Chains to study the formation of induction heads, revealing how transformers capture causal dependencies and learn transition probabilities in-context. However, they rely on a fixed causal structure that fails to capture the complexity of natural languages, where the relationship between tokens dynamically changes with context. To this end, our framework varies the causal structure through interleaved Markov chains with different lags while keeping the transition probabilities fixed. This setting unveils the formation of *Selective Induction Heads*, a new circuit that endows transformers with the ability to select the correct causal structure in-context. We empirically demonstrate that attention-only transformers learn this mechanism to predict the next token by identifying the correct lag and copying the corresponding token from the past. We provide a detailed construction of a 3-layer transformer to implement the selective induction head, and a theoretical analysis proving that this mechanism asymptotically converges to the maximum likelihood solution. Our findings advance the theoretical understanding of how transformers select causal structures, providing new insights into their functioning and interpretability.

## 1 Introduction

As autoregressive generative models continue to scale and are increasingly deployed in real-world applications, the question of how Transformer models (Vaswani et al., 2017) function internally becomes pressing. Yet the inherent complexity of natural language hinders the ability to fully comprehend how these models make decisions and work internally. To address this challenge, many recent works have attempted to formulate synthetic frameworks that simplify the problem and enable theoretical analysis while still capturing the remarkable properties and phenomena observed in large language models, such as in-context learning (Brown, 2020; Garg et al., 2022; Bai et al., 2024; Von Oswald et al., 2023a; Sander et al., 2024). Mechanistic interpretability (Olsson et al., 2022) emerges as a line of research focused on reverse-engineering the complex computations performed inside a transformer in order to understand how a certain output is produced for a given input. This research has uncovered the formation of induction heads (Olsson et al., 2022) i.e., interpretable circuits embedded within the transformer's weights, capable of simple operations such as copying tokens. By examining such circuits and their combinations, one can understand the algorithms that transformers implement to solve a given task. For instance, (Nichani et al., 2024; Bietti et al., 2024; Edelman et al., 2024) demonstrated that induction heads enable transformers to implement in-context bigrams for next-token predictions in Markov Chains. Such mechanisms are not limited to simplified models: Nguyen (2024) showed that transformers may rely on N-gram rules even in natural language processing. Yet the process by which transformers select between such learned rules remains poorly understood. While in-context learning studies how transformers solve tasks from demonstrations in the prompt, *in-context selection* focuses on how transformers

---

*Corresponding authors: francesco.dangelo@epfl.ch

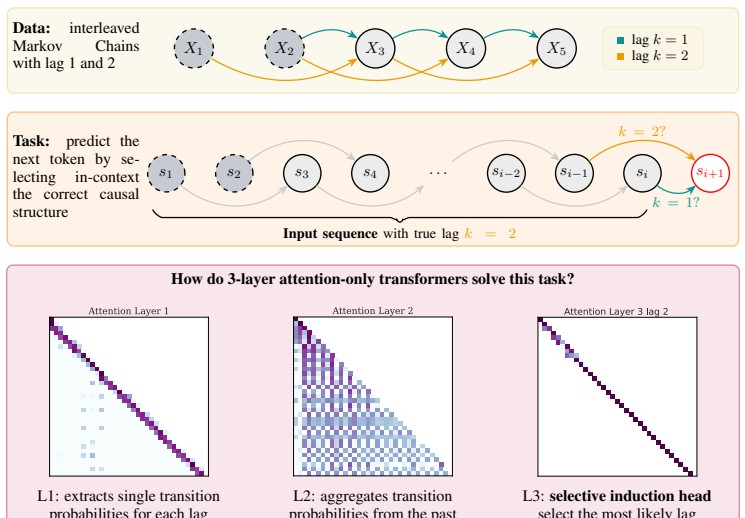

Figure 1: **Summary of the framework. Top:** we define a new task based on Interleaved Markov Chains of different lags ($k = 1$ and $k = 2$ in the example). **Middle:** given a sequence generated from a chain of unknown lag, the model has to identify the true lag, and use it to predict the distribution of the next token. **Bottom:** attention-only transformers can solve this task with 3 layers. The first computes the transition probabilities for each lag seen during training, the second aggregates these probabilities over the entire past, and finally the third layer implements the selective induction head, which selects the correct lag.

select the most suitable approach to solve a given task from those encountered during training, using instances present in the context. For example, Bai et al. (2024) examines how transformers perform in-context algorithm selection when pre-trained on mixtures of linear and logistic regression with different noise. Similarly, Yadlowsky et al. (2023) studied the ability of transformers to perform model selection between different function class families. Our work takes another step toward understanding this selection process, leveraging Markov chains with different causal structures.

**In-context causal structure selection.** Recently, Markov chains have been employed to formulate interesting sequence-to-sequence tasks that can be solved by transformers with interpretable solutions (Ildiz et al., 2024; Nichani et al., 2024; Makkuva et al., 2024; Edelman et al., 2024). In particular, Nichani et al. (2024) show that transformers trained on Markov Chain sequences learn circuits that capture the causal structure, i.e., the set of parent tokens for each token in the sequence and estimate transition probabilities in-context. The existing works relying on Markov chains, fail to model the nuanced relationships typical of natural language. The same word pair can have different causal relationships depending on the surrounding context. While an effective model should recognize these contextual dependencies, previous research has overlooked this consideration by adopting fixed causal structures. To address this limitation, we propose *a new synthetic task* designed to mimic different causal dependencies (Sec. 3). We consider *Interleaved Markov Chains*, with fixed transition probabilities between states but different underlying causal structures (Fig. 1), and theoretically study how 3-layers attention-only transformers learn to correctly predict the next token in a sequence.

**Selective induction heads.** To solve the task at hand—correctly predicting the next token in-context in a sequence generated within this setup (middle of Fig. 1)—transformers need to learn a circuit that adapts to the given context to select the correct causal structure among those seen during training. We call this circuit a *selective induction head*, as it differs fundamentally from the induction heads introduced so far in the literature, where the circuit learns either to copy a token from a certain position fixed by the unique structure of the data or by comparing its semantics. In our task, the transformer (with attention maps depicted in Fig. 1) needs to learn to aggregate all past information to determine from which past position the corresponding token should be copied in order to predict the next token.

**A transformer construction for in-context selection.** To understand and formalize the selective heads, we provide an *interpretable* construction of the self-attention layer weights in a 3-layer attention-only disentangled transformer (Friedman et al., 2023) that implements this mechanism

(Sec. 4). We empirically demonstrate that the constructed transformer matches the performance of both disentangled and standard transformed trained from scratch (Sec. 5) and that 2-layer attention-only transformers cannot solve the task. Moreover, we observe that the attention maps of the trained and constructed transformers present the same patterns, further supporting the validity of our algorithm. Finally, we theoretically analyze the predictor implemented by this construction (Sec. 4.3) showing that, in certain cases, it asymptotically converges to the maximum likelihood solution. Our findings provide insights into the mechanisms by which transformers perform model selection.

We defer to the appendix a discussion of related works (App. A), additional theoretical analyses, omitted proofs (App. B), extra experiments (Apps. D, F, C), and generalizations of our transformer construction (Apps. H, I, J).

## 2 (DISENTANGLED) TRANSFORMER MODELS

In the following we introduce the necessary background and notation about the models we use later.

**Transformers.** The architecture of decoder-only transformers is built on two fundamental components, the attention mechanism and the multi-layer perceptron (MLP). Given a finite alphabet $\mathcal{S}$, transformers map an input sequence $s = s_{1:T} = (s_1, \ldots, s_T) \in \mathcal{S}^T$ to a sequence of vectors $z = (z_1, \ldots, z_T)$ where $z_i \in \mathbb{R}^d$. Each element of the input sequence $s_i$ is first encoded using its corresponding one-hot vector, $e_{s_i} \in {0, 1}^{|\mathcal{S}|}$. These one-hot representations are then mapped to $d$-dimensional vectors via an embedding matrix $E \in \mathbb{R}^{d \times |\mathcal{S}|}$. To incorporate positional information, a positional embedding matrix $F \in \mathbb{R}^{d \times T}$ is added. With a slight abuse of notation, let $e_i$ denote the $i$-th element of the canonical basis of $\mathbb{R}^T$ such that each input element $s_i$ is mapped to a vector $x_i \in \mathbb{R}$ via $x_i = E e_{s_i} + F e_i$. The information of the different tokens is then mixed by the causal self-attention heads: denoting the key, query and value matrices $K, Q \in \mathbb{R}^{d \times d_{QK}}$, $V \in \mathbb{R}^{d \times d}$, and given an input $h \in \mathbb{R}^{d \times T}$, one gets

$$\mathrm{Attn}(h; Q, K) := \mathcal{A}(h; Q, K)h^\top, \quad \text{with} \quad \mathcal{A}(h; Q, K) := \mathrm{Softmax}\left(\mathcal{M}(h^\top Q K^\top h); \alpha\right),$$

where $\mathrm{Softmax}(v; \alpha)_i := \frac{\exp(v_i/\alpha)}{\sum_j \exp(v_j/\alpha)}$ is applied row-wise and $\alpha > 0$ is a temperature parameter. In the following, we call $A = QK^\top \in \mathbb{R}^{d \times d}$ the *attention matrix*, $\mathcal{A} \in \mathbb{R}^{T \times T}$ the *attention*, and $\mathrm{Attn} : \mathbb{R}^{d \times T} \to \mathbb{R}^{d \times T}$ the *attention layer*. The causality of the self-attention is enforced by a mask $\mathcal{M}$, to prevent the model from attending to future tokens, i.e. $\mathcal{M}(A)_{ij} = A_{ij}$ if $i \geq j$, $-\infty$ otherwise. For a model with $L$ layers and $\{H_l\}_{l \in [L]}$ attentions heads per layer, we denote by $Q^{(l,h)}, K^{(l,h)}, V^{(l,h)}$ the attention parameters for the $i$-th head in the $l$-th layer, $W_1^{(l)}, W_2^{(l)} \in \mathbb{R}^{d \times d_{\mathrm{FF}}}$ the parameters of the MLP at layer $l$, and $W_O \in \mathbb{R}^{|\mathcal{S}| \times d}$ the parameters of the output linear layer. Then, with $h^{(0)} = (x_1, \ldots, x_T) \in \mathbb{R}^{d \times T}$ as computed above, the decoder transformer $\mathcal{T}(s_{1:T})$ can be written for $l = 1, \ldots, L$, as

$$\hat{h}^{(l)} = h^{(l-1)} + \sum_{h=1}^{H_l} \mathrm{Attn}(h^{(l-1)}; Q^{(l,h)}, K^{(l,h)})V^{(l,h)}, \quad h^{(l)} = \hat{h}^{(l)} + W_2^{(l)}\sigma\left(W_1^{(\ell)\top}\hat{h}^{(l)}\right)$$

where the output is given by $W_O h^{(L)} \in \mathbb{R}^{|\mathcal{S}| \times T}$.

**Disentangled Transformers.** To improve the interpretability of the operations implemented by the models, Friedman et al. (2023) propose a transformer architecture in which each layer's output is concatenated, rather than added, to its input. This construction makes the residual stream explicitly disentangled, but increases the embedding dimension (constant for standard transformers) with depth. Additionally, in such *disentangled transformers* the MLP layers are removed, the attention heads are parameterized by a single matrix $\tilde{A} := QK^\top \in \mathbb{R}^{d_\ell \times d_\ell}$, and the value matrices are absorbed into the output layer $\widetilde{W}_O$. Both the token and positional embedding are one-hot encoding, i.e. $E$ and $F$ are identity matrices, and we encode the input $s_i$ as $[e_{s_i}, e_i]$ via concatenation rather than addition. Altogether, the disentangled transformer $\widetilde{\mathcal{T}}(s_{1:T})$ is formalized for $l = 1, \ldots, L$ as

$$\hat{h}^{(l,h)} = \mathrm{Attn}(h^{(l-1)}; \tilde{A}^{(l,h)}) \quad \text{for} \quad h = 1, \ldots, H_l, \quad \text{and} \quad h^{(l)} = [h^{(l-1)}, \hat{h}^{(l,1)}, \ldots, \hat{h}^{(l,H_l)}],$$

where the outputs is $\widetilde{W}_O h^{(L)}$. Due to the concatenation, the embedding dimensions grows over layers as $d_l = (1 + H_l) \cdot d_{l-1}$ with $d_0 = |\mathcal{S}| + T$. Importantly, Nichani et al. (2024) demonstrate that disentangled transformers are equivalent to standard transformers using only attention layers.

## 3 MARKOV CHAINS AND CAUSAL STRUCTURE SELECTION

To address the limitations of existing synthetic settings based on Markov chains and better capture the complexity of natural language, we propose a novel framework. In this framework, the model must learn to select the correct causal structure in-context in order to solve the task and generate the input sequence. In the following, we describe this task in detail and outline its solution.

**Interleaved Markov Chains.** The framework consists of sequences of length $T$ on a finite alphabet of tokens $\mathcal{S}$, generated by $K$ distinct sources. Let $\mathcal{U} = \{U_1, \ldots, U_K\}$ be the set of sources and $\mathcal{K} = \{k_1, \ldots, k_K\}$ a set of positive integers; each source $U_j$ consists of $k_j$ interleaved and identical irreducible aperiodic Markov chains (Batu et al., 2004; Minot & Lu, 2014). All the sources are defined by the same transition matrix $P^\star \in \mathcal{P}^{|\mathcal{S}| \times |\mathcal{S}|}$, where $\mathcal{P}$ is the set of row-stochastic matrices. This model is equivalent to a time-homogeneous Markov chain $(X_t^{(j)})_{t \geq 0}$ of order $k_j$, whose transition probabilities depend only on a single state $k_j$ steps back:

$$\mathbb{P}(X_t = s_t \mid X_{t-1} = s_{t-1}, \ldots, X_1 = s_1) = \mathbb{P}(X_t = s_t \mid X_{t-k_j} = s_{t-k_j}) = s_{t-k_j}^\top P^\star s_t.$$

Here, we call $k_j \in \mathcal{K}$ the *lag* parameter, as defined by Berchtold & Raftery (2002), where $\mathcal{K} \subseteq [\![1, t]\!]$ is the set of possible lags. The lag, represented by the edges in Fig. 1, encodes the causal structure by explicitly representing the causal relationship between the variables in the Markov chain.

**Data.** Given $P^\star$ and $\mathcal{K}$, a lag is uniformly sampled from $\mathcal{K}$ for each sequence. Denoting the maximum lag by $\hat{k} = \max(\mathcal{K})$, the first $\hat{k}$ elements of each sequence are sampled from the stationary distribution $\pi$ of $P^\star$, ensuring a constant number of independent variables for all sources. The likelihood of a sequence of lag $k$ is $\mathbb{P}(X_1, \ldots, X_T \mid k) = \prod_{i=1}^{\hat{k}} \pi(X_i) \prod_{j=\hat{k}+1}^{T} \mathbb{P}(X_j \mid X_{j-k+1})$.

**Task.** In this setting, the task is to predict the next state $s_{T+1}$ given an input sequence $s_{1:T}$ generated from one of the sources, sampled at random. However, the identity of the source, and therefore the lag, is unknown. This task amounts to solving the following minimization problem:

$$f^\star = \inf_f \mathbb{E}_{\substack{k \sim \text{Unif}[1, \ldots, \hat{k}] \\ (X_{1:T}) \sim \mathbb{P}(X_1, \ldots, X_T \mid k)}} \mathcal{D}_{KL} \left( \mathbb{P}(X_{t+1} \mid X_{t-k+1}) \, \| \, f(X_1, \ldots, X_T) \right), \tag{1}$$

where $\mathcal{D}_{KL}$ is the Kullback–Leibler divergence. Eq. (1) admits a closed form solution which is the Bayesian model average (BMA), defined as the average of the transition probabilities for each lag, weighted by their posterior probabilities:

$$\mathbb{P}(X_{t+1} \mid X_{1:T}) = \sum_{k \in \mathcal{K}} w_k(X_{1:T}) \mathbb{P}(X_{t+1} \mid X_{t-k+1}) \text{ with } w_k(X_{1:T}) = \frac{\mathbb{P}(X_{1:T} \mid k) \mathbb{P}(k)}{\sum_{k \in \mathcal{K}} \mathbb{P}(X_{1:T} \mid k) \mathbb{P}(k)}.$$

Asymptotically, the posterior distribution concentrates around the maximum likelihood (ML) estimate (Rousseau & Mengersen, 2011). Let $k^*$ be the lag that maximizes the likelihood for a sequence $(s_1, \ldots, s_T)$, i.e., $k^* = \arg\max_{k \in \mathcal{K}} \mathbb{P}(X_1 = s_1, \ldots, X_T = s_T \mid k)$. As $T \to \infty$, the posterior probability $w_k$ converges to 1 for $k^*$ and to 0 for the other lags, i.e., $w_k \to \mathbb{1}[k = k^*]$ where $\mathbb{1}$ is the indicator function. Then, BMA reduces to selecting the lag with the highest likelihood:

$$\mathbb{Q}(X_{t+1} \mid X_1, \ldots, X_T) = \sum_{k \in \mathcal{K}} \mathbb{1}[k = k^*] \mathbb{P}(X_{t+1} \mid X_{t-k+1}). \tag{2}$$

It is important to note that an interleaved Markov chain of lag $k$ is mathematically equivalent to a $k$-th order Markov chain with a specific transition structure. Thus, given a set of orders $\mathcal{K}$ and a sequence generated according to one such order, one could theoretically solve the task by learning in-context the corresponding $(\hat{k} + 1)$-gram transition probabilities (Nichani et al., 2024; Edelman et al., 2024). However, such an approach fails to leverage the low-dimensional structure of the problem, resulting in a suboptimal sample complexity of $\mathcal{O}(|\mathcal{S}|^{\hat{k}+1})$.

## 4 HOW CAN TRANSFORMER DO IN-CONTEXT SELECTION?

We now want to understand which algorithm transformers learn during training. We focus on disentangled transformers as defined in Sec. 2, which allow for a more interpretable analysis of the model internal computations. The following proposition, which is the main result of the paper, shows how a disentangled transformer can implement a predictor to solve the in-context selection task.

**Proposition 1.** *Let $\mathcal{K}$ be a contiguous subset of integers, i.e., $\mathcal{K} = [\![\hat{k} - K + 1, \hat{k}]\!]$ for $K = |\mathcal{K}|$ and $\hat{k} = \max(\mathcal{K})$. For any $T \geq \hat{k}$ there exists a three-layer disentangled transformer $\widetilde{\mathcal{T}}$ with $K$ heads in the second layer such that, defining $\tilde{p}_{i,k} := \frac{X_{i-k}^\top P^\star X_i}{\sum_{l \in \mathcal{K}} X_{i-l}^\top P^\star X_i}$ for $i > 1$:*

$$\widetilde{\mathcal{T}}(X_{1:T})_T = \sum_{k \in \mathcal{K}} \tilde{w}_k(X_{1:T}) \mathbb{P}(X_{t+1} \mid X_{t-k+1}) \text{ with } \tilde{w}_k(X_{1:T}) = \frac{\exp\left(\frac{\beta}{(T-\hat{k})} \sum_{i=\hat{k}+1}^{T} \tilde{p}_{i,k}\right)}{\sum_{m \in \mathcal{K}} \exp\left(\frac{\beta}{(T-\hat{k})} \sum_{i=\hat{k}+1}^{T} \tilde{p}_{i,m}\right)}. \tag{3}$$

The predictor implemented by transformers in Eq. (3) resembles BMA but differs in how it aggregates past information. Instead of using the posterior of each model as in BMA, our method employs weights proportional to the exponential of the average of normalized transition probabilities $\tilde{p}_{i,k}$. We analyze this predictor in Sec. 4.3 and discuss its convergence to ML. Proposition 1 illustrates how, for large $\beta$, transformers can implement *selective induction heads*, a mechanism that adapts to the input sequence by copying the token correspondent to the lag that maximizes the average normalized transition probabilities. The proof of Proposition 1, involves an explicit construction for the weights of the disentangled transformer implementing the solution in Eq. (3) (an alternative third layer in App. H). Notably, this construction produces attention maps similar to those in standard transformers (Fig. 1), suggesting our algorithm aligns with trained transformer implementations.

### 4.1 PROOF OF PROPOSITION 1: CONSTRUCTION FOR CONTIGUOUS LAGS

To aid intuition, we use a running example with visual illustrations for $T = 10$, $\mathcal{K} = \{1, 2, 3\}$. We recall that each input element $s_i$ is encoded as $h_i^{(0)} = [e_{s_i}, e_i] \in \{0, 1\}^{|\mathcal{S}| + T}$.

**First layer: extraction of transition probabilities.** The first attention matrix, $\tilde{A}^{(1)}$, consists of two blocks: the first block operates on the semantic component of the input tokens, learning the transpose of the logarithm of the transition matrix. The second block $A^{(1)}$ learns the causal relationships induced by each possible lag $s_{i-k} \to s_i$ for $k \in \mathcal{K}$:

$$\tilde{A}^{(1)} = \begin{pmatrix} \log P^\top & 0 \\ 0 & A^{(1)} \end{pmatrix}$$

$$A_{ij}^{(1)} = \begin{cases} +\lambda & \text{if } i - j \in \mathcal{K} \\ -\lambda & \text{if } i - j \notin \mathcal{K}. \end{cases}$$

We can compute the first layer's attention as: $[e_{s_i}, e_i]^\top \tilde{A}^{(1)} [e_{s_j}, e_j] = (\log P)_{s_j, s_i} + \lambda \text{sign}(A_{ij}^{(1)})$, and applying the softmax ($\alpha = 1$): $\mathcal{A}^{(1)}(h_{1:T}^{(0)}; \tilde{A}^{(1)})_{ij} = \frac{e^{(\log P)_{s_j, s_i} + \lambda \text{sign}(A_{ij}^{(1)})}}{\sum_{r \in \mathcal{K}} e^{(\log P)_{s_r, s_i} + \lambda} + \sum_{r \notin \mathcal{K}} e^{(\log P)_{s_r, s_i} - \lambda}}$.

For $\lambda \to \infty$ (in practice, for $\lambda$ large enough) and denoting $\tilde{p}_{i,k} := \frac{P_{s_{i-k}, s_i}}{\sum_{r \in \mathcal{K}, r < i} P_{s_{i-r}, s_i}}$ for $i > 1$,

$$\lim_{\lambda \to \infty} \mathcal{A}^{(1)}(h_{1:T}^{(0)}; \tilde{A}^{(1)})_{ij} = \begin{cases} \tilde{p}_{i, i-j} & \text{if } i - j \in \mathcal{K} \\ 1 & \text{if } i = j = 1 \\ 0 & \text{elsewhere}. \end{cases}$$

Therefore, the output at index $i$ after the first layer corresponds to a weighted average of the past tokens $h_{i-k}^{(0)}$ for $k \in \mathcal{K}$ where the weights are given by the normalized probabilities $\tilde{p}_{i,k}$: $\hat{h}_i^{(1)} = \text{Attn}(h_{1:T}^{(0)}; \tilde{A}^{(1)})_i = \sum_{k \in \mathcal{K}, k < i} \tilde{p}_{i,k} h_{i-k}^{(0)}$ for $i > 1$ and $h_1^{(0)}$ for $i = 1$. With the input vectors $h_i^{(0)}$ being the concatenation of the one-hot encoding of the state and position $[e_{s_i}, e_i]$, the first $|\mathcal{S}|$ entries of $\hat{h}_i^{(1)}$ correspond to $\tilde{s}_i = \sum_{k \in \mathcal{K}, k < i} \tilde{p}_{i,k} e_{s_{i-k}}$ for $i > 1$ and $\tilde{s}_1 = e_{s_1}$. The remaining entries, due to the non-overlapping positional encodings, directly copy the normalized transition probabilities for the transition $s_{i-k} \to s_i$ into the $|\mathcal{S}| + (i - k)$-th element of $\hat{h}_i^{(1)}$. To build intuition, we refer to the example in Eq. (12) where the colors highlight transition probabilities of the same lag:

$$\tag{4}$$

The operation of the first layer is now explicit: for each token $h_i^{(0)}$, it extracts the normalized transition probabilities $\tilde{p}_{i,k}$ for each possible lag and stores them in the element $\hat{h}_{i,S+T-k}^{(1)}$. The resulting vector is subsequently concatenated to the residual stream to be fed to the second layer.

**Second layer: aggregation of transition probabilities.** To predict the next token, the model needs to determine which lag generated the sequence based on the past transitions. This selection requires aggregating the normalized transition probabilities from the past, and storing them in the embedding of the current token. However, since consecutive tokens store transition probabilities in overlapping positions, the attention needs to learn a convex combination of tokens that avoid mixing information from different transitions while maximizing the number of $\tilde{p}$ stored (to not discard useful information). For instance, when aggregating the past for the token at $i = 10$ in Eq. (12), summing $\hat{h}_9^{(1)}$ and $\hat{h}_{10}^{(1)}$ would mix $\tilde{p}_{10,2}$ and $\tilde{p}_{9,1}$ together. This mixing can be avoided, for example, by only selecting tokens every 3 steps $(\hat{h}_4^{(1)}, \hat{h}_7^{(1)}, \hat{h}_{10}^{(1)})$ copying transitions without blending information. More generally, the attention $\mathcal{A}^{(2,1)}$ should attend to every $K$-th token from the current one, which is equivalent to having non-zero entries along the diagonals at positions $nK$ for $n \in \mathbb{N}$ and $nK < T$. This structure can be enforced by constructing the attention matrix $\tilde{A}^{(2,1)}$ with a single non-zero block operating on the tokens' positional encoding, as follows:

$$\tilde{A}^{(2,1)} = \begin{pmatrix} 0 & 0 & \vdots & 0 \\ 0 & A^{(2,1)} & \vdots & 0 \\ \hdashline 0 & \vdots & 0 & 0 \end{pmatrix} \quad A^{(2,1)} = \begin{pmatrix} \text{...} \end{pmatrix} \quad \mathcal{A}^{(2,1)} = \begin{pmatrix} \text{...} \end{pmatrix}$$

where the first $\hat{k}$ rows and columns are empty because the first $\hat{k}$ elements of the sequence are sampled independently from the stationary distribution and therefore not informative. This construction resolves the issue of overlapping transitions, but copying only a subset of tokens implies losing information from the excluded $\hat{h}_i$. Introducing additional attention heads $\tilde{A}^{(2,2)}, \ldots, \tilde{A}^{(2,H_2)}$ with the same form as $\tilde{A}^{(2,1)}$ above overcomes this limitation. The resulting attentions $\mathcal{A}^{(2,2)}, \ldots \mathcal{A}^{(2,H_2)}$ still follow a diagonal structure as $\mathcal{A}^{(2,1)}$ to avoid overlapping transitions, but they are shifted to copy different tokens. For the given example, we can design $A^{(2,2)}$ as in Eq. (5) to attend to $\hat{h}_9^{(1)}$ and $\hat{h}_6^{(1)}$, and similarly, construct $A^{(2,3)}$ for $\hat{h}_8^{(1)}$ and $\hat{h}_5^{(1)}$:

$$A^{(2,2)} = \begin{pmatrix} \text{...} \end{pmatrix} \qquad A^{(2,3)} = \begin{pmatrix} \text{...} \end{pmatrix} \tag{5}$$

Each head has a diagonal structure with non-zero entries along the diagonals at position $nK + h - 1$ for $n \geq 0$ and $h \in \{1, \ldots, H_2\}$, the attention matrices can be formalized as:

$$A_{ij}^{(2,h)} = \lambda \begin{cases} +1, & \text{if } i \geq j > \hat{k} \text{ and } (i - j) \mod K = h - 1 \\ -1, & \text{otherwise,} \end{cases}$$

where the condition $i \geq j > \hat{k}$ ensures that all entries in the first $\hat{k}$ rows and columns are set to $-\lambda$ and imposes a lower triangular structure due to causal masking. The modulo operation instead assigns each diagonal multiple of $K$ to $+\lambda$ (allowing attention) and the remaining diagonals to $-\lambda$ (masking attention), while $h$ determines the shift of the the first positive diagonal to ensure the heads do not overlap. The output of each head in the second layer is given by $[[e_{s_i}, e_i], \hat{h}_i^{(1)}]^\top \tilde{A}^{(2,h)} [[e_{s_j}, e_j], \hat{h}_j^{(1)}] = A_{i,j}^{(2,h)} = \lambda \text{sign}(A_{i,j}^{(2,h)})$. Applying softmax and in the limit as $\lambda \to \infty$, the rows of the attention become uniform for positive entries and zero otherwise:

$$\mathcal{A}_{ij}^{(2,h)} = \text{Softmax}(A^{(2,h)}; 1)_{ij} = \mathbb{1}\left[A_{ij}^{(2,h)} = \lambda\right] \left(\sum_{m=1}^{i} \mathbb{1}\left[A_{im}^{(2,h)} = \lambda\right]\right)^{-1}.$$

The output $\hat{h}_i^{(2,h)}$ of each head is then concatenated into the residual stream. Fig. 2 shows the output for the 10th token for each attention head $\hat{h}_{10}^{(2,1)}, \hat{h}_{10}^{(2,2)}$ and $\hat{h}_{10}^{(2,3)}$ to visualize the mechanism

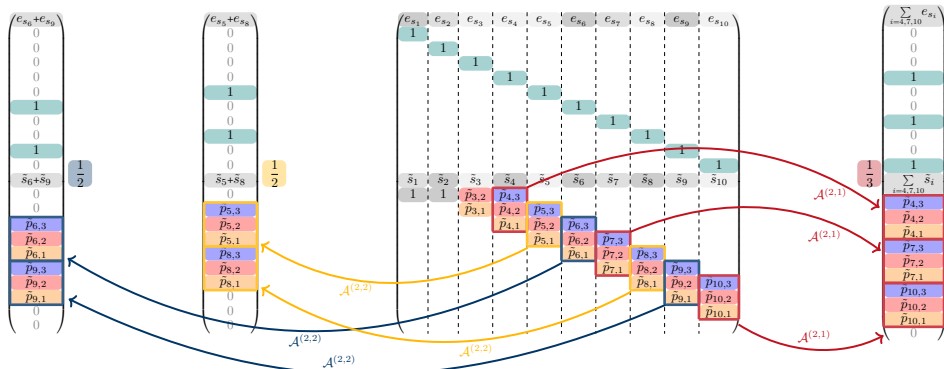

Figure 2: **Visualization of the mechanism of the second attention layer in our construction.** The matrix represents the input of the second layer $h^{(1)}$ whereas the single vectors the output for the 10$^{\text{th}}$ token $\hat{h}_{10}^{(2,1)}, \hat{h}_{10}^{(2,2)}, \hat{h}_{10}^{(2,3)}$. Each of the three attention heads (arrows of different colors) copies non-overlapping transition probabilities at distance 3 from each other from the past. By doing this, the output of the second layer for the current token (10) contains all $\tilde{p}$ for each lag without loss of information.

of the second layer. The arrows of different colors represent how each head aggregates transition probabilities by attending to non-overlapping past tokens and averaging them with uniform weights. When concatenating the output of the different heads $h_i^{(2)} = [h_i^{(0)}, \hat{h}_i^{(1)}, \hat{h}_i^{(2,1)}, \ldots, \hat{h}_i^{(2,H_2)}]$, we can see how the 10th token stores the transition probabilities of its entire past for each lag.

**Third layer: average of transition probabilities and lag selection.** The third layer sums the normalized transition probabilities from the second layer embeddings and uses the result to infer the correct lag. This mechanism is implemented through the combination of multiple blocks within the third attention matrix, $\tilde{A}^{(3)}$, which is structured as follows:

$$\tilde{A}^{(3)} = \begin{pmatrix} \begin{smallmatrix} 0 & 0 \\ 0 & A^{(3)} \end{smallmatrix} & 0 & 0 & 0 & \cdots & 0 & 0 \\ 0 & 0 & 0 & 0 & \cdots & 0 & 0 \\ 0 & 0 & 0 & 0 & \cdots & 0 & 0 \\ 0 & 0 & \begin{smallmatrix} 0 & 0 \\ 0 & B^{(3)} \end{smallmatrix} & 0 & \cdots & 0 & 0 \\ \vdots & \vdots & \vdots & \vdots & \ddots & \vdots & \vdots \\ 0 & 0 & 0 & 0 & \cdots & 0 & 0 \\ 0 & 0 & 0 & 0 & \cdots & \begin{smallmatrix} 0 & 0 \\ 0 & B^{(3)} \end{smallmatrix} & 0 \end{pmatrix} \quad A^{(3)} = \begin{pmatrix} +\lambda & -\lambda & -\lambda & -\lambda & -\lambda & -\lambda & -\lambda & -\lambda & -\lambda & -\lambda \\ +\lambda & +\lambda & -\lambda & -\lambda & -\lambda & -\lambda & -\lambda & -\lambda & -\lambda & -\lambda \\ +\lambda & +\lambda & +\lambda & -\lambda & -\lambda & -\lambda & -\lambda & -\lambda & -\lambda & -\lambda \\ -\lambda & +\lambda & +\lambda & +\lambda & -\lambda & -\lambda & -\lambda & -\lambda & -\lambda & -\lambda \\ -\lambda & -\lambda & +\lambda & +\lambda & +\lambda & -\lambda & -\lambda & -\lambda & -\lambda & -\lambda \\ -\lambda & -\lambda & -\lambda & +\lambda & +\lambda & +\lambda & -\lambda & -\lambda & -\lambda & -\lambda \\ -\lambda & -\lambda & -\lambda & -\lambda & +\lambda & +\lambda & +\lambda & -\lambda & -\lambda & -\lambda \\ -\lambda & -\lambda & -\lambda & -\lambda & -\lambda & +\lambda & +\lambda & +\lambda & -\lambda & -\lambda \\ -\lambda & -\lambda & -\lambda & -\lambda & -\lambda & -\lambda & +\lambda & +\lambda & +\lambda & -\lambda \\ -\lambda & -\lambda & -\lambda & -\lambda & -\lambda & -\lambda & -\lambda & +\lambda & +\lambda & +\lambda \end{pmatrix} \quad B^{(3)} = \beta \begin{pmatrix} 0 & 1 & 0 & 0 & 1 & 0 & 0 & 1 & 0 & 0 \\ 0 & 0 & 1 & 0 & 0 & 1 & 0 & 0 & 1 & 0 \\ 1 & 0 & 0 & 1 & 0 & 0 & 1 & 0 & 0 & 1 \\ 0 & 1 & 0 & 0 & 1 & 0 & 0 & 1 & 0 & 0 \\ 0 & 0 & 1 & 0 & 0 & 1 & 0 & 0 & 1 & 0 \\ 1 & 0 & 0 & 1 & 0 & 0 & 1 & 0 & 0 & 1 \\ 0 & 1 & 0 & 0 & 1 & 0 & 0 & 1 & 0 & 0 \\ 0 & 0 & 1 & 0 & 0 & 1 & 0 & 0 & 1 & 0 \\ 1 & 0 & 0 & 1 & 0 & 0 & 1 & 0 & 0 & 1 \\ 0 & 1 & 0 & 0 & 1 & 0 & 0 & 1 & 0 & 0 \end{pmatrix}$$

$$A_{ij}^{(3)} = \lambda \begin{cases} +1 & \text{if } i - j + 1 \in \mathcal{K} \\ -1 & \text{if } i - j + 1 \notin \mathcal{K} \end{cases}, \qquad B_{ij}^{(3)} = \beta \begin{cases} +1, & \text{if } (i - j) \mod K = K - 1 \\ 0, & \text{otherwise} \end{cases}.$$

The matrix $A^{(3)}$ acts on the positional embedding of the input, similarly to the matrix $A^{(1)}$ in the first layer. The difference is that the position of the diagonals is now shifted by one. This shift ensures that the only non-zero entries after softmax are the ones on the diagonals corresponding to $k - 1$ for $k \in \mathcal{K}$. The matrix $B^{(3)}$ is instead responsible for the sum of the normalized transitions. Each block operates on the output of a corresponding head in the second layer. To understand how, consider the following tokens in output of the first head in the second layer,

$$\hat{h}_{10}^{(2,1)} = \tfrac{1}{3} \cdot \left( \sum_{i=4,7,10} e_{s_i} \quad 0 \ 0 \ 0 \ 1 \ 0 \ 0 \ 1 \ 0 \ 0 \ 1 \quad \sum_{i=4,7,10} \tilde{s}_i \quad \underbrace{\tilde{p}_{4,3} \ \tilde{p}_{4,2} \ \tilde{p}_{4,1} \ \tilde{p}_{7,3} \ \tilde{p}_{7,2} \ \tilde{p}_{7,1} \ \tilde{p}_{10,3} \ \tilde{p}_{10,2} \ \tilde{p}_{10,1} \ 0} \right)$$

$$\hat{h}_{9}^{(2,1)} = \tfrac{1}{2} \cdot \left( e_{s_6} + e_{s_9} \quad 0 \ 0 \ 0 \ 0 \ 0 \ 1 \ 0 \ 0 \ 1 \ 0 \quad \tilde{s}_6 + \tilde{s}_9 \quad 0 \quad 0 \quad \tilde{p}_{6,3} \ \tilde{p}_{6,2} \ \tilde{p}_{6,1} \ \tilde{p}_{9,3} \ \tilde{p}_{9,2} \ \tilde{p}_{9,1} \ 0 \quad 0 \right)$$

$$\hat{h}_{8}^{(2,1)} = \tfrac{1}{2} \cdot \left( e_{s_5} + e_{s_8} \quad 0 \ 0 \ 0 \ 0 \ 1 \ 0 \ 0 \ 1 \ 0 \ 0 \quad \tilde{s}_5 + \tilde{s}_8 \quad 0 \quad \underbrace{\tilde{p}_{5,3} \ \tilde{p}_{5,2} \ \tilde{p}_{5,1} \ \tilde{p}_{8,3} \ \tilde{p}_{8,2} \ \tilde{p}_{8,1} \ 0 \quad 0 \quad 0}_{\hat{p}_8^{(2,1)}} \right)$$

$$\underbrace{\phantom{0 \ 0 \ 0 \ 1 \ 0 \ 0 \ 1 \ 0 \ 0}}_{\hat{m}_8^{(2,1)}}$$

where we define $\hat{p}_i^{(2,h)} \in \mathbb{R}^T$ as the block of $\hat{h}_i^{(2,h)}$ which contains the normalized transition probabilities and $\hat{m}_i^{(2,h)} \in \mathbb{R}^T$ contains a copy of the second attention. With the structure of $\tilde{A}^{(3)}$, we can see how $B^{(3)}$ acts on these two blocks: $h_i^{(2)\top} \tilde{A}^{(3)} h_j^{(2)} = \sum_{h=1}^K p_i^{(2,h)\top} B^{(3)} m_j^{(2,h)} + e_i A^{(3)} e_j$. This operation sums the transition probabilities such that the entry corresponding to the lag-$k$ transition for the next token contains the sum of the transitions with the same lag: $h_i^{(2)\top} \tilde{A}^{(3)} h_{i-k+1}^{(2)} \propto$

$\sum_{j \le i} \tilde{p}_{j,k}$. To illustrate this process, consider the following:

$$\hat{p}_{10}^{(2,1)\top} B^{(3,1)} \hat{m}_8^{(2,1)} = \frac{\beta}{3} \begin{pmatrix} \tilde{p}_{4,3} \\ \tilde{p}_{4,2} \\ \tilde{p}_{4,1} \\ \tilde{p}_{7,3} \\ \tilde{p}_{7,2} \\ \tilde{p}_{7,1} \\ \tilde{p}_{10,3} \\ \tilde{p}_{10,2} \\ \tilde{p}_{10,1} \\ 0 \end{pmatrix}^{\top} \begin{pmatrix} 0 & 1 & 0 & 0 & 1 & 0 & 0 & 1 & 0 & 0 \\ 0 & 0 & 1 & 0 & 0 & 1 & 0 & 0 & 1 & 0 \\ 1 & 0 & 0 & 1 & 0 & 0 & 1 & 0 & 0 & 1 \\ 0 & 1 & 0 & 0 & 1 & 0 & 0 & 1 & 0 & 0 \\ 0 & 0 & 1 & 0 & 0 & 1 & 0 & 0 & 1 & 0 \\ 1 & 0 & 0 & 1 & 0 & 0 & 1 & 0 & 0 & 1 \\ 0 & 1 & 0 & 0 & 1 & 0 & 0 & 1 & 0 & 0 \\ 0 & 0 & 1 & 0 & 0 & 1 & 0 & 0 & 1 & 0 \\ 1 & 0 & 0 & 1 & 0 & 0 & 1 & 0 & 0 & 1 \\ 0 & 1 & 0 & 0 & 1 & 0 & 0 & 1 & 0 & 0 \end{pmatrix} \begin{pmatrix} 0 \\ 0 \\ 0 \\ 0 \\ 1/2 \\ 0 \\ 0 \\ 1/2 \\ 0 \\ 0 \end{pmatrix} = \frac{\beta}{3} \begin{pmatrix} \tilde{p}_{4,3} \\ \tilde{p}_{4,2} \\ \tilde{p}_{4,1} \\ \tilde{p}_{7,3} \\ \tilde{p}_{7,2} \\ \tilde{p}_{7,1} \\ \tilde{p}_{10,3} \\ \tilde{p}_{10,2} \\ \tilde{p}_{10,1} \\ 0 \end{pmatrix}^{\top} \begin{pmatrix} 1 \\ 0 \\ 0 \\ 1 \\ 0 \\ 0 \\ 1 \\ 0 \\ 0 \\ 0 \end{pmatrix} = \frac{\beta}{3}(\tilde{p}_{4,3} + \tilde{p}_{7,3} + \tilde{p}_{10,3}).$$

The additional blocks containing $B^{(3)}$ act on the outputs of the other heads, performing the same operation by summing the transitions of the same lag stored in the respective outputs. Considering all heads and only the non-zero entries after softmax, occurring at $j = i - k + 1$ due to $A^{(3)}$, we get

$$h_i^{(2)\top} \tilde{A}^{(3)} h_{i-k+1}^{(2)} = \sum_{h=1}^{K} p_i^{(2,h)\top} B^{(3)} m_{i-k+1}^{(2,h)} + \lambda = \sum_{h=1}^{K} \frac{\beta}{T_{h,i}+1} \sum_{n=0}^{T_{h,i}} \tilde{p}_{\hat{k}+h+nK,k} + \lambda,$$

where $T_{h,i} = \lfloor \frac{i-\hat{k}-h}{K} \rfloor$. Taking the softmax ($\alpha = 1$) and the limit $\lambda \to \infty$, then considering the last token $T$ (for large $T$) where $T_{h,T} + 1 \approx \frac{T-\hat{k}}{K}$, and absorbing $K$ into the temperature $\beta$, we recover the weights in Eq. (3):

$$\mathcal{A}^{(3)}(h_{1:T}^{(2)}; \tilde{A}^{(3)})_{T(T-k+1)} = \exp\left(\frac{\beta}{(T-\hat{k})} \sum_{i=\hat{k}+1}^{T} \tilde{p}_{i,k}\right) \Big/ \sum_{r \in \mathcal{K}} \exp\left(\frac{\beta}{(T-\hat{k})} \sum_{i=\hat{k}+1}^{T} \tilde{p}_{i,r}\right), \quad \text{for } k \in \mathcal{K}. \quad (6)$$

## 4.2 SELECTIVE INDUCTION HEAD AND NEXT TOKEN PREDICTION

**Selective induction head.** Eq. (16) shows how the last attention layer computes a weighted average of the tokens at a distance $k$ from the next one, with weights proportional to the average of the normalized transition probabilities of lag $k$. In practice, trained models often learn large values of $\beta$. Thus, we consider the limit $\beta \to \infty$, where the softmax converges to the hardmax:

$$\mathcal{A}_{i,j}^{(3)} = \mathbb{1}\left[i - j + 1 = k^\star\right] \quad \text{with} \quad k^\star = \operatorname{argmax}_k \sum_{j<i} \tilde{p}_{j,k}.$$

Here, the transformer selects the causal structure (i.e., the lag) corresponding to the largest $\sum_{j<i} \tilde{p}_{j,k}$. Given the current token $i$, the third layer then copies the token from the position $i-k^\star+1$, i.e., $\hat{h}_i^{(3)} = \sum_{j=\hat{k}}^{i} \mathbb{1}\left[i - j + 1 = k^\star\right] h_j^{(2)} = h_{i-k^\star+1}^{(2)}$. After concatenation to the residual stream, the tokens are of the following form:

$$h_i^{(3)} = \left[e_{s_i}, e_i, \hat{h}_i^{(1)}, \hat{h}_i^{(2,1)}, \dots, \hat{h}_i^{(2,H_2)}, e_{s_{i-k^\star+1}}, e_{i-k^\star+1}, \hat{h}_{i-k^\star+1}^{(1)}, \hat{h}_{i-k^\star+1}^{(2,1)}, \dots, \hat{h}_{i-k^\star+1}^{(2,H_2)}\right].$$

**Output layer: next token prediction.** Finally, the output layer $\widetilde{W}_O \in \mathbb{R}^{S \times \sum_l d_l}$ contains all zero blocks, except for the one acting on the semantics of the token copied by the third attention. This block learns the transition matrix $P^\star$ to predict the transition probabilities to the next token via

$$\widetilde{W}_O = \begin{pmatrix} 0_{S \times S} & 0_{S \times T} & 0_{S \times d_0} & 0_{S \times 2d_0} & \dots & 0_{S \times 2d_0} & P^{\star\top} & 0_{S \times T} & 0_{S \times d_0} & 0_{S \times 2d_0} & \dots & 0_{S \times 2d_0} \end{pmatrix}$$

i.e. $\widetilde{W}_O h^{(3)} = P^{\star\top} e_{s_{i-k^\star+1}} = P_{s_{i-k^\star+1}}^\star$. This layer shows how transformers can learn a selective induction head; a mechanism that adapts to the input sequence by copying the token corresponding to the argmax of some quantity extracted by the previous layers and stored in the embeddings.

## 4.3 EQUIVALENCE WITH MAXIMUM LIKELIHOOD

The disentangled transformer we propose does not rely on likelihood. Due to the normalization applied by the softmax function, the model computes the sum of normalized probabilities $\tilde{p}$. For the inference to be accurate, the cumulative sum corresponding to the correct lag must exceed those of any other lag. This fact is formalized in terms of expected values in the following claim:

**Claim 1.** *Let $\mathcal{K}$ be a subset of integers and $X_t$ an interleaved Markov chain of lag $k \in \mathcal{K}$, then, for $r \in \mathcal{K}$ and $i \ge \hat{k}$,*

$$\mathbb{E}\left[\frac{X_{i-r}^\top P^\star X_i}{\sum_{l \in \mathcal{K}} X_{i-l}^\top P^\star X_i}\right] \le \mathbb{E}\left[\frac{X_{i-k}^\top P^\star X_i}{\sum_{l \in \mathcal{K}} X_{i-l}^\top P^\star X_i}\right].$$

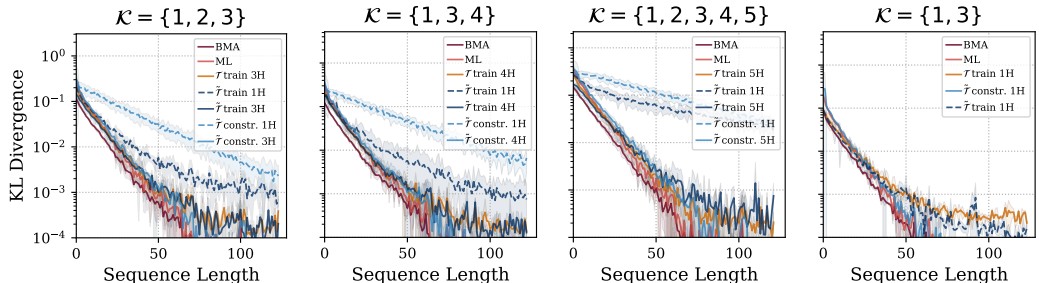

Figure 3: **Performance of our constructed transformers, trained transformers, and theoretical estimator (BMA, ML).** First plot: lags 1,2,3. Second: the model solve the task with non-contiguous lags. Third: the model is effective with additional lags. Fourth: one head is enough for two lags.

While specific cases (e.g., two lags, no normalization, or independent lags) are proven in App. B, we leave the complete proof of Claim 1 for future work. However, we provide empirical validation of this claim in App. F. Due to ergodicity, the average $\frac{1}{T}\sum_{i=1}^{T}\tilde{p}_{i,r}$ converges to its expected value, and for large enough $T$ it is higher for the correct lag. Therefore applying the exponential and scaling the temperature $\beta$ leads to the same result as MLE in the asymptotic limit, as shown in Fig. 4c.

### 4.4 GENERALIZATIONS AND SPECIAL CASES

**Single-head transformers:** the construction above allows the model to store all the past transition probabilities in the embedding of the current token by scaling the number of heads with the total number of lags $K$. Since all heads perform the same operation, reducing the number of heads implements an equivalent algorithm but with *worse sample complexity* compared to ML, as some past transitions are discarded. Thus, a 3-layer single-head transformer still solves the task by implementing the algorithm in Proposition 1, but it only uses $\frac{T-\hat{k}-1}{K}+1$ samples to estimate the correct lag.
**Non-contiguous lags:** in App. I, we provide examples of constructions to handle non-contiguous lags, where the core approach remains similar to that in Sec. 4.1. Depending on the specific case, the number of heads needed ranges between the number of lags and $\hat{k}-\min(\mathcal{K})+1$.
**Two lags:** by the construction in Sec. 4.1, handling two contiguous lags requires two attention heads in the second layer for optimal sample complexity. However, we provide in App. J an alternative construction for the third layer which enables a single-head model to match the performance of the two-head model for any two lags, whether contiguous or not (see empirical results in Fig.3).

## 5 EXPERIMENTS AND DISCUSSION

We conduct a series of experiments to empirically validate our construction and determine whether transformers trained via gradient descent learns it. **Setup.** We train 3-layer disentangled transformers ($\tilde{\mathcal{T}}$) and 3-layer standard transformers ($\mathcal{T}$) with learned positional and semantic embedding both with $\alpha = \sqrt{d_{QK}}$ using cross-entropy loss. At each step, we generate a fresh batch (size 256) of sequences (length 128) via Alg. 1, and train using Adam optimizer with fixed learning rate 0.001 and no weight decay. For the standard transformer embedding size we tested $128, 64$ and $d_{QK} = 32$. For the constructions, we fix $\beta = 100$ and $\lambda = 500$. We report $\mathcal{D}_{KL}$ between the true and predicted next-token distribution along the sequence. We generate different tasks with alphabet size $|\mathcal{S}| = 5$ (no differences are observed for other sizes) varying the number and values of lags: $\mathcal{K} = \{1, 2, 3\}$ (our example) and $\mathcal{K} = \{1, 2, 3, 4, 5\}$ for the case of contiguous lags (optimal number of heads 3 and 5 according to Proposition 1), $\mathcal{K} = \{1, 3, 4\}$ to show non-contiguous lags (4 heads needed, see App. I), and $\mathcal{K} = \{1, 3\}$ for the special case of two lags.

**Main results.** We observe in Fig. 3 how the construction with optimal number of heads, indicated as $\tilde{\mathcal{T}}$ constr., matches the performance of the ML. Moreover, both the disentangled transformers trained ($\tilde{\mathcal{T}}$ train) and the standard one ($\mathcal{T}$ train) match the performance of the theoretical construction. Interestingly, we observe that when the number of heads is fixed to 1 the trained transformers can find solutions which perform better than the construction: this indicates the existence of more efficient, yet difficult to interpret, ways of aggregating the transition probabilities, and that gradient descent can find them. Finally, we illustrate how for the case of two lags (right in Fig. 3) our construction

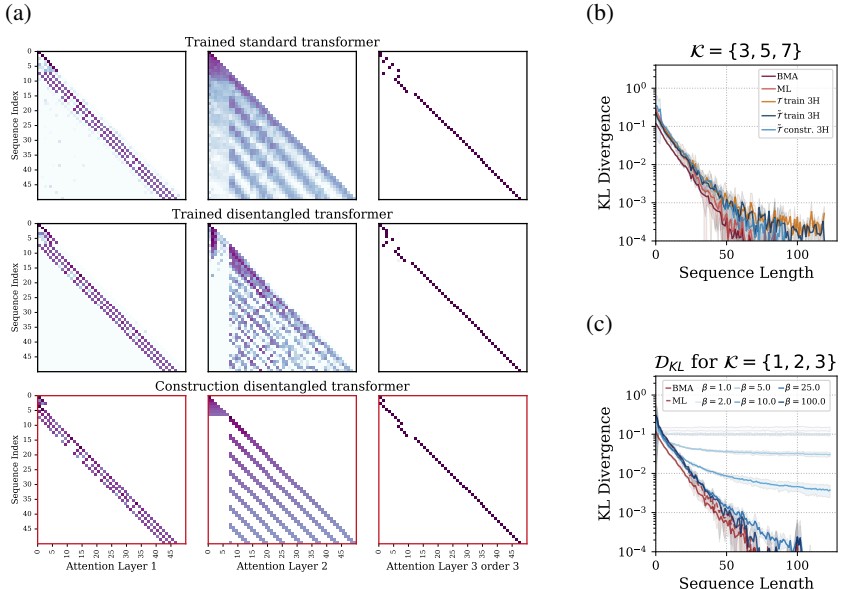

Figure 4: **Attention maps trained vs constructed transformer.** Fig. 4a reports the heatmaps of the attention maps of the trained standard and disentangled transformer and our construction for lags 3,5,7 and Fig. 4b. The first and second layers display a remarkable similarity and the second layers show a similar diagonal structure. Fig. 4c shows how the estimator in Eq. (3) matches ML for large $\beta$.

with single head (detailed in App J) attains the optimal sample complexity, while the construction in App. I would require 2 heads. Moreover, in this case, it appears that the trained transformers can obtain performances closer to the BMA rather than the ML for small sequence lengths.

**Trained vs constructed attention maps.** In Fig. 4a we report the attention maps for the three layers for $\mathcal{K} = \{3, 5, 7\}$ for trained standard transformer (top), trained disentangled transformer (middle) and our construction (bottom, see construction in App. I).The attention maps of the first and third layers are nearly identical between the trained and theoretical models, with these layers functioning precisely as expected from the theoretical construction. Notably, the attention entries of the first layer are proportional to $\log P^\star$, even when the model is trained from scratch, for both the disentangled and standard cases. For the second layer (aggregation), the trained transformers converge to a slightly different structure, likely because aggregation is a combinatorial problem with multiple valid implementations. Nevertheless, a clear diagonal pattern emerges, closely resembling our construction. Furthermore, as demonstrated in Fig. 4b, all models achieve comparable performance on the task. These findings strongly suggest that the trained transformers find a solution that aligns closely with our construction. Remarkably, we also show that standard transformers trained with learned positional and semantic embeddings and attention parameterized by $Q, K, V$ produce attention maps in agreement with our construction. This provides compelling evidence that our construction is not merely a byproduct of the disentangled transformer's architecture but it is also implemented by standard transformers. Interestingly, even with an embedding dimension (64) smaller than the sequence length (128), standard transformers efficiently store and utilize past transitions while matching optimal performance.

## 6 CONCLUSIONS

We introduced a novel synthetic task based on interleaved Markov chains to study how attention-only transformers perform in-context causal structure selection. Our findings demonstrated that a 3-layer transformer can solve this task with near-optimal sample complexity, effectively showcasing the emergence of selective induction heads, attention circuits that aggregate past information and select the correct causal structure. Moreover, we provided a fully interpretable construction of a disentangled transformer implementing these circuits to solve the task, and empirically verified that both disentangled and standard transformers trained with Adam closely align with this construction. Finally, we theoretically analyze the algorithm implemented by this construction showing that, in certain cases, it asymptotically converges to ML.

ACKNOWLEDGMENTS

This work was supported by the Swiss National Science Foundation (grant number 212111) and partially funded by an unrestricted gift from Google.

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

**Organization of the Appendix.** The Appendix is organized as follows. In App. A, we present a detailed review of related work on in-context learning and mechanistic interpretability. App. B extends the statistical analysis of the estimator implemented by the transformer in Prop.1 and includes omitted proofs. App. C reports additional experiments and discussions about using more than 3 layers and varying the number of heads in the second layer. Additional attention maps for different tasks and both disentangled and trained transformers as well as the construction are provided in App. D. App. F includes several experiments to validate Claim 1. App. E details the algorithm used to generate the interleaved Markov chains. A complete construction for the contiguous case, along with detailed intuitions, is given in App. G. App. H discusses an alternative third layer construction using positional embedding. The construction for non-contiguous lags is presented in App. I. Finally, App. J explains the single-head construction for the case of two lags.

## A  RELATED WORK

Following the initial empirical observations of the emergent in-context learning capabilities of transformers (Brown, 2020), several works have attempted to understand this phenomenon. Xie et al. (2021) sought to formulate in-context learning as Bayesian inference, while Garg et al. (2022) studied the ability of transformers to learn simple functions, such as linear models or multilayer perceptrons, in context. A subsequent line of work (Akyürek et al., 2022; Bai et al., 2024; Von Oswald et al., 2023a;b; Raventós et al., 2024) shows that transformer layers might implement gradient descent to solve in-context linear regression. Ahn et al. (2023) extend this idea to higher-order algorithms. Importantly, Olsson et al. (2022) postulate that in-context learning is tied to the emergence of induction heads. Bietti et al. (2024), subsequently extended this idea, showing the development of induction heads to learn bigrams in-context and showcasing a connection with associative memories. Our work is inspired by foundational efforts in causal interpretability and mechanistic understanding of neural networks, particularly transformers (Olsson et al., 2022). These include studies on causal mediation analysis (Mueller et al., 2024), causal abstraction as a theoretical framework (Geiger et al., 2023), and the broader scope of mechanistic interpretability in language models (Saphra & Wiegreffe, 2024). More closely related to our work is the literature analyzing transformers through the lens of Markov chains. In particular, Nichani et al. (2024) shows how transformers trained on sequences generated by Markov chains on a graph learn simple circuits to capture the underlying causal structure and implement the Bayes-optimal solution by estimating transition probabilities in context. Similarly, Edelman et al. (2024) illustrate the formation of statistical induction heads that accurately compute posterior probabilities based on bigram statistics. Makkuva et al. (2024) used Markov chains to study the loss landscape of transformers, while Rajaraman et al. (2024) shows that a constant depth is sufficient to learn k-th order Markov chains. Along this line, Svete & Cotterell (2024) demonstrates that transformers with hard or sparse attention can exactly represent any n-gram model. Extending this framework to Hidden Markov Models, Hu et al. (2024) highlights the limitations of transformers in learning such models compared to RNNs.

## B  STATISTICAL ANALYSIS OF OUR PREDICTOR

For our estimator to select the correct lag, the following inequalities must hold for a lag $k$ and a sequence of length $T$ generated accordingly:

$$\sum_{i=1}^{T} \tilde{p}_{i,k} > \sum_{j=1}^{T} \tilde{p}_{j,r} \quad \forall, r \neq k; \text{and}; r \in \mathcal{K}.$$

These results enable us to recover the MLE estimator in the high-temperature limit and approximate the BMA at finite temperatures. Assuming the process is ergodic, and by taking the limit of the inequality above, we require the following condition:

$$\mathbb{E}\Big[\frac{X_{i-r}^{\top} P^{\star} X_i}{\sum_{l \in \mathcal{K}} X_{i-l}^{\top} P^{\star} X_i}\Big] \leq \mathbb{E}\Big[\frac{X_{i-k}^{\top} P^{\star} X_i}{\sum_{l \in \mathcal{K}} X_{i-l}^{\top} P^{\star} X_i}\Big] \quad \text{for } r \in \mathcal{K} \text{ and } i \geq \max(\mathcal{K}),$$

as formalized in Claim 1.

We leave the complete proof of this result as future work, but we have fully validated it empirically in Section F. We provide here the proofs of Claim 1 for three specific cases.

**Two-lag case.** In the case of two lags, we can show the following general result for any two distributions, $P$ and $Q$, over $1, \ldots, k$ for $k \geq 0$.

**Lemma 1.**

$$\sum_{i=1}^{k} \frac{P(i)Q(i)}{P(i)+Q(i)} \leq \sum_{i=1}^{k} \frac{P(i)^2}{P(i)+Q(i)}. \tag{7}$$

*Proof.* We first show that

$$\sum_{i=1}^{k} \frac{P(i)^2 - Q(i)^2}{P(i)+Q(i)} = \sum_{i=1}^{k} \frac{(P(i)-Q(i))(P(i)+Q(i))}{P(i)+Q(i)} = \sum_{i=1}^{k}(P(i)-Q(i)) = 0.$$

Then, by using Cauchy-Schwarz inequality, we obtain:

$$\sum_{i=1}^{k} \frac{P(i)Q(i)}{P(i)+Q(i)} \leq \sqrt{\sum_{i=1}^{k} \frac{P(i)^2}{P(i)+Q(i)}} \sqrt{\sum_{i=1}^{k} \frac{Q(i)^2}{P(i)+Q(i)}} = \sum_{i=1}^{k} \frac{P(i)^2}{P(i)+Q(i)}.$$

$\square$

The result in the two-lag case follows directly. Let $\mu$ denote the distribution of the lag-$k$ interleaved process $X_t$, (i.e, $\mu(s_i, s_j, s_k) = \mathbb{P}(X_i = s_i, X_j = s_j, X_k = s_k)$). For any lag $\{r\}$ we have

$$\mathbb{E}\left[\frac{X_{i-r}^{\top} P^{\star} X_i}{X_{i-r}^{\top} P^{\star} X_i + X_{i-k}^{\top} P^{\star} X_i}\right] = \sum_{s_{i-k}, s_{i-r}, s_i} \mu(s_{i-k}, s_{i-r}, s_i) \frac{P_{s_{i-k}, s_i} P_{s_{i-r}, s_i}}{P_{s_{i-k}, s_i} + P_{s_{i-r}, s_i}}$$

$$= \sum_{s_{i-k}, s_{i-r}} \mu(s_{i-k}, s_{i-r}) \sum_{s_i} \frac{P_{s_{i-k}, s_i} P_{s_{i-r}, s_i}}{P_{s_{i-k}, s_i} + P_{s_{i-r}, s_i}}$$

By applying Lemma 1, we directly obtain

$$\mathbb{E}\left[\frac{X_{i-r}^{\top} P^{\star} X_i}{X_{i-r}^{\top} P^{\star} X_i + X_{i-k}^{\top} P^{\star} X_i}\right] \leq \mathbb{E}\left[\frac{X_{i-k}^{\top} P^{\star} X_i}{X_{i-r}^{\top} P^{\star} X_i + X_{i-k}^{\top} P^{\star} X_i}\right]$$

which proves Claim 1 in the case of two lags.

**Independent lags.** In the case where all lags in $\mathcal{K}$ are such that $(X_{i-l})_{l \in \mathcal{K}}$ are independent, we can prove Claim 1. Indeed, in this case, the distribution of the observed lags can be factorized as $\mu((s_{i-l})_{l \in \mathcal{K}}) = \prod_{l \in \mathcal{K}} \mu(s_{i-l})$. Thus we have

$$\mathbb{E}\left[\frac{X_{i-r}^{\top} P^{\star} X_i - X_{i-k}^{\top} P^{\star} X_i}{\sum_{l \in \mathcal{K}} X_{i-l}^{\top} P^{\star} X_i}\right] = \sum_{s_i, (s_{i-l})_{l \in \mathcal{K}}} \mu((s_{i-l})_{l \in \mathcal{K}}) \frac{P_{s_{i-k}, s_i}(P_{s_{i-r}, s_i} - P_{s_{i-k}, s_i})}{\sum_{l \in \mathcal{K}} P_{s_{i-l}, s_i}}$$

$$= \sum_{s_i, (s_{i-l})_{l \in \mathcal{K}}} \mu((s_{i-l})_{l \in \mathcal{K}, l \neq k, r}) \mu(s_{i-k}) \mu(s_{i-r}) \frac{P_{s_{i-k}, s_i}(P_{s_{i-r}, s_i} - P_{s_{i-k}, s_i})}{\sum_{l \in \mathcal{K}} P_{s_{i-l}, s_i}}$$

Then, by observing that $a(a-b) + b(b-a) = (a-b)^2 \geq 0$, the result follows from:

$$2 \sum_{s_{i-r}, s_{i-l}} \mu(s_{i-k}) \mu(s_{i-r}) \frac{P_{s_{i-k}, s_i}(P_{s_{i-r}, s_i} - P_{s_{i-k}, s_i})}{\sum_{l \in \mathcal{K}} P_{s_{i-l}, s_i}}$$

$$= \sum_{s_{i-r}, s_{i-l}} \mu(s_{i-k}) \mu(s_{i-r}) \frac{P_{s_{i-k}, s_i}(P_{s_{i-r}, s_i} - P_{s_{i-k}, s_i}) + P_{s_{i-r}, s_i}(P_{s_{i-k}, s_i} - P_{s_{i-r}, s_i})}{\sum_{l \in \mathcal{K}} P_{s_{i-l}, s_i}}$$

$$= \sum_{s_{i-r}, s_{i-l}} \mu(s_{i-k}) \mu(s_{i-r}) \frac{(P_{s_{i-r}, s_i} - P_{s_{i-k}, s_i})^2}{\sum_{l \in \mathcal{K}} P_{s_{i-l}, s_i}} \geq 0$$

We observe that similar techniques can be applied to prove Claim 1 in the case of a symmetric Markov kernel $P^{\star}$.

**No normalization case.** Due to various reasons, including the normalization of the softmax in the attention layer, our estimator relies on a score computed by aggregating the normalized probabilities $\tilde{p}_{i,k}$. If we were to use the unnormalized probabilities, we could rely on the following result, which simplifies Claim 1 by excluding the normalization step.

**Lemma 2.** *Let $\mathcal{K}$ be a subset of integers and $X_t$ a stationary interleaved Markov chain of lag $k \in \mathcal{K}$, then*

$$\mathbb{E}[X_{i-r}^\top P^\star X_i] \leq \mathbb{E}[X_{i-k}^\top P^\star X_i] \quad \text{for } r \in \mathcal{K} \text{ and } i \geq \max(\mathcal{K}). \tag{8}$$

*Proof.*

$$\mathbb{E}[X_{i-r}^\top P^\star X_i] = \sum_{s_{i-r}, s_{i-k}, s_i} \mu(s_{i-r}, s_{i-k}, s_i) P_{s_{i-r}, s_i}$$

$$= \sum_{s_{i-r}, s_{i-k}, s_i} \mu(s_{i-r}, s_{i-k}) P_{s_{i-k}, s_i} P_{s_{i-r}, s_i}$$

$$\leq \sum_{s_i} \sqrt{\sum_{s_{i-r}, s_{i-k}} \mu(s_{i-r}, s_{i-k}) P_{s_{i-k}, s_i}^2} \sqrt{\sum_{s_{i-r}, s_{i-k}} \mu(s_{i-r}, s_{i-k}) P_{s_{i-r}, s_i}^2}$$

$$\leq \sum_{s_i} \sqrt{\sum_{s_{i-k}} (\sum_{s_{i-r}} \mu(s_{i-r}, s_{i-k})) P_{s_{i-k}, s_i}^2} \sqrt{\sum_{s_{i-r}} (\sum_{s_{i-k}} \mu(s_{i-r}, s_{i-k})) P_{s_{i-r}, s_i}^2},$$

where the inequality follows from the Cauchy-Schwarz inequality.

Assuming that $\mu(s_{i-r}, s_{i-k})$ is a coupling of the stationary measure $\pi$, we then have:

$$\mathbb{E}[X_{i-r}^\top P^\star X_i] \leq \sum_{s_i} \sqrt{\sum_{s_{i-k}} \pi(s_{i-k}) P_{s_{i-k}, s_i}^2} \sqrt{\sum_{s_{i-r}} \pi(s_{i-r}) P_{s_{i-r}, s_i}^2}$$

$$= \sum_{s_i, s_{i-k}} \pi(s_{i-k}) P_{s_{i-k}, s_i}^2$$

$$= \mathbb{E}[X_{i-k}^\top P^\star X_i].$$

It remains to prove that $\mu(s_{i-r}, s_{i-k})$ is a coupling of the stationary measure $\pi$.

First, let's assume that $r$ and $k$ are such that $X_{i-r}$ and $X_{i-k}$ are independent. In this case $\mu(s_{i-r}, s_{i-k}) = \mu(s_{i-r})\mu(s_{i-k})$ and we have both that $\sum_{s_{i-r}} \mu(s_{i-r}, s_{i-k})) = \mu(s_{i-k})$ and $\sum_{s_{i-k}} \mu(s_{i-r}, s_{i-k}) = \mu(s_{i-r})$.

Alternatively, if $r$ and $k$ are such that $X_r$ and $X_k$ come from the same Markov Chain with $r > k$. We have thus that $X_{i-r} \sim \mu$ and $X_{i-k}|X_{i-r} \sim s_{i-r}^\top P^l$ for some $l \geq 0$ and $\mu(s_{i-r}, s_{i-k}) = \pi(s_{i-r}) s_{i-r}^\top P^l s_{i-k}$. Since $P$ is a stochastic matrix, summing over $s_{i-k}$ gives $\sum_{s_{i-k}} \mu(s_{i-r}, s_{i-k}) = \mu(s_{i-r}) = \pi(s_{i-r})$. Finally, by definition of the stationary distribution $\pi$, summing over $s_{i-r}$ gives $\sum_{s_{i-r}} \mu(s_{i-r}, s_{i-k}) = \pi(s_{i-k})$. $\square$

## C  SCALING HEADS AND LAYERS

In this section, we investigate how varying the number of heads in the second layer and the number of layers affects the model's performance. We train standard transformers with learned positional and semantic embeddings in the same setup as reported in Section 5. In Figure 5 (left) we consider the task given by $\mathcal{K} = \{1, 2, 3\}$ and first show the behavior of the model with 2 layers and different combinations of heads $[1, 1], [3, 1], [1, 3], [3, 3]$[1], the results show that transformers with 2 layers can't solve the task. Second, we show that increasing the number of layers beyond 3 does not change the performance. In Figure 5 (right) instead we consider the task defined by $\mathcal{K} = \{1, 2, 3, 4, 5\}$ and train transformers with fewer, equal to, or more than $K$ heads. As predicted by our construction increasing the number of heads leads to performances that get closer to the maximum likelihood up to having the number of heads equal to the number of lags in the set $K$. Beyond this point adding more heads does not improve performance, this is expected as ML is optimal. Figures 6a,6b,6c illustrates the attention maps for a 3-layer transformer with only $1, 2, 3$ heads respectively in the second layer, despite the task having 5 lags. Remarkably, even with fewer than K heads, the layers remain consistent with our theoretical construction, displaying analogous patterns: the first layer extracts transition probabilities, the second aggregates them, and the third implements the selective head. However, in the case of fewer heads, the second layer appears to find an efficient way to superpose information—a mechanism we could not yet interpret. Understanding this behaviour in the second layer remains an open question for future work.

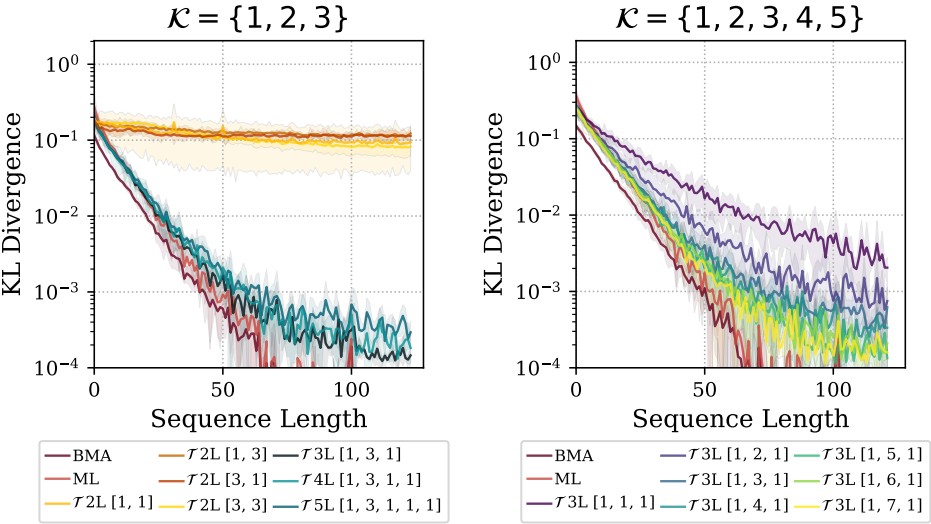

Figure 5: **(left) Scaling number of layers:** we train standard transformers with learned position and semantic embeddings. Transformers with 2 layers can't solve the task for any combination of heads. Transformers with more than 3 layers achieve the same performance as for 3 layers. **(right) Scaling number of heads in the second layer:** we train standard transformers with learned position and semantic embeddings increasing the number of heads in the second layer. As predicted by the construction increasing the number of layers leads to performance closer to the Maximum Likelihood.

---

[1]With this notation we intend the following [#heads layer 1, . . . , #heads layer L]

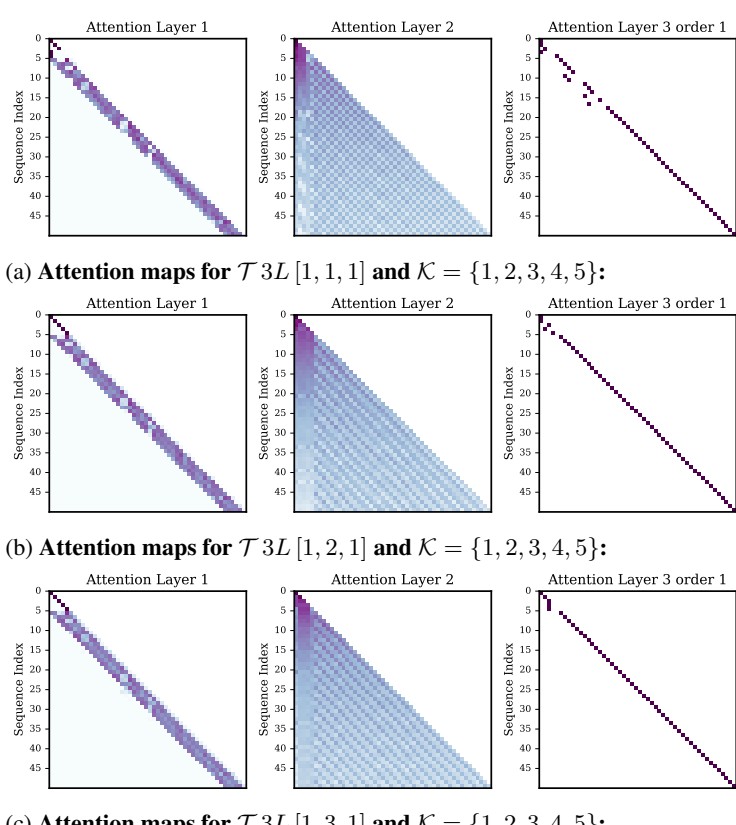

(a) **Attention maps for $\mathcal{T}3L\,[1,1,1]$ and $\mathcal{K} = \{1,2,3,4,5\}$:**

(b) **Attention maps for $\mathcal{T}3L\,[1,2,1]$ and $\mathcal{K} = \{1,2,3,4,5\}$:**

(c) **Attention maps for $\mathcal{T}3L\,[1,3,1]$ and $\mathcal{K} = \{1,2,3,4,5\}$:**

Figure 6: **Attention maps for different heads in the second layer and $\mathcal{K} = \{1,2,3,4,5\}$:** we observe how even with fewer heads the transformer learns layers which are consistent with the operations in our construction. In particular the first layer is still extracting the transition probabilities, the second is aggregating them and the third one implements the selective head.

## D  ADDITIONAL ATTENTION MAPS PLOTS

As an additional confirmation for our construction, we report here comparison of the attention maps after softmax for the task introduced in Figure 1. We compare, trained standard 3-layer attention-only transformer with learned positional encoding and one attention head per layer, trained disentangled transformer and our construction. The standard transformer was trained in the same setup already introduced in Section 5. In Figure 7 we train on data with $\mathcal{K} = 1, 2$, we observe a remarkable similarity between the attention maps of our construction and the trained transformer. This further confirms that the disentangled transformer is a good proxy to study the residual stream and the flow of information inside the transformer in a more interpretable way. Moreover, it confirms that our construction is realistic and aligns with what transformers learn in practice by gradient descent. Moreover, In order to showcase the adaptivity in-context of the selective induction head depending on the true lag of the input sequence, in Figures 8, 9, 10 we train on lags $\mathcal{K} = \{1, 2, 3\}$ and test on sequences generated with each one of the training lags, similarly in Figures 11, 12, 13 we train on lags $\mathcal{K} = \{1, 2, 3\}$ and test on each. As expected, the third layer adapts to the input sequence selecting the correct lag and copying the correspondent token via the selective induction head.

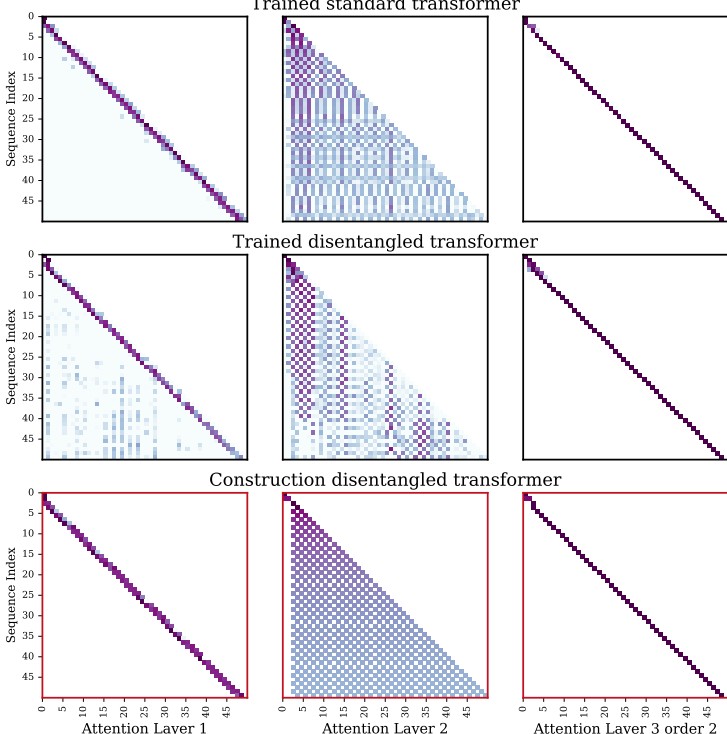

Figure 7: **Attention maps $\mathcal{K} = \{1, 2\}$ and true lag $k = 2$.**

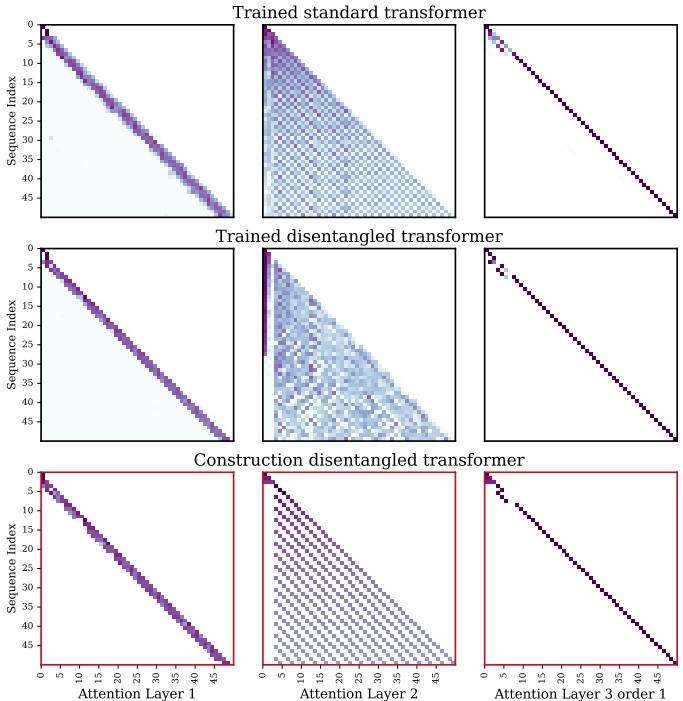

Figure 8: **Attention maps** $\mathcal{K} = \{1, 2, 3\}$ **and true lag** $k = 1$.

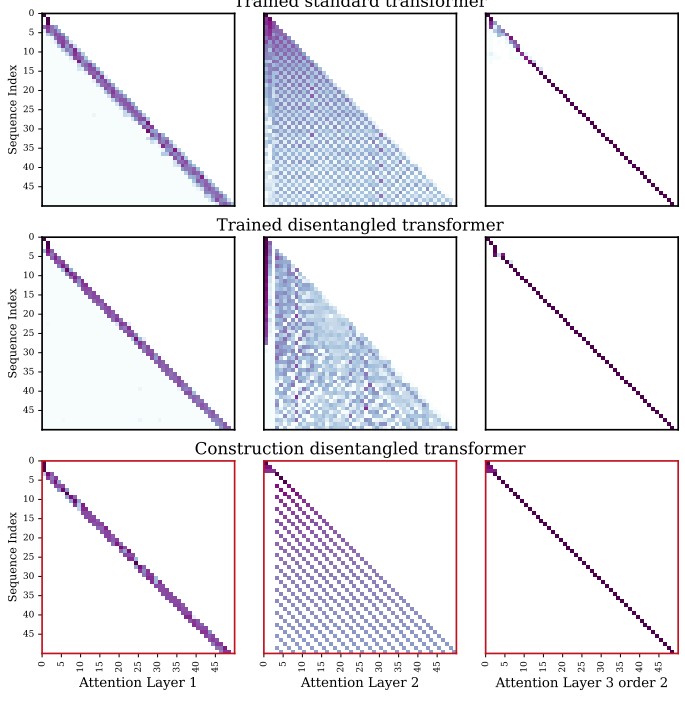

Figure 9: **Attention maps** $\mathcal{K} = \{1, 2, 3\}$ **and true lag** $k = 2$.

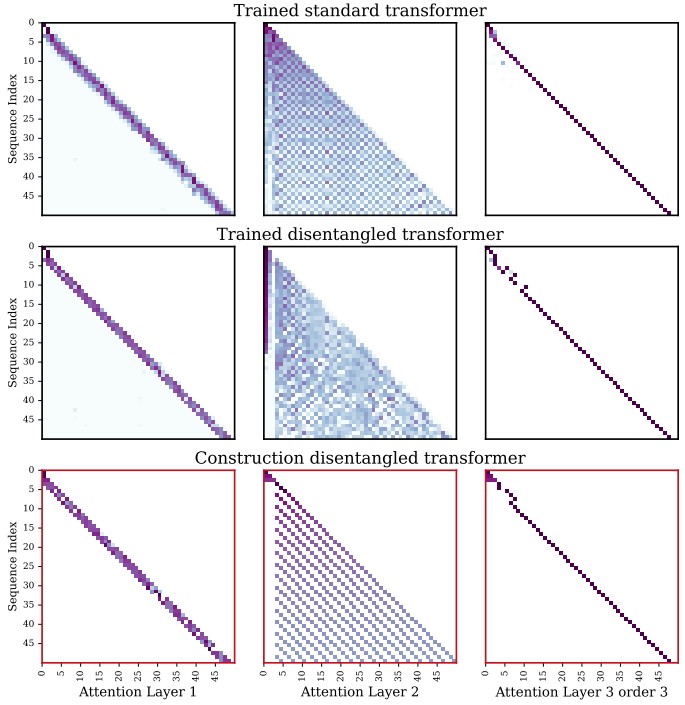

Figure 10: **Attention maps** $\mathcal{K} = \{1, 2, 3\}$ **and true lag** $k = 3$.

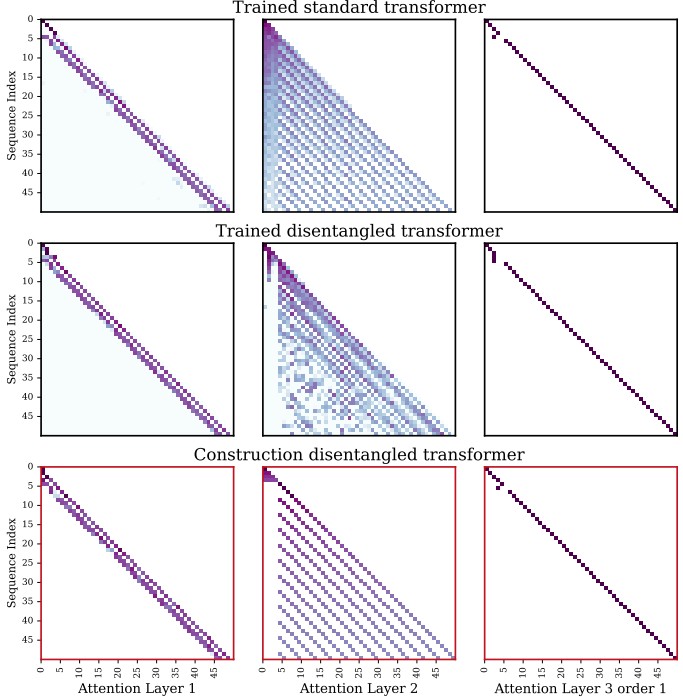

Figure 11: **Attention maps** $\mathcal{K} = \{1, 3, 4\}$ **and true lag** $k = 1$.

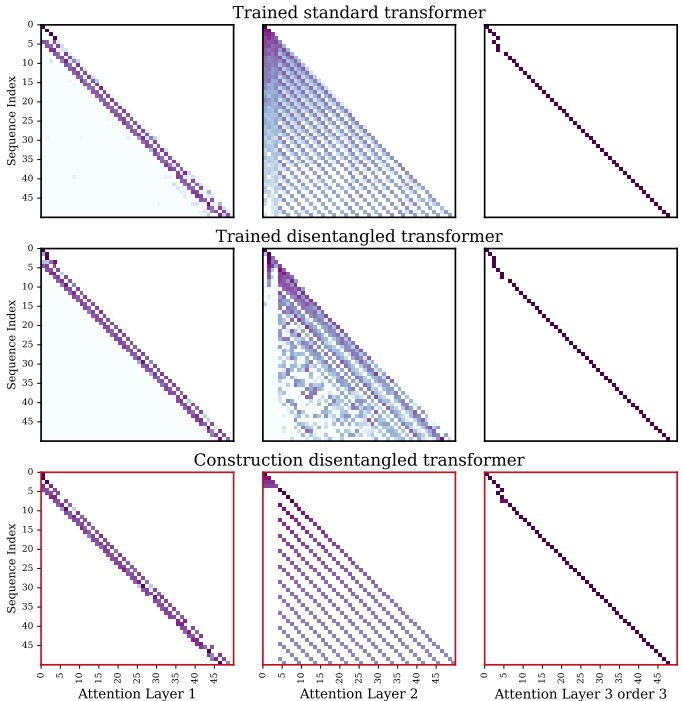

Figure 12: **Attention maps** $\mathcal{K} = \{1, 3, 4\}$ **and true lag** $k = 3$.

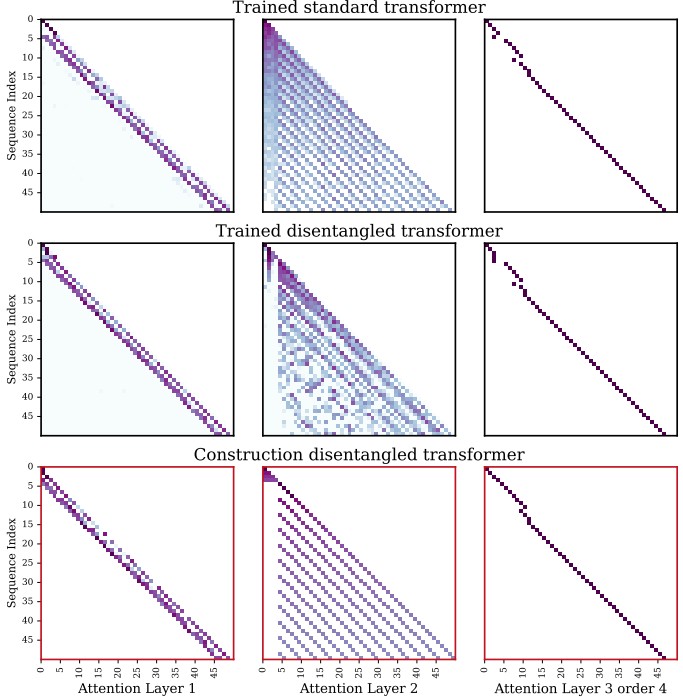

Figure 13: **Attention maps** $\mathcal{K} = \{1, 3, 4\}$ **and true lag** $k = 4$.

# E    TASK DETAILS

In this section, we illustrate the algorithm we used to generate batches of new samples at each iteration.

---

**Algorithm 1** Generate Dataset of N Sequences from Interleaved Markov Chains

---

**Require:** $N$ (sequences), $T$ (length), $\mathcal{S}$ (state space), $\mathcal{K}$ (set of lag values), $P^*$ (transition matrix)
**Ensure:** Dataset $\mathcal{D}$ of $N$ sequences
1:  $\mathcal{D} \leftarrow \emptyset$
2:  $\pi \leftarrow$ stationary distribution of $P^*$
3:  **for** $i = 1$ to $N$ **do**
4:      $k \leftarrow \text{Uniform}(\mathcal{K})$                                          ▷ Randomly select lag for this sequence
5:      $X_0 \leftarrow$ Sample from $\pi$                                                        ▷ Initialize first state
6:      $S \leftarrow [X_0]$                                                                    ▷ Initialize sequence
7:      **for** $t = 1$ to $T - 1$ **do**
8:          **if** $t < \hat{k}$ **then**
9:              $X_t \leftarrow$ Sample from $\pi$                                          ▷ Sample from stationary distribution
10:         **else**
11:             $X_t \leftarrow$ Sample from $P^*[X_{t-k}, :]$                            ▷ Transition based on lag $k$
12:         Append $X_t$ to $S$
13:     Add $S$ to $\mathcal{D}$
        **return** $\mathcal{D}$

---

# F    EMPIRICAL VALIDATION OF CLAIM 1

To empirically validate Claim 1 we first sample a set of 12 lags uniformly between 1 and 30; we then sample 1000 different transition matrices and for each matrix and each lag 1000 sequences of length 1000 according to the respective Interleaved Markov chain. For each lag and each set of sequences we then compute the expectation in Claim 1 by averaging the last transition in each sampled sequences. We then compute the following quantity:

$$\mathbb{E}\Big[\frac{X_{i-k}^\top P^\star X_i}{\sum_{l\in\mathcal{K}} X_{i-l}^\top P^\star X_i}\Big] - \max_{\substack{r\neq k \\ r\in\mathcal{K}}} \mathbb{E}\Big[\frac{X_{i-r}^\top P^\star X_i}{\sum_{l\in\mathcal{K}} X_{i-l}^\top P^\star X_i}\Big] \qquad (9)$$

and report it in the histogram in Fig.14. We can see that all values in the histogram are positive therefore confirming our claim. Similarly, the results in Fig.15 report the quanity in the claim for each single lag. As per our claim, the expected normalized transition probabilities of the true lag is always larger than the same quantity for any other lag. As a further confirmation of the claim, in Fig.17 and Fig.18 we report the cumulative average of the normalized transition probabilities along the sequence for a single sequence. We observe that even with few samples (small $t$) the cumulative average for the true order is always larger than the same quantity for the other lags.

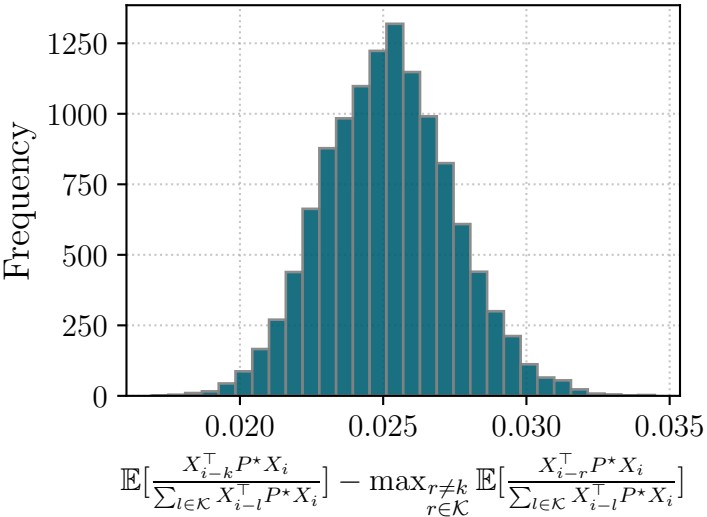

Figure 14: **Difference of the expected Normalized Transition Probabilities for the true lag and the maximum over all other lags for $|\mathcal{S}| = 10$.** The histogram shows how the quantity in Eq. 9 is always positive.

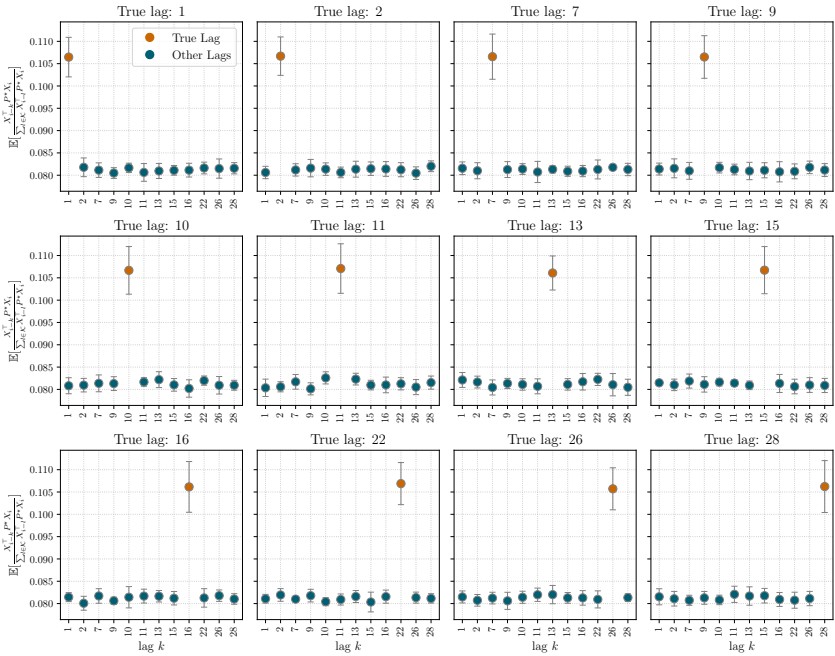

Figure 15: **Expected Normalized Transition Probabilities for $|\mathcal{S}| = 10$:** The sampled set of lags is $\mathcal{K} = \{1, 2, 7, 9, 10, 11, 13, 15, 16, 22, 26, 28\}$, we sampled 10 different transition matrices and for each lag and each matrix sampled 1000 sequences of length 1000. The expected normalized transition probability is always larger for the true lag.

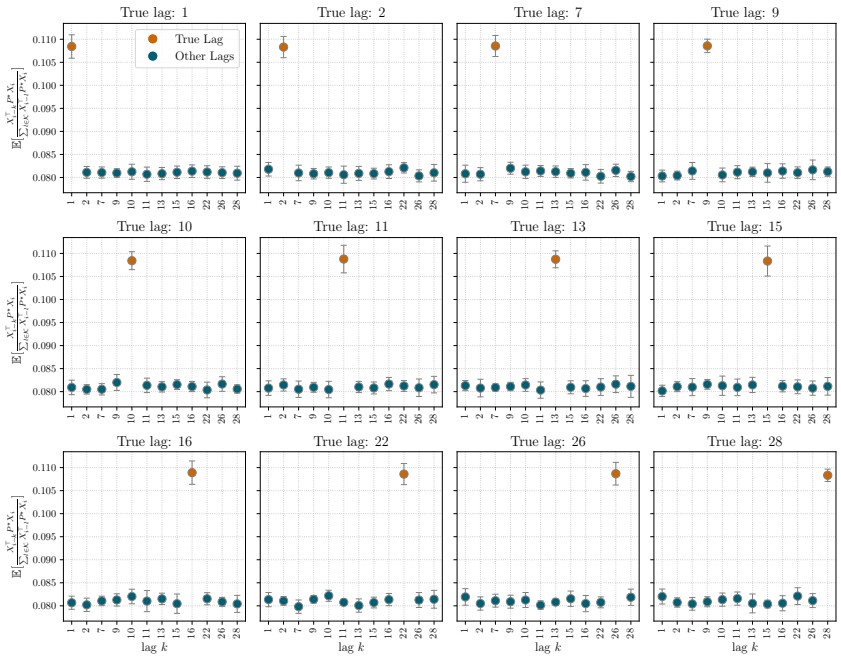

Figure 16: **Expected Normalized Transition Probabilities for** $|\mathcal{S}| = 25$**:** The sampled set of lags is $\mathcal{K} = \{1, 2, 7, 9, 10, 11, 13, 15, 16, 22, 26, 28\}$, we sampled 10 different transition matrices and for each lag and each matrix sampled 1000 sequences of length 1000. The expected normalized transition probability is always larger for the true lag.

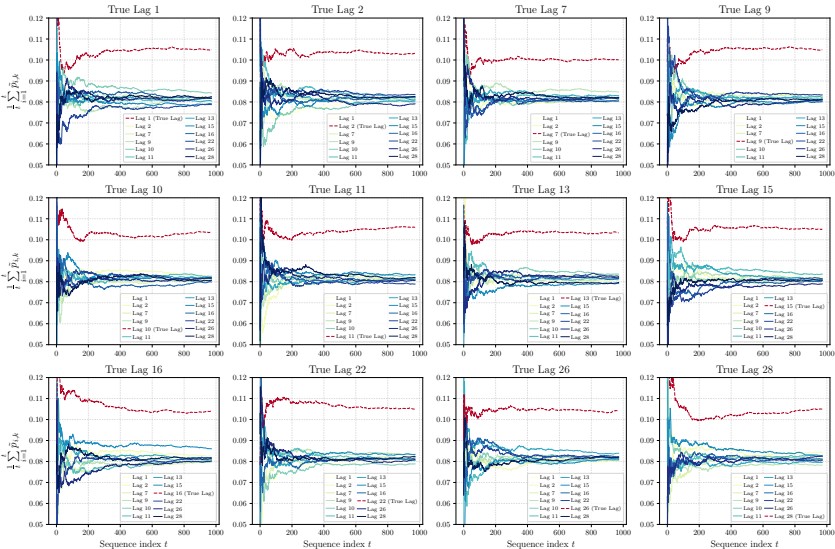

Figure 17: **Cumulative average of Normalized Transition Probabilities for** $|\mathcal{S}| = 10$**:** The sampled set of lags is $\mathcal{K} = \{1, 2, 7, 9, 10, 11, 13, 15, 16, 22, 26, 28\}$, we report one sequence sampled according to one the transition matrix. The cumulative average of normalized transition probability quickly becomes larger for the true lag.

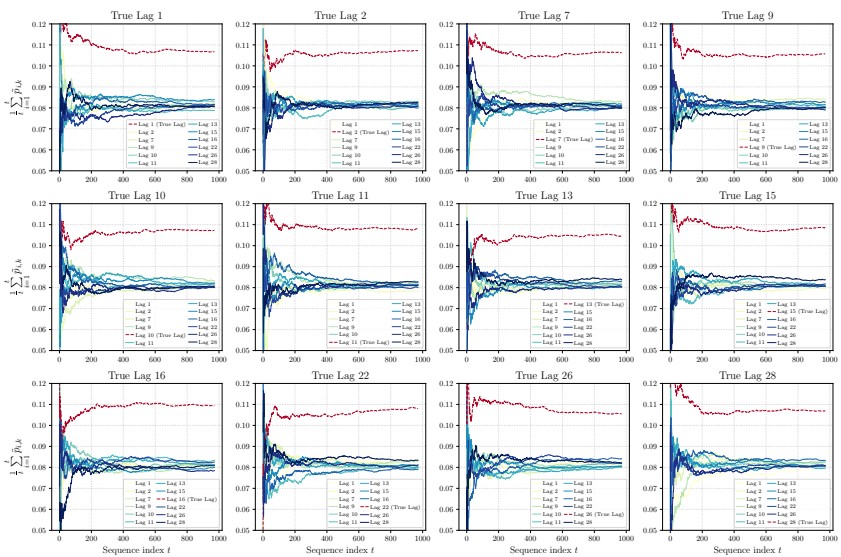

Figure 18: **Cumulative average of Normalized Transition Probabilities for $|\mathcal{S}| = 25$:** The sampled set of lags is $\mathcal{K} = \{1, 2, 7, 9, 10, 11, 13, 15, 16, 22, 26, 28\}$, we report one sequence sampled according to one the transition matrix. The cumulative average of normalized transition probability quickly becomes larger for the true lag.

## G  COMPLETE CONSTRUCTION FOR CONTIGUOUS LAGS.

**First layer: extraction of transition probabilities.** The first attention matrix, $\tilde{A}^{(1)}$, consists of two blocks: the first block operates on the semantic component of the input tokens, learning the transpose of the logarithm of the transition matrix. The second block $A^{(1)}$ learns the causal relationships induced by each possible lag $s_{i-k} \to s_i$ for $k \in \mathcal{K}$:

$$\widetilde{A}^{(1)} = \begin{pmatrix} \log P^\top & 0 \\ 0 & A^{(1)} \end{pmatrix}$$

$$A_{ij}^{(1)} = \begin{cases} +\lambda & \text{if} \quad i - j \in \mathcal{K} \\ -\lambda & \text{if} \quad i - j \notin \mathcal{K} . \end{cases}$$

$$A^{(1)} = \begin{pmatrix} -\lambda & -\lambda & -\lambda & -\lambda & -\lambda & -\lambda & -\lambda & -\lambda & -\lambda & -\lambda \\ +\lambda & -\lambda & -\lambda & -\lambda & -\lambda & -\lambda & -\lambda & -\lambda & -\lambda & -\lambda \\ +\lambda & +\lambda & -\lambda & -\lambda & -\lambda & -\lambda & -\lambda & -\lambda & -\lambda & -\lambda \\ +\lambda & +\lambda & +\lambda & -\lambda & -\lambda & -\lambda & -\lambda & -\lambda & -\lambda & -\lambda \\ -\lambda & +\lambda & +\lambda & +\lambda & -\lambda & -\lambda & -\lambda & -\lambda & -\lambda & -\lambda \\ -\lambda & -\lambda & +\lambda & +\lambda & +\lambda & -\lambda & -\lambda & -\lambda & -\lambda & -\lambda \\ -\lambda & -\lambda & -\lambda & +\lambda & +\lambda & +\lambda & -\lambda & -\lambda & -\lambda & -\lambda \\ -\lambda & -\lambda & -\lambda & -\lambda & +\lambda & +\lambda & +\lambda & -\lambda & -\lambda & -\lambda \\ -\lambda & -\lambda & -\lambda & -\lambda & -\lambda & +\lambda & +\lambda & +\lambda & -\lambda & -\lambda \\ -\lambda & -\lambda & -\lambda & -\lambda & -\lambda & -\lambda & +\lambda & +\lambda & +\lambda & -\lambda \end{pmatrix}$$

We can compute the first layer's attention as:

$$[e_{s_i}, e_i]^\top \tilde{A}^{(1)} [e_{s_j}, e_j] = (\log P)_{s_j, s_i} + \lambda \operatorname{sign}(A_{ij}^{(1)})$$

and applying the softmax:

$$\mathcal{A}^{(1)}(h_{1:T}^{(0)}; \tilde{A}^{(1)})_{ij} = \frac{e^{(\log P)_{s_j, s_i} + \lambda \operatorname{sign}(A_{i,j}^{(1)})}}{\sum_{r=1}^{i} e^{(\log P)_{s_r, s_i} + \lambda \operatorname{sign}(A_{i,j}^{(1)})}} = \frac{e^{(\log P)_{s_j, s_i} + \lambda \operatorname{sign}(A_{ij}^{(1)})}}{\sum_{r \in \mathcal{K}} e^{(\log P)_{s_r, s_i} + \lambda} + \sum_{r \notin \mathcal{K}} e^{(\log P)_{s_r, s_i} - \lambda}} .$$

For $\lambda \to \infty$ (in practice, for $\lambda$ large enough) and having denoted $\tilde{p}_{i,k} := \frac{P_{s_{i-k}, s_i}}{\sum_{r \in \mathcal{K}, r < i} P_{s_{i-r}, s_i}}$ for $i > 1$,

$$\lim_{\lambda \to \infty} \mathcal{A}^{(1)}(h_{1:T}^{(0)}; \tilde{A}^{(1)})_{ij} = \begin{cases} \tilde{p}_{i, i-j} & \text{if } i - j \in \mathcal{K} \\ 1 & \text{if } i = j = 1 \\ 0 & \text{elsewhere} \end{cases} .$$

Therefore, the output at index $i$ after the first layer corresponds to a weighted average of the past tokens $h_{i-k}^{(0)}$ for $k \in \mathcal{K}$ where the weights are given by the normalized probabilities $\tilde{p}_{i,k}$:

$$\hat{h}_i^{(1)} = \operatorname{Attn}(h_{1:T}^{(0)}; \tilde{A}^{(1)})_i = \begin{cases} \sum_{j=1}^{i} \mathbb{1}\left[i - j \in \mathcal{K}\right] \frac{P_{s_j, s_i}}{\sum_{r \in \mathcal{K}} P_{s_r, s_i}} h_j^{(0)} & \text{if } i > 1 \\ h_1^{(0)} & \text{if } i = 1 \end{cases} \tag{10}$$

$$= \begin{cases} \sum_{k \in \mathcal{K}, k < i} \tilde{p}_{i,k} h_{i-k}^{(0)} & \text{if } i > 1 \\ h_1^{(0)} & \text{if } i = 1 \end{cases} \tag{11}$$

With the input vectors $h_i^{(0)}$ being the concatenation of the one-hot encoding of the state and position $[e_{s_i}, e_i]$, the first $|\mathcal{S}|$ entries of $\hat{h}_i^{(1)}$ correspond to $\tilde{s}_i = \sum_{k \in \mathcal{K}, k < i} \tilde{p}_{i,k} e_{s_{i-k}}$ for $i > 1$ and $\tilde{s}_1 = e_{s_1}$. The remaining entries, due to the one-hot positional encoding, directly copy the normalized transition probabilities for the transition $s_{i-k} \to s_i$ into the $|\mathcal{S}| + (i - k)$-th element of $\hat{h}_i^{(1)}$. To build intuition, we refer to the example in Eq. (12) where the colors highlight transition probabilities of the same lag:

$$\mathcal{A}^{(1)} = \begin{pmatrix} 1 & 0 & 0 & 0 & 0 & 0 & 0 & 0 & 0 & 0 \\ 1 & 0 & 0 & 0 & 0 & 0 & 0 & 0 & 0 & 0 \\ \tilde{p}_{3,2} & \tilde{p}_{2,1} & 0 & 0 & 0 & 0 & 0 & 0 & 0 & 0 \\ \tilde{p}_{4,3} & \tilde{p}_{4,2} & \tilde{p}_{4,1} & 0 & 0 & 0 & 0 & 0 & 0 & 0 \\ 0 & \tilde{p}_{5,3} & \tilde{p}_{5,2} & \tilde{p}_{5,1} & 0 & 0 & 0 & 0 & 0 & 0 \\ 0 & 0 & \tilde{p}_{6,3} & \tilde{p}_{6,2} & \tilde{p}_{6,1} & 0 & 0 & 0 & 0 & 0 \\ 0 & 0 & 0 & \tilde{p}_{7,3} & \tilde{p}_{7,2} & \tilde{p}_{7,1} & 0 & 0 & 0 & 0 \\ 0 & 0 & 0 & 0 & \tilde{p}_{8,3} & \tilde{p}_{8,2} & \tilde{p}_{8,1} & 0 & 0 & 0 \\ 0 & 0 & 0 & 0 & 0 & \tilde{p}_{9,3} & \tilde{p}_{9,2} & \tilde{p}_{9,1} & 0 & 0 \\ 0 & 0 & 0 & 0 & 0 & 0 & \tilde{p}_{10,3} & \tilde{p}_{10,2} & \tilde{p}_{10,1} & 0 \end{pmatrix}$$

$$\hat{h}^{(1)} = \begin{pmatrix} \tilde{s}_1 & \tilde{s}_2 & \tilde{s}_3 & \tilde{s}_4 & \tilde{s}_5 & \tilde{s}_6 & \tilde{s}_7 & \tilde{s}_8 & \tilde{s}_9 & \tilde{s}_{10} \\ 1 & 1 & \tilde{p}_{3,2} & \tilde{p}_{4,3} & 0 & 0 & 0 & 0 & 0 & 0 \\ 0 & 0 & \tilde{p}_{3,1} & \tilde{p}_{4,2} & \tilde{p}_{5,3} & 0 & 0 & 0 & 0 & 0 \\ 0 & 0 & 0 & \tilde{p}_{4,1} & \tilde{p}_{5,2} & \tilde{p}_{6,3} & 0 & 0 & 0 & 0 \\ 0 & 0 & 0 & 0 & \tilde{p}_{5,1} & \tilde{p}_{6,2} & \tilde{p}_{7,3} & 0 & 0 & 0 \\ 0 & 0 & 0 & 0 & 0 & \tilde{p}_{6,1} & \tilde{p}_{7,2} & \tilde{p}_{8,3} & 0 & 0 \\ 0 & 0 & 0 & 0 & 0 & 0 & \tilde{p}_{7,1} & \tilde{p}_{8,2} & \tilde{p}_{9,3} & 0 \\ 0 & 0 & 0 & 0 & 0 & 0 & 0 & \tilde{p}_{8,1} & \tilde{p}_{9,2} & \tilde{p}_{10,3} \\ 0 & 0 & 0 & 0 & 0 & 0 & 0 & 0 & \tilde{p}_{9,1} & \tilde{p}_{10,2} \\ 0 & 0 & 0 & 0 & 0 & 0 & 0 & 0 & 0 & \tilde{p}_{10,1} \\ 0 & 0 & 0 & 0 & 0 & 0 & 0 & 0 & 0 & 0 \end{pmatrix} \tag{12}$$

The operation of the first layer is now explicit: for each token $h_i^{(0)}$, it extracts the normalized transition probabilities $\tilde{p}_{i,k}$ for each possible lag and stores them in the element $\hat{h}_{i, S+T-k}^{(1)}$. The resulting vector is subsequently concatenated to the residual stream to be fed to the second layer $h_i^{(1)} = [[e_{s_i}, e_i], \hat{h}_i^{(1)}]$.

**Second layer, aggregation of transition probabilities:** To predict the next token, the model needs to determine which lag is more likely to have generated the sequence based on the observed transitions up to the current token. In order to do so, we need to aggregate the normalized transition probabilities of the entire past and copy them into the embedding of the current token. However, since consecutive tokens store transition probabilities in overlapping positions, simply averaging a token's history via a uniform attention would cause information from different transitions to mix. For example, when aggregating the past for the token at $i = 10$ in Eq. (11), summing $\hat{h}_9^{(1)}$ and $\hat{h}_{10}^{(2)}$ would mix $\tilde{p}_{10,2}$ and $\tilde{p}_{9,1}$ together. To avoid this, the second attention needs to use the positional encoding to learn a convex combination of the past tokens, which minimizes the overlap and maximizes the number of transitions stored. For example, by selecting one token every 3 steps $\hat{h}_4^{(1)}, \hat{h}_7^{(1)}, \hat{h}_{10}^{(1)}$ we can copy all the transitions without mixing information. More generally, the attention $\mathcal{A}^{(2)}$ should attend to every $K$-th token starting from the current one. This corresponds to a diagonal structure, with non-zero entries along the diagonals at positions $nK$ for $n \in \mathbb{N}$ and $nK < T$. This structure can be enforced by constructing the attention matrix $\tilde{A}^{(2,h)}$ with a single non-zero block that operates on the positional embeddings and takes the following form:

$$\tilde{A}^{(2,h)} = \begin{pmatrix} 0 & 0 & 0 \\ 0 & A^{(2,1)} & 0 \\ 0 & 0 & 0 \end{pmatrix} \quad A^{(2,1)} = \begin{pmatrix} -\lambda & -\lambda & -\lambda & -\lambda & -\lambda & -\lambda & -\lambda & -\lambda & -\lambda & -\lambda \\ -\lambda & -\lambda & -\lambda & -\lambda & -\lambda & -\lambda & -\lambda & -\lambda & -\lambda & -\lambda \\ -\lambda & -\lambda & -\lambda & -\lambda & -\lambda & -\lambda & -\lambda & -\lambda & -\lambda & -\lambda \\ -\lambda & -\lambda & -\lambda & +\lambda & -\lambda & -\lambda & -\lambda & -\lambda & -\lambda & -\lambda \\ -\lambda & -\lambda & -\lambda & -\lambda & +\lambda & -\lambda & -\lambda & -\lambda & -\lambda & -\lambda \\ -\lambda & -\lambda & -\lambda & -\lambda & -\lambda & +\lambda & -\lambda & -\lambda & -\lambda & -\lambda \\ -\lambda & -\lambda & -\lambda & +\lambda & -\lambda & -\lambda & +\lambda & -\lambda & -\lambda & -\lambda \\ -\lambda & -\lambda & -\lambda & -\lambda & +\lambda & -\lambda & -\lambda & +\lambda & -\lambda & -\lambda \\ -\lambda & -\lambda & -\lambda & -\lambda & -\lambda & +\lambda & -\lambda & -\lambda & +\lambda & -\lambda \\ -\lambda & -\lambda & -\lambda & +\lambda & -\lambda & -\lambda & +\lambda & -\lambda & -\lambda & +\lambda \end{pmatrix} \quad \mathcal{A}^{(2,1)} = \begin{pmatrix} 0 & 0 & 0 & 0 & 0 & 0 & 0 & 0 & 0 & 0 \\ 0 & 0 & 0 & 0 & 0 & 0 & 0 & 0 & 0 & 0 \\ 0 & 0 & 0 & 0 & 0 & 0 & 0 & 0 & 0 & 0 \\ 0 & 0 & 0 & 1 & 0 & 0 & 0 & 0 & 0 & 0 \\ 0 & 0 & 0 & 0 & 1 & 0 & 0 & 0 & 0 & 0 \\ 0 & 0 & 0 & 0 & 0 & 1 & 0 & 0 & 0 & 0 \\ 0 & 0 & 0 & \tfrac{1}{2} & 0 & 0 & \tfrac{1}{2} & 0 & 0 & 0 \\ 0 & 0 & 0 & 0 & \tfrac{1}{2} & 0 & 0 & \tfrac{1}{2} & 0 & 0 \\ 0 & 0 & 0 & 0 & 0 & \tfrac{1}{2} & 0 & 0 & \tfrac{1}{2} & 0 \\ 0 & 0 & 0 & \tfrac{1}{3} & 0 & 0 & \tfrac{1}{3} & 0 & 0 & \tfrac{1}{3} \end{pmatrix}$$

where the first $\hat{k}$ rows and columns are empty because the first $\hat{k}$ elements of the sequence are sampled independently from the stationary distribution and therefore no transitions are present.

The current construction allows us to overcome the issue caused by overlapping transitions; however, only copying a subset of tokens implies losing the information contained in the remaining $\hat{h}_i$ which were excluded by the attention. To overcome this limitation, we can introduce additional attention heads, for a total of $K$. Each one of them still presents a diagonal structure in order not to sum overlapping transitions but now shifted such that each head copies different tokens. For the given example, we can construct $A^{(2,2)}$ as in Eq. (13) to attend $\hat{h}_9^{(1)}$ and $\hat{h}_6^{(1)}$ which were discarded before, and similarly, the third head can be constructed with $A^{(2,3)}$ to sum $\hat{h}_8^{(1)}$ and $\hat{h}_5^{(1)}$. This mechanism corresponds to having each additional head with a diagonal structure and non-zero entries along the diagonals at position $nK + h - 1$ for $n \in \mathbb{N}$ and $h \in \{1, \ldots, H_2\}$ being the index of the head.

$$A^{(2,2)} = \begin{pmatrix} -\lambda & -\lambda & -\lambda & -\lambda & -\lambda & -\lambda & -\lambda & -\lambda & -\lambda & -\lambda \\ -\lambda & -\lambda & -\lambda & -\lambda & -\lambda & -\lambda & -\lambda & -\lambda & -\lambda & -\lambda \\ -\lambda & -\lambda & -\lambda & -\lambda & -\lambda & -\lambda & -\lambda & -\lambda & -\lambda & -\lambda \\ -\lambda & -\lambda & -\lambda & -\lambda & -\lambda & -\lambda & -\lambda & -\lambda & -\lambda & -\lambda \\ -\lambda & -\lambda & -\lambda & +\lambda & -\lambda & -\lambda & -\lambda & -\lambda & -\lambda & -\lambda \\ -\lambda & -\lambda & -\lambda & -\lambda & +\lambda & -\lambda & -\lambda & -\lambda & -\lambda & -\lambda \\ -\lambda & -\lambda & -\lambda & -\lambda & -\lambda & +\lambda & -\lambda & -\lambda & -\lambda & -\lambda \\ -\lambda & -\lambda & -\lambda & +\lambda & -\lambda & -\lambda & +\lambda & -\lambda & -\lambda & -\lambda \\ -\lambda & -\lambda & -\lambda & -\lambda & +\lambda & -\lambda & -\lambda & +\lambda & -\lambda & -\lambda \\ -\lambda & -\lambda & -\lambda & -\lambda & -\lambda & +\lambda & -\lambda & -\lambda & +\lambda & -\lambda \end{pmatrix} \qquad A^{(2,3)} = \begin{pmatrix} -\lambda & -\lambda & -\lambda & -\lambda & -\lambda & -\lambda & -\lambda & -\lambda & -\lambda & -\lambda \\ -\lambda & -\lambda & -\lambda & -\lambda & -\lambda & -\lambda & -\lambda & -\lambda & -\lambda & -\lambda \\ -\lambda & -\lambda & -\lambda & -\lambda & -\lambda & -\lambda & -\lambda & -\lambda & -\lambda & -\lambda \\ -\lambda & -\lambda & -\lambda & -\lambda & -\lambda & -\lambda & -\lambda & -\lambda & -\lambda & -\lambda \\ -\lambda & -\lambda & -\lambda & -\lambda & -\lambda & -\lambda & -\lambda & -\lambda & -\lambda & -\lambda \\ -\lambda & -\lambda & -\lambda & +\lambda & -\lambda & -\lambda & -\lambda & -\lambda & -\lambda & -\lambda \\ -\lambda & -\lambda & -\lambda & -\lambda & +\lambda & -\lambda & -\lambda & -\lambda & -\lambda & -\lambda \\ -\lambda & -\lambda & -\lambda & -\lambda & -\lambda & +\lambda & -\lambda & -\lambda & -\lambda & -\lambda \\ -\lambda & -\lambda & -\lambda & +\lambda & -\lambda & -\lambda & +\lambda & -\lambda & -\lambda & -\lambda \\ -\lambda & -\lambda & -\lambda & -\lambda & +\lambda & -\lambda & -\lambda & +\lambda & -\lambda & -\lambda \end{pmatrix}$$

(13)

The attention matrices $\{A^{(2,1)}, \ldots, A^{(2,H_2)}\}$ can be formalized as:

$$A_{ij}^{(2,h)} = \lambda \begin{cases} 1, & \text{if } i \geq j > \hat{k} \text{ and } (i - j) \mod K = h - 1 \\ -1, & \text{otherwise,} \end{cases}$$

where the condition $i \geq j \geq \hat{k}$ ensures that all entries in the first $\hat{k}$ rows and columns are set to $-\lambda$ and imposes a lower triangular structure due to causal masking. The modulo operation instead assigns each diagonal multiple of $K$ to $+\lambda$ (allowing attention) and the remaining diagonals to $-\lambda$ (masking attention), while $h$ determines the shift of the the first positive diagonal to ensure the heads do not overlap. The output of each head in the second layer is given by computing:

$$[[e_{s_i}, e_i], \hat{h}_i^{(1)}]^\top \tilde{A}^{(2,h)} [[e_{s_j}, e_j], \hat{h}_j^{(1)}] = A_{i,j}^{(2,h)} = \lambda \text{sign}(A_{i,j}^{(2,h)}),$$

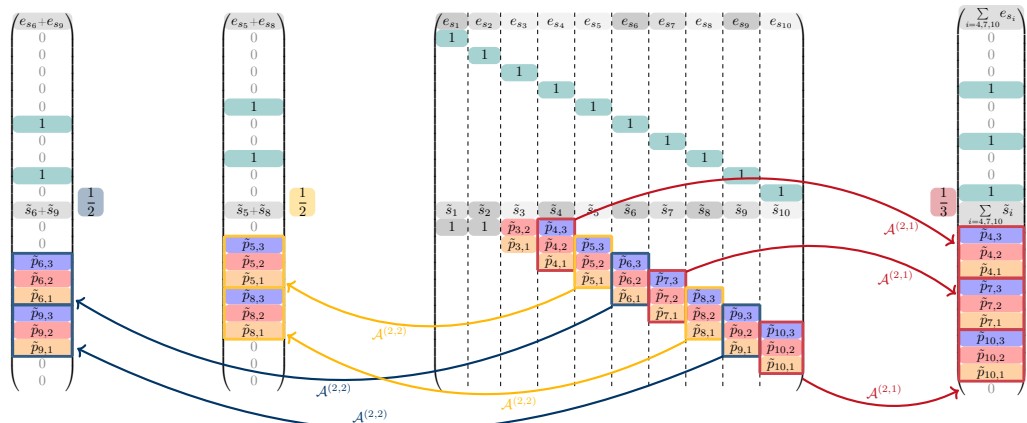

Figure 19: **Visualization of the mechanism of the second attention layer in our construction.** The matrix represents the input of the second layer $h^{(1)}$ whereas the single vectors the output for the 10th token $\hat{h}_{10}^{(2,1)}, \hat{h}_{10}^{(2,2)}, \hat{h}_{10}^{(2,3)}$. Each of the three attention heads (arrows of different colors) copies non-overlapping transition probabilities at distance 3 from each other from the past. By doing this, the output of the second layer for the current token (10) contains all $\tilde{p}$ for each lag without loss of information.

and applying softmax and in the limit as $\lambda \to \infty$, the rows of the attention become uniform for positive entries and zero otherwise:

$$\mathcal{A}_{ij}^{(2,h)} = \left(\text{Softmax}(A^{(2,h)})\right)_{ij} = \frac{\mathbb{1}\left[(i - j - h + 1) \mod K = 0\right]}{\sum_{m=1}^{i} \mathbb{1}\left[(m - j - h + 1) \mod K = 0\right]}$$

which in turns implies that the attention computes $\hat{h}_i^{(2)} = \text{Attn}(h_{1:T}^{(1)}; \tilde{A}^{(2,h)})_i = \sum_{j=\hat{k}}^{i} \mathcal{A}_{ij}^{(2,h)} h_j^{(1)}$. The output of each head is then concatenated into the residual stream. The visualization in Fig. 19 is helpful in fully understanding the mechanism implemented in the second layer. Due to space constraints, we only show the output for the 10th token for each attention head $\hat{h}_{10}^{(2,1)}, \hat{h}_{10}^{(2,2)}$ and $\hat{h}_{10}^{(2,3)}$. We can see how each attention head, represented by the arrows of different colors, aggregates the transition probabilities by attending to the tokens in the past that do not overlap and averaging them with uniform weights. It is important to remember that the input of the second attention layer is given by the concatenation of the input and output of the first layer, $h^{(1)} = [e_{s_i}, e_i, \hat{h}_i^{(1)}]$; thus when the tokens are averaged by the attention, the output will also contains an average of the position $e_i$ and semantics $e_{s_i}$ of the attended tokens which are represented by the first half of the vectors $\hat{h}_{10}^{(2,h)}$. Interestingly the block summing the positional encoding is basically storing a copy of the attention row $\mathcal{A}_i^{(2)}$ in the token $i$. Finally, when concatenating the output of the difference heads, we can see how the 10th-token will store the transition probabilities for each lag and of its entire past.

**Third layer, average of transition probabilities and lag selection.** Once the heads in the second layer aggregate the normalized transition probabilities from previous steps and store them within each token's embedding, the third layer sums these probabilities and uses the result to determine the correct lag that generated the sequence.

To build some intuition, let's revisit our running example. Suppose the current token is at position $i = 10$, and we are predicting the 11th token in the sequence. Given a set of possible lags $\{1, 2, 3\}$, the third attention mechanism must focus on one of the tokens at positions 8, 9, or 10. This ensures that transitions for all possible lags are considered: If the sequence was generated from the source of lag 1, the token at position 10 needs to be copied to predict the transition probabilities for the 11th token. For lag 2, the token at position 9 is copied, and so on. To determine the correct lag based on the sequence's history, our construction relies on the sum of past transitions up to the current token, $\sum_{j<i} \tilde{p}_{j,k}$. Therefore, to select the correct lag, the third attention is constructed so that the entries corresponding to the transitions of possible lags are proportional to the respective

cumulative sums. For example, to select which token among the ones in position $8, 9, 10$ should be copied to predict $11$, the third attention must be such that $\mathcal{A}_{10,10}^{(3)}$ is proportional exclusively to the sum of transitions of lag 1, i.e., $\sum_{j \leq 10} \tilde{p}_{j,1}$ while $\mathcal{A}_{10,9}^{(3)}$ is exclusively proportional to $\sum_{j \leq 10} \tilde{p}_{j,2}$ and $\mathcal{A}_{10,8}^{(3)} \propto \sum_{j \leq 10} \tilde{p}_{j,3}$. Then, in the limit of the softmax converging to the hardmax, the attention collapses to the entry corresponding to the larger sum and select the correspondent lag by copying the associated token. More generally, for this to apply to all rows, the third attention matrix must be constructed such that $\mathcal{A}_{i,i}^{(3)} \propto \sum_{j \leq i} \tilde{p}_{j,1}$, while $\mathcal{A}_{i,i-1}^{(3)} \propto \sum_{j \leq i} \tilde{p}_{j,2}$, and $\mathcal{A}_{i,i-2}^{(3)}$ to $\sum_{j \leq i} \tilde{p}_{j,3}$, with all remaining entries set to zero.

This selection mechanism is implemented through the combination of multiple blocks within the third attention matrix, $\tilde{A}^{(3)}$, which acts on the concatenated tokens $h_i^{(2)} = [h_i^{(0)}, \hat{h}_i^{(1)}, \hat{h}_i^{(2,1)}, \ldots, \hat{h}_i^{(2,H_2)}]$ and is structured as follows:

with the general formulation of the two matrices $A^{(3)}$ and $B^{(3)}$ given by:

$$A_{ij}^{(3)} = \lambda \begin{cases} +1 & \text{if } i - j + 1 \in \mathcal{K} \\ -1 & \text{if } i - j + 1 \notin \mathcal{K} \end{cases}, \qquad B_{ij}^{(3)} = \beta \begin{cases} +1, & \text{if } (i - j) \mod K = K - 1 \\ 0, & \text{otherwise}. \end{cases}$$

The matrix $A^{(3)}$ acts on the positional embedding of the input, similarly to the matrix $A^{(1)}$ in the first layer. The difference is that the position of the diagonals is now shifted by one. This shift ensures that the only non-zero entries after softmax are the ones on the diagonals corresponding to $k - 1$ for $k \in \mathcal{K}$. The matrix $B^{(3)}$ is instead responsible for the sum of the normalized transitions. Each block operates on the output of a corresponding head in the second layer. To understand how, consider the following tokens in output of the first head in the second layer,

where we define $\hat{p}_i^{(2,h)} \in \mathbb{R}^T$ as the block of $\hat{h}_i^{(2,h)}$ which contains the normalized transition probabilities and $\hat{m}_i^{(2,h)} \in \mathbb{R}^T$ the block that contains the copy of the second attention. By the structure of $\tilde{A}^{(3)}$ and the concatenation of the outputs $h^{(2)} = [h^{(0)}, \hat{h}_i^{(1)}, \hat{h}_i^{(2,1)}, \ldots, \hat{h}_i^{(2,H_2)}]$ we can see how $B^{(3)}$ act on these two blocks:

$$h^{(2)\top} \tilde{A}^{(3)} h^{(2)} = \sum_{h=1}^{K} p_i^{(2,h)\top} B^{(3)} m_j^{(2,h)} + e_i A^{(3)} e_j$$

where each operation involving $B^{(3)}$ is selectively summing the transition probabilities from the correspondent head. As previously mentioned, for the sum to be selective, it must hold that $h_i^{(2)\top} \tilde{A}^{(3)} h_{i-k+1}^{(2)} \propto \sum_{j \leq i} \tilde{p}_{j,k}$, where we replace $j$ with $i - k + 1$, because, after applying softmax, these will be the only non-zero entries due to $A^{(3)}$. The key idea behind this operation is that $B^{(3,h)} m_{i-k+1}^{(2,h)}$ is a boolean vector, such that when multiplied by $\hat{p}_i^{(2,h)}$, it sums only the entries that correspond to transitions of lag $k$. For instance, consider the product $\hat{p}_{10}^{(2,1)} B^{(3,h)} \hat{m}_8^{(2,1)}$ in Eq. (14), it should sum the transitions of lag 3 stored in $\hat{h}^{(2,1)}$ to give $\mathcal{A}_{10,8}$ such that the $8^{\text{th}}$ token can be copied to predict the $11^{\text{th}}$ if the lag of the sequence is 3. However, we can notice how simply taking the inner product $\hat{p}_{10}^{(2,1)\top} \hat{m}_8^{(2,h)}$ would lead to the wrong computations summing over the transitions

of lag 2 instead of 3 and excluding the transition correspondent to $\tilde{p}_{4,2}$. This happens because all the transitions are stored starting from the element $i-1$ of $\hat{p}_i^{(2,1)}$ and not $i$. To account for this, the matrix $B^{(3,1)}$ performs a permutation such that the mask is shifted by one position. Along with permuting the mask, the matrix $B^{(3,1)}$ also removes the normalization factor ($1/2$) and includes the missing transitions in the mask. To achieve this, each column of $B^{(3,h)}$ follows a pattern in which the entries are spaced at intervals of $K$, and the pattern shifts by one position between successive columns such that all possible sequences are present allowing to sum over all possible lags. This shift creates a cyclic arrangement where the columns repeat every $K$ as the transitions within the vector $\hat{p}_i^{(2,h)}$.

$$\hat{p}_{10}^{(2,1)\top} B^{(3,1)} \hat{m}_8^{(2,1)} = \frac{\beta}{3} \begin{pmatrix} \tilde{p}_{4,3} \\ \tilde{p}_{4,2} \\ \tilde{p}_{4,1} \\ \tilde{p}_{7,3} \\ \tilde{p}_{7,2} \\ \tilde{p}_{7,1} \\ \tilde{p}_{10,3} \\ \tilde{p}_{10,2} \\ \tilde{p}_{10,1} \\ 0 \end{pmatrix}^{\top} \begin{pmatrix} 0&1&0&0&1&0&0&1&0&0 \\ 0&0&1&0&0&1&0&0&1&0 \\ 1&0&0&1&0&0&1&0&0&1 \\ 0&1&0&0&1&0&0&1&0&0 \\ 0&0&1&0&0&1&0&0&1&0 \\ 1&0&0&1&0&0&1&0&0&1 \\ 0&1&0&0&1&0&0&1&0&0 \\ 0&0&1&0&0&1&0&0&1&0 \\ 1&0&0&1&0&0&1&0&0&1 \\ 0&1&0&0&1&0&0&1&0&0 \end{pmatrix} \begin{pmatrix} 0 \\ 0 \\ 0 \\ 0 \\ 1/2 \\ 0 \\ 0 \\ 0 \\ 1/2 \\ 0 \end{pmatrix} = \frac{\beta}{3} \begin{pmatrix} \tilde{p}_{4,3} \\ \tilde{p}_{4,2} \\ \tilde{p}_{4,1} \\ \tilde{p}_{7,3} \\ \tilde{p}_{7,2} \\ \tilde{p}_{7,1} \\ \tilde{p}_{10,3} \\ \tilde{p}_{10,2} \\ \tilde{p}_{10,1} \\ 0 \end{pmatrix}^{\top} \begin{pmatrix} 1 \\ 0 \\ 0 \\ 1 \\ 0 \\ 0 \\ 1 \\ 0 \\ 0 \\ 0 \end{pmatrix} \tag{14}$$

$$= \frac{\beta}{3} \left( \tilde{p}_{10,3} + \tilde{p}_{7,3} + \tilde{p}_{4,3} \right)$$

More in general, the matrix $B^{(3)}$ takes the following form:

$$B_{ij}^{(3)} = \beta \begin{cases} +1, & \text{if } ((i-j-K+1) \mod K = 0) \\ -1, & \text{otherwise} \end{cases} \tag{15}$$

The additional blocks containing $B^{(3)}$ act on the outputs of the other heads, performing the same operation by selectively summing the transitions of the same lag stored in the respective outputs. Considering all heads and only the non-zero entries after softmax, occurring at $j = i - k + 1$ due to $A^{(3)}$, we get

$$h_i^{(2)\top} \tilde{A}^{(3)} h_{i-k+1}^{(2)} = \sum_{h=1}^{K} p_i^{(2,h)\top} B^{(3)} m_{i-k+1}^{(2,h)} + \lambda = \sum_{h=1}^{K} \frac{\beta}{T_{h,i}+1} \sum_{n=0}^{T_{h,i}} \tilde{p}_{\hat{k}+h+nK,k} + \lambda,$$

where $T_{h,i} = \lfloor \frac{i-\hat{k}-h}{K} \rfloor$. Applying the softmax taking the limit $\lambda \to \infty$ results in non-zero entries of the attention only where $A_{i,j}^{(3)} = +\lambda$ similarly to the first layer. Moreover, for large $i$ for which $T_{h,i}+1 \approx \frac{i-\hat{k}}{K}$ and absorbing $K$ inside the temperature $\beta$, considering the last token $T$ we recover the weights in Eq. (3):

$$\mathcal{A}^{(3)}(h_{1:T}^{(2)}; \tilde{A}^{(3)})_{T,j} = \begin{cases} \dfrac{\exp\left( \frac{\beta}{(T-\hat{k})} \sum\limits_{n=\hat{k}+1}^{T} \tilde{p}_{n,n-j+1} \right)}{\sum\limits_{r \in \mathcal{K}} \exp\left( \frac{\beta}{(T-\hat{k})} \sum\limits_{n=\hat{k}+1}^{T} \tilde{p}_{n,r} \right)} & \text{if } T-j+1 = k \quad \text{for } k \in \mathcal{K}, \\ 0 & \text{elsewhere} \end{cases} \tag{16}$$

## G.1 Selective Induction Head and Next Token Prediction

**Selective induction head.** Eq. (16) shows how the last attention layer computes a weighted average of the tokens at a distance $k$ from the next one, with weights proportional to the average of the normalized transition probabilities of lag $k$. In practice, trained models often learn large values of $\beta$. Thus, we consider the limit $\beta \to \infty$, where the softmax converges to the hardmax:

$$\mathcal{A}_{i,j}^{(3)} = \mathbb{1}\left[ i-j+1 = k^{\star} \right] \quad \text{with} \quad k^{\star} = \operatorname{argmax}_k \sum_{n=\hat{k}+1}^{i} \tilde{p}_{n,k}.$$

Here, the transformer selects the causal structure (i.e., the lag) corresponding to the largest $\sum_{n=\hat{k}+1}^{i} \tilde{p}_{j,k}$. Given the current token $i$, the third layer then copies the token from the position

$i - k^\star + 1$, i.e., $\hat{h}_i^{(3)} = \sum_{j=\hat{k}}^i \mathbb{1}\left[i - j + 1 = k^\star\right] h_j^{(2)} = h_{i-k^\star+1}^{(2)}$. After concatenation to the residual stream, the tokens are of the following form:

$$h_i^{(3)} = \left[e_{s_i}, e_i, \hat{h}_i^{(1)}, \hat{h}_i^{(2,1)}, \ldots, \hat{h}_i^{(2,H_2)}, e_{s_{i-k^\star+1}}, e_{i-k^\star+1}, \hat{h}_{i-k^\star+1}^{(1)}, \hat{h}_{i-k^\star+1}^{(2,1)}, \ldots, \hat{h}_{i-k^\star+1}^{(2,H_2)}\right].$$

**Output layer: next token prediction.** Finally, the output layer $\widetilde{W}_O \in \mathbb{R}^{S \times \sum_l d_l}$ contains all zero blocks, except for the one acting on the semantics of the token copied by the third attention. This block learns the transition matrix $P^\star$ to predict the transition probabilities to the next token via

$$\widetilde{W}_O = \left(0_{S\times S} \ \vdots\ 0_{S\times T} \ \vdots\ 0_{S\times d_0} \ \vdots\ 0_{S\times 2d_0} \ \vdots\ \ldots \ \vdots\ 0_{S\times 2d_0} \ \vdots\ P^{\star\top} \ \vdots\ 0_{S\times T} \ \vdots\ 0_{S\times d_0} \ \vdots\ 0_{S\times 2d_0} \ \vdots\ \ldots \ \vdots\ 0_{S\times 2d_0}\right),$$

i.e., $\widetilde{W}_O h^{(3)} = P^{\star\top} e_{s_{i-k^\star+1}} = P^\star_{s_{i-k^\star+1}}$. This final layer shows how transformers can learn a selective induction head. This mechanism adapts to the input sequence by copying the token corresponding to the argmax of a specific quantity extracted by the previous layers and stored in the embeddings.

# H ALTERNATIVE THIRD LAYER CONSTRUCTION USING POSITIONAL EMBEDDING

In this section, we illustrate an alternative but equivalent construction that implements the same predictor as in Proposition 1. The first and second layers remain identical, the only difference is in the third layer which implements the selective sum of the normalized transition probabilities. This selection mechanism is implemented through the combination of multiple blocks within the third attention matrix, $\tilde{A}^{(3)}$, which, in this alternative construction is structured as follows:

$$\tilde{A}^{(3)} = \begin{pmatrix}
\begin{smallmatrix} 0 & 0 \\ 0 & A^{(3)} \end{smallmatrix} & 0 & 0 & 0 & \ldots & 0 & 0 \\
0 & 0 & 0 & 0 & \ldots & 0 & 0 \\
0 & 0 & 0 & 0 & \ldots & 0 & 0 \\
\begin{smallmatrix} 0 & 0 \\ 0 & B^{(3,1)} \end{smallmatrix} & 0 & 0 & 0 & \ldots & 0 & 0 \\
\vdots & \vdots & \vdots & \vdots & \ddots & \vdots & \vdots \\
0 & 0 & 0 & 0 & \ldots & 0 & 0 \\
\begin{smallmatrix} 0 & 0 \\ 0 & B^{(3,H_2)} \end{smallmatrix} & 0 & 0 & 0 & \ldots & 0 & 0
\end{pmatrix} \tag{17}$$

We can notice how, compared to the construction in Section 4, the blocks $B^{(3,1)}, \ldots, B^{(3,H_2)}$ are now positioned all in the first column. Moreover, they are not parameterized by the same matrix contrary to the other construction. The matrix $A^{(3)}$ acts on the positional embedding of the input similarly to the matrix $A^{(1)}$ in the first layer as in the previous construction:

$$A^{(3)}_{ij} = \lambda_1 \begin{cases} +1 & \text{if } j - i + 1 \in \mathcal{K} \\ -1 & \text{if } j - i + 1 \notin \mathcal{K} \end{cases} \qquad A^{(3)} = \begin{pmatrix}
+\lambda & -\lambda & -\lambda & -\lambda & -\lambda & -\lambda & -\lambda & -\lambda & -\lambda & -\lambda \\
+\lambda & +\lambda & -\lambda & -\lambda & -\lambda & -\lambda & -\lambda & -\lambda & -\lambda & -\lambda \\
+\lambda & +\lambda & +\lambda & -\lambda & -\lambda & -\lambda & -\lambda & -\lambda & -\lambda & -\lambda \\
-\lambda & +\lambda & +\lambda & +\lambda & -\lambda & -\lambda & -\lambda & -\lambda & -\lambda & -\lambda \\
-\lambda & -\lambda & +\lambda & +\lambda & +\lambda & -\lambda & -\lambda & -\lambda & -\lambda & -\lambda \\
-\lambda & -\lambda & -\lambda & +\lambda & +\lambda & +\lambda & -\lambda & -\lambda & -\lambda & -\lambda \\
-\lambda & -\lambda & -\lambda & -\lambda & +\lambda & +\lambda & +\lambda & -\lambda & -\lambda & -\lambda \\
-\lambda & -\lambda & -\lambda & -\lambda & -\lambda & +\lambda & +\lambda & +\lambda & -\lambda & -\lambda \\
-\lambda & -\lambda & -\lambda & -\lambda & -\lambda & -\lambda & +\lambda & +\lambda & +\lambda & -\lambda \\
-\lambda & -\lambda & -\lambda & -\lambda & -\lambda & -\lambda & -\lambda & +\lambda & +\lambda & +\lambda
\end{pmatrix}$$

This ensures that the only non-zero entries after softmax will be the ones on the diagonals corresponding to the lags seen during training. The matrices $B^{(3,1)}, \ldots, B^{(3,H_2)}$ are again responsible for the summation; each matrix operates on the output of a corresponding head in the second layer. To understand how this selective sum is implemented, let us consider the output of the first head in the second layer $h^{(2)} = [[e_{s_i}, e_i], \hat{h}_i^{(1)}, \hat{h}_i^{(2,1)}, \ldots, \hat{h}_i^{(2,H_2)}]$ in our example for the tokens 8,9 and 10:

$$\hat{h}_{10}^{(2,1)} = \tfrac{1}{3} \cdot \left( \underbrace{\sum_{i=4,7,10} e_{s_i}}\ 0\ 0\ 0\ 1\ 0\ 0\ 1\ 0\ 0\ 1\ \sum_{i=4,7,10}\tilde{s}_i\ \ \tilde{p}_{4,3}\ \tilde{p}_{4,2}\ \tilde{p}_{4,1}\ \tilde{p}_{7,3}\ \tilde{p}_{7,2}\ \tilde{p}_{7,1}\ \tilde{p}_{10,3}\ \tilde{p}_{10,2}\ \tilde{p}_{10,1}\ 0 \right)$$

$$\hat{h}_{9}^{(2,1)} = \tfrac{1}{2} \cdot \left( e_{s_6} + e_{s_9}\ \ 0\ 0\ 0\ 0\ 0\ 1\ 0\ 0\ 1\ 0\ \ \tilde{s}_6 + \tilde{s}_9\ \ 0\ \ 0\ \tilde{p}_{6,3}\ \tilde{p}_{6,2}\ \tilde{p}_{6,1}\ \tilde{p}_{9,3}\ \tilde{p}_{9,2}\ \tilde{p}_{9,1}\ 0\ \ 0 \right)$$

$$\hat{h}_{8}^{(2,1)} = \tfrac{1}{2} \cdot \left( e_{s_5} + e_{s_8}\ \ 0\ 0\ 0\ 0\ 1\ 0\ 0\ 1\ 0\ 0\ \ \tilde{s}_5 + \tilde{s}_8\ \ 0\ \tilde{p}_{5,3}\ \tilde{p}_{5,2}\ \tilde{p}_{5,1}\ \tilde{p}_{8,3}\ \tilde{p}_{8,2}\ \tilde{p}_{8,1}\ 0\ \ 0\ \ 0 \right)$$

$$\underbrace{\phantom{0\ 0\ 0\ 0\ 1\ 0\ 0\ 1\ 0\ 0}}_{\hat{m}_8^{(2,1)}} \qquad \underbrace{\phantom{\tilde{p}_{5,3}\ \tilde{p}_{5,2}\ \tilde{p}_{5,1}\ \tilde{p}_{8,3}\ \tilde{p}_{8,2}\ \tilde{p}_{8,1}}}_{\hat{p}_8^{(2,1)}}$$

and define $\hat{p}_i^{(2,h)} \in \mathbb{R}^T$ as the block of $\hat{h}_i^{(2,h)}$ which contains the normalized transition probabilities. By the structure in Eq. (17) we can see how, when computing the attention, the matrices $B^{(3,h)}$ act on these two blocks:

$$h_i^{(2)\top} \tilde{A}^{(3)} h_j^{(2)} = \sum_{h=1}^K p_i^{(2,h)\top} B^{(3,h)} e_j + e_i A^{(3)} e_j$$

here we notice how the difference compared to the construction in Section 4 lies in the fact that, due to the position we are not using the copy of the attention to construct the boolean vector but directly the one hot encoding of the position. Each operation involving $B^{(3,h)}$ is still selectively summing the transition probabilities from the correspondent head but with a slightly different mechanism. Let us consider the product $h_i^{(2)\top} \tilde{A}^{(3)} h_{i-k}^{(2)}$ which will be the only non zero entries after softmax, and show how it only sums the transitions of lag $k$. The main idea is that $B^{(3,h)}$ are boolean matrices such that each column sums only the entries containing the transitions for one of the lags. To achieve this, each column in the matrix follows a pattern in which the entries are spaced at intervals of $K$, and the pattern shifts by one position between successive columns. This shift creates a cyclic arrangement across the columns which repeat with frequency $K$. For each head $h$, the matrix $B^{(3,h)}$ is structured such that the product $\hat{p}_i^{(2,h)} B^{(3,h)}$ results in a vector where each element is the sum of the transitions for a given $k$.In particular, the first element of the vector corresponds to the sum of $k_{\min}$, the $K$-th element corresponds to the sum of $k_{\max}$, and this pattern repeats cyclically for subsequent elements. To give an example, consider the product $\hat{p}_{10}^{(2,1)} B^{(3,1)} e_8$ in Eq. (I.2), which sums the transitions stored in $\hat{h}^{(2,1)}$. Notice the structure of $B^{(3,1)}$; the first column aligns with the transitions of lag 1 in $\hat{p}_{10}^{(2,1)}$. Given that the index of $\hat{p}_i^{(2,1)}$ is 10 and the index of $e_j$ is 8, the sum constructs $\mathcal{A}_{(10,8)}^{(3)}$, which is used to copy the 8$^{\text{th}}$ token to predict the 11$^{\text{th}}$ if the lag of the sequence is 3. Hence, $\hat{p}_{10}^{(2,1)} B^{(3,1)} e_8$ has to sum the transitions of lag 3:

$$\hat{p}_{10}^{(2,1)\top} B^{(3,1)} e_8 = \frac{\beta}{3} \begin{pmatrix} \tilde{p}_{4,3} \\ \tilde{p}_{4,2} \\ \tilde{p}_{4,1} \\ \tilde{p}_{7,3} \\ \tilde{p}_{7,2} \\ \tilde{p}_{7,1} \\ \tilde{p}_{10,3} \\ \tilde{p}_{10,2} \\ \tilde{p}_{10,1} \\ 0 \end{pmatrix}^\top \begin{pmatrix} 0 & 1 & 0 & 0 & 1 & 0 & 0 & 1 & 0 & 0 \\ 0 & 0 & 1 & 0 & 0 & 1 & 0 & 0 & 1 & 0 \\ 1 & 0 & 0 & 1 & 0 & 0 & 1 & 0 & 0 & 1 \\ 0 & 1 & 0 & 0 & 1 & 0 & 0 & 1 & 0 & 0 \\ 0 & 0 & 1 & 0 & 0 & 1 & 0 & 0 & 1 & 0 \\ 1 & 0 & 0 & 1 & 0 & 0 & 1 & 0 & 0 & 1 \\ 0 & 1 & 0 & 0 & 1 & 0 & 0 & 1 & 0 & 0 \\ 0 & 0 & 1 & 0 & 0 & 1 & 0 & 0 & 1 & 0 \\ 1 & 0 & 0 & 1 & 0 & 0 & 1 & 0 & 0 & 1 \\ 0 & 1 & 0 & 0 & 1 & 0 & 0 & 1 & 0 & 0 \end{pmatrix} \begin{pmatrix} 0 \\ 0 \\ 0 \\ 0 \\ 0 \\ 0 \\ 0 \\ 1 \\ 0 \\ 0 \end{pmatrix} = \frac{\beta}{3} \begin{pmatrix} \tilde{p}_{4,1} + \tilde{p}_{7,1} + \tilde{p}_{10,1} \\ \tilde{p}_{4,3} + \tilde{p}_{7,3} + \tilde{p}_{10,3} \\ \tilde{p}_{4,2} + \tilde{p}_{7,2} + \tilde{p}_{10,2} \\ \tilde{p}_{4,1} + \tilde{p}_{7,1} + \tilde{p}_{10,1} \\ \tilde{p}_{4,3} + \tilde{p}_{7,3} + \tilde{p}_{10,3} \\ \tilde{p}_{4,2} + \tilde{p}_{7,2} + \tilde{p}_{10,2} \\ \tilde{p}_{4,1} + \tilde{p}_{7,1} + \tilde{p}_{10,1} \\ \tilde{p}_{4,3} + \tilde{p}_{7,3} + \tilde{p}_{10,3} \\ \tilde{p}_{4,2} + \tilde{p}_{7,2} + \tilde{p}_{10,2} \\ \tilde{p}_{4,1} + \tilde{p}_{7,1} + \tilde{p}_{10,1} \end{pmatrix}^\top \begin{pmatrix} 0 \\ 0 \\ 0 \\ 0 \\ 0 \\ 0 \\ 0 \\ 1 \\ 0 \\ 0 \end{pmatrix}$$

$$= \frac{\beta}{3} \left( \tilde{p}_{10,3} + \tilde{p}_{7,3} + \tilde{p}_{4,3} \right)$$

We can see how the operation implemented by this different parameterization is the same then in the other construction. Therefore the overall predictor remains unchanged. The additional matrices $B^{(3,h)}$, which act on the outputs of the other heads $\hat{h}^{(2,h)}$, perform the same operation by summing the transitions stored in the outputs of the respective heads. The difference in the construction of the matrix $B^{(3,h)}$ for $h \neq 1$ is that the columns are shifted by $h$ positions relative to $h = 1$. Specifically, for each $h$, the columns are shifted by $h$ positions compared to the matrix $B^{(3,1)}$. In more generality, the matrix $B^{(3,h)}$ is constructed as follows:

$$B_{ij}^{(3,h)} = \beta \begin{cases} +1, & \text{if } ((i - j - h + 1) \mod K = 0) \\ 0, & \text{otherwise} \end{cases}$$

where $h$ takes into account for the shift.

## I   CONSTRUCTION FOR ANY SET OF LAGS

The construction illustrated in Section 4 considers only contiguous lags, i.e. set of lags that are intervals of the positive integers. However, both our interleaved Markov chain framework and the Transformer construction can be extended to any set of lags. The implemented algorithm is the same, but the structure of the weights in the different layers becomes more complex because the mechanism with which the transition probabilities are aggregated depends on the relative distance

between the lags in the set. Due to the difficulties in finding a general formulation of the matrices involved for any set of lags as well as the optimal number of heads which depends now not only on the number of lags but on the relative distance between them, we limit this section into illustrate two example with $T = 10$ and $\mathcal{K} = \{1, 3\}$ and $T = 12$ $\mathcal{K} = \{1, 3, 4\}$ for which we will visualize the matrices and operations involved.

## I.1 EXAMPLE FOR $\mathcal{K} = \{1, 3\}$

**First layer:** The structure of the first layer remains unchanged from Section 4. The important difference is that now the diagonals in the matrix $A^{(1)}$ with positive entries are only 2nd and 4th:

$$\widetilde{A}^{(1)} = \begin{pmatrix} \log P^\top & 0 \\ 0 & A^{(1)} \end{pmatrix}$$

$$A^{(1)} = \begin{pmatrix} -\lambda & -\lambda & -\lambda & -\lambda & -\lambda & -\lambda & -\lambda & -\lambda & -\lambda & -\lambda & -\lambda & -\lambda \\ -\lambda & -\lambda & -\lambda & -\lambda & -\lambda & -\lambda & -\lambda & -\lambda & -\lambda & -\lambda & -\lambda & -\lambda \\ +\lambda & -\lambda & +\lambda & -\lambda & -\lambda & -\lambda & -\lambda & -\lambda & -\lambda & -\lambda & -\lambda & -\lambda \\ -\lambda & +\lambda & -\lambda & +\lambda & -\lambda & -\lambda & -\lambda & -\lambda & -\lambda & -\lambda & -\lambda & -\lambda \\ -\lambda & -\lambda & +\lambda & -\lambda & +\lambda & -\lambda & -\lambda & -\lambda & -\lambda & -\lambda & -\lambda & -\lambda \\ -\lambda & -\lambda & -\lambda & +\lambda & -\lambda & +\lambda & -\lambda & -\lambda & -\lambda & -\lambda & -\lambda & -\lambda \\ -\lambda & -\lambda & -\lambda & -\lambda & +\lambda & -\lambda & +\lambda & -\lambda & -\lambda & -\lambda & -\lambda & -\lambda \\ -\lambda & -\lambda & -\lambda & -\lambda & -\lambda & +\lambda & -\lambda & +\lambda & -\lambda & -\lambda & -\lambda & -\lambda \\ -\lambda & -\lambda & -\lambda & -\lambda & -\lambda & -\lambda & +\lambda & -\lambda & +\lambda & -\lambda & -\lambda & -\lambda \\ -\lambda & -\lambda & -\lambda & -\lambda & -\lambda & -\lambda & -\lambda & +\lambda & -\lambda & +\lambda & -\lambda \end{pmatrix}$$

$$A^{(1)}_{ij} = \begin{cases} +\lambda & \text{if} \quad j - i \in \mathcal{K} \\ -\lambda & \text{if} \quad j - i \notin \mathcal{K}. \end{cases}$$

The output token at index $i$ after the first layer still corresponds to a weighted average of the past tokens $h^{(0)}_{i-k}$ for $k \in \mathcal{K}$ where the weights are given by the normalized probabilities $\tilde{p}_{i,k}$:

$$\hat{h}^{(1)}_i = \text{Attn}(h^{(0)}_{1:T}; \tilde{A}^{(1)})_i = \begin{cases} \sum_{j=1}^{i} \mathbb{1}\left[i - j \in \mathcal{K}\right] \dfrac{P_{s_j, s_i}}{\sum_{r \in \mathcal{K}} P_{s_r, s_i}} h^{(0)}_j & \text{if } i > 1 \\ h^{(0)}_1 & \text{if } i = 1 \end{cases}$$

$$= \begin{cases} \sum_{k \in \mathcal{K}, k < i} \tilde{p}_{i,k} h^{(0)}_{i-k} & \text{if } i > 1 \\ h^{(0)}_1 & \text{if } i = 1 \end{cases}$$

Due to the lack of the entries on the 3rd diagonal, both the attention and the output token will change accordingly:

$$\mathcal{A}^{(1)} = \begin{pmatrix} 1 & 0 & 0 & 0 & 0 & 0 & 0 & 0 & 0 & 0 \\ 1/2 & 1/2 & 0 & 0 & 0 & 0 & 0 & 0 & 0 & 0 \\ 1/3 & 1/3 & 1/3 & 0 & 0 & 0 & 0 & 0 & 0 & 0 \\ \tilde{p}_{4,3} & 0 & \tilde{p}_{4,1} & 0 & 0 & 0 & 0 & 0 & 0 & 0 \\ 0 & \tilde{p}_{5,3} & 0 & \tilde{p}_{5,1} & 0 & 0 & 0 & 0 & 0 & 0 \\ 0 & 0 & \tilde{p}_{6,3} & 0 & \tilde{p}_{6,1} & 0 & 0 & 0 & 0 & 0 \\ 0 & 0 & 0 & \tilde{p}_{7,3} & 0 & \tilde{p}_{7,1} & 0 & 0 & 0 & 0 \\ 0 & 0 & 0 & 0 & \tilde{p}_{8,3} & 0 & \tilde{p}_{8,1} & 0 & 0 & 0 \\ 0 & 0 & 0 & 0 & 0 & \tilde{p}_{9,3} & 0 & \tilde{p}_{9,1} & 0 & 0 \\ 0 & 0 & 0 & 0 & 0 & 0 & \tilde{p}_{10,3} & 0 & \tilde{p}_{10,1} & 0 \end{pmatrix} \quad \hat{h}^{(1)} = \begin{pmatrix} \tilde{s}_1 & \tilde{s}_2 & \tilde{s}_3 & \tilde{s}_4 & \tilde{s}_5 & \tilde{s}_6 & \tilde{s}_7 & \tilde{s}_8 & \tilde{s}_9 & \tilde{s}_{10} \\ 1 & 1/2 & 1/3 & \tilde{p}_{4,3} & 0 & 0 & 0 & 0 & 0 & 0 \\ 0 & 1/2 & 1/3 & 0 & \tilde{p}_{5,3} & 0 & 0 & 0 & 0 & 0 \\ 0 & 0 & 1/3 & \tilde{p}_{4,1} & 0 & \tilde{p}_{6,3} & 0 & 0 & 0 & 0 \\ 0 & 0 & 0 & 0 & \tilde{p}_{5,1} & 0 & \tilde{p}_{7,3} & 0 & 0 & 0 \\ 0 & 0 & 0 & 0 & 0 & \tilde{p}_{6,1} & 0 & \tilde{p}_{8,3} & 0 & 0 \\ 0 & 0 & 0 & 0 & 0 & 0 & \tilde{p}_{7,1} & 0 & \tilde{p}_{9,3} & 0 \\ 0 & 0 & 0 & 0 & 0 & 0 & 0 & \tilde{p}_{8,1} & 0 & \tilde{p}_{10,3} \\ 0 & 0 & 0 & 0 & 0 & 0 & 0 & 0 & \tilde{p}_{9,1} & 0 \\ 0 & 0 & 0 & 0 & 0 & 0 & 0 & 0 & 0 & \tilde{p}_{10,1} \\ 0 & 0 & 0 & 0 & 0 & 0 & 0 & 0 & 0 & 0 \end{pmatrix}$$
(18)

**Second layer.** Similarly to the construction for contiguous lags, the second layer is responsible for aggregating the normalized transition probabilities such that they are stored in the embedding of the current vector for its entire history. The second attention needs to learn an effective way of doing a convex combination of the input tokens such that the overlap is minimized and all the transitions are stored without mixing them. Consider the token at $i = 10$ in Eq. (26), summing two consecutive tokens such as $\hat{h}^{(1)}_9$ and $\hat{h}^{(2)}_{10}$ ,contrary to the contiguous case in Eq. (12), does not lead to any mixing due to the absence of transitions of lag 2. Therefore, 2 attention heads are still sufficient to copy all the transitions in the past as long as they learn to attend two consecutive tokens each.

Therefore, the optimal way to combine past tokens strictly depends on the number of tokens and the relative distance between them. Hence, finding a general formula for the positions at which the second attention $\mathcal{A}^{(2)}$ should be attended to minimize overlap, is challenging and beyond the scope of this work. Similar considerations apply to the optimal number of heads required, which depends on the solution of the previous problem. However, the task for arbitrary sets of lags, can always be solved by consider the correspondent contiguous problem with $\hat{k} - \min(\mathcal{K} + 1)$ heads. However there are cases in which we can leverage the structure given by the distance between the lags to use fewer heads. One example is the one considered in this section with $\mathcal{K} = \{1, 3\}$ we only need two

heads to achieve optimal sample complexity. The form of the matrix $\tilde{A}^{(2,h)}$ remains unchanged:

$$
\tilde{A}^{(2,h)} = \left( \begin{array}{cc|c} 0 & 0 & 0 \\ 0 & A^{(2,1)} & 0 \\ \hline 0 & 0 \end{array} \right)
$$

Considering the case illustrated in Eq. (26), in order for the two heads to copy all the tokens without overlap, it is sufficient to sum two consecutive tokens and skip two. Therefore, the first attention has the pattern : $(0,0,1,1)$ while the second one $(1,1,0,0)$ as illustrated in the following:

$$
A^{(2,1)} = \begin{pmatrix} \text{-}\lambda & \text{-}\lambda & \text{-}\lambda & \text{-}\lambda & \text{-}\lambda & \text{-}\lambda & \text{-}\lambda & \text{-}\lambda & \text{-}\lambda & \text{-}\lambda \\ \text{-}\lambda & \text{-}\lambda & \text{-}\lambda & \text{-}\lambda & \text{-}\lambda & \text{-}\lambda & \text{-}\lambda & \text{-}\lambda & \text{-}\lambda & \text{-}\lambda \\ \text{-}\lambda & \text{-}\lambda & \text{-}\lambda & \text{-}\lambda & \text{-}\lambda & \text{-}\lambda & \text{-}\lambda & \text{-}\lambda & \text{-}\lambda & \text{-}\lambda \\ \text{-}\lambda & \text{-}\lambda & \text{-}\lambda & \text{+}\lambda & \text{-}\lambda & \text{-}\lambda & \text{-}\lambda & \text{-}\lambda & \text{-}\lambda & \text{-}\lambda \\ \text{-}\lambda & \text{-}\lambda & \text{-}\lambda & \text{+}\lambda & \text{+}\lambda & \text{-}\lambda & \text{-}\lambda & \text{-}\lambda & \text{-}\lambda & \text{-}\lambda \\ \text{-}\lambda & \text{-}\lambda & \text{-}\lambda & \text{-}\lambda & \text{+}\lambda & \text{+}\lambda & \text{-}\lambda & \text{-}\lambda & \text{-}\lambda & \text{-}\lambda \\ \text{-}\lambda & \text{-}\lambda & \text{-}\lambda & \text{-}\lambda & \text{-}\lambda & \text{+}\lambda & \text{+}\lambda & \text{-}\lambda & \text{-}\lambda & \text{-}\lambda \\ \text{-}\lambda & \text{-}\lambda & \text{-}\lambda & \text{+}\lambda & \text{-}\lambda & \text{-}\lambda & \text{+}\lambda & \text{+}\lambda & \text{-}\lambda & \text{-}\lambda \\ \text{-}\lambda & \text{-}\lambda & \text{-}\lambda & \text{+}\lambda & \text{+}\lambda & \text{-}\lambda & \text{-}\lambda & \text{+}\lambda & \text{+}\lambda & \text{-}\lambda \\ \text{-}\lambda & \text{-}\lambda & \text{-}\lambda & \text{+}\lambda & \text{+}\lambda & \text{-}\lambda & \text{-}\lambda & \text{+}\lambda & \text{+}\lambda \end{pmatrix} \quad \mathcal{A}^{(2,2)} = \begin{pmatrix} \text{-}\lambda & \text{-}\lambda & \text{-}\lambda & \text{-}\lambda & \text{-}\lambda & \text{-}\lambda & \text{-}\lambda & \text{-}\lambda & \text{-}\lambda & \text{-}\lambda \\ \text{-}\lambda & \text{-}\lambda & \text{-}\lambda & \text{-}\lambda & \text{-}\lambda & \text{-}\lambda & \text{-}\lambda & \text{-}\lambda & \text{-}\lambda & \text{-}\lambda \\ \text{-}\lambda & \text{-}\lambda & \text{-}\lambda & \text{-}\lambda & \text{-}\lambda & \text{-}\lambda & \text{-}\lambda & \text{-}\lambda & \text{-}\lambda & \text{-}\lambda \\ \text{-}\lambda & \text{-}\lambda & \text{-}\lambda & \text{-}\lambda & \text{-}\lambda & \text{-}\lambda & \text{-}\lambda & \text{-}\lambda & \text{-}\lambda & \text{-}\lambda \\ \text{-}\lambda & \text{-}\lambda & \text{-}\lambda & \text{-}\lambda & \text{-}\lambda & \text{-}\lambda & \text{-}\lambda & \text{-}\lambda & \text{-}\lambda & \text{-}\lambda \\ \text{-}\lambda & \text{-}\lambda & \text{-}\lambda & \text{+}\lambda & \text{-}\lambda & \text{-}\lambda & \text{-}\lambda & \text{-}\lambda & \text{-}\lambda & \text{-}\lambda \\ \text{-}\lambda & \text{-}\lambda & \text{-}\lambda & \text{+}\lambda & \text{+}\lambda & \text{-}\lambda & \text{-}\lambda & \text{-}\lambda & \text{-}\lambda & \text{-}\lambda \\ \text{-}\lambda & \text{-}\lambda & \text{-}\lambda & \text{-}\lambda & \text{+}\lambda & \text{+}\lambda & \text{-}\lambda & \text{-}\lambda & \text{-}\lambda & \text{-}\lambda \\ \text{-}\lambda & \text{-}\lambda & \text{-}\lambda & \text{-}\lambda & \text{-}\lambda & \text{+}\lambda & \text{+}\lambda & \text{-}\lambda & \text{-}\lambda & \text{-}\lambda \\ \text{-}\lambda & \text{-}\lambda & \text{-}\lambda & \text{+}\lambda & \text{-}\lambda & \text{-}\lambda & \text{+}\lambda & \text{+}\lambda & \text{-}\lambda & \text{-}\lambda \end{pmatrix} \tag{19}
$$

where the first $\hat{k}$ rows and columns are empty because the first $\hat{k}$ elements of the sequence are sampled independently from the stationary distribution and therefore no transitions are present.

The attention computes the same operation as before:

$$
\hat{h}_i^{(2)} = \mathrm{Attn}(h_{1:T}^{(1)}; \tilde{A}^{(2,h)})_i = \sum_{j=\hat{k}}^{i} \frac{\mathbb{1}\left[ A_{ij}^{(2,h)} = +\lambda \right]}{\sum_{m=1}^{i} \mathbb{1}\left[ A_{im}^{(2,h)} = +\lambda \right]} h_j^{(1)} .
$$

The output of each head is then concatenated into the residual stream. The structure of the third layer for the general case of any set of lags, also needs some modifications to take into account the particular structure that was enforced in the second layer. We extend the construction introduced in Section H using the positional embeddings. First of all, the matrix $A^{(3)}$ remains unchanged compared to the previous constructions, it has positive values along the diagonals correspondent to the lags shifted by one position to take into account the fact that we are predicting the next token in the sequence:

$$
A_{ij}^{(3)} = \begin{cases} +\lambda & \text{if} \quad j - i + 1 \in \mathcal{K} \\ -\lambda & \text{if} \quad j - i + 1 \notin \mathcal{K} \end{cases} \quad A^{(3)} = \begin{pmatrix} \text{-}\lambda & \text{-}\lambda & \text{-}\lambda & \text{-}\lambda & \text{-}\lambda & \text{-}\lambda & \text{-}\lambda & \text{-}\lambda & \text{-}\lambda & \text{-}\lambda \\ \text{-}\lambda & \text{-}\lambda & \text{-}\lambda & \text{-}\lambda & \text{-}\lambda & \text{-}\lambda & \text{-}\lambda & \text{-}\lambda & \text{-}\lambda & \text{-}\lambda \\ \text{+}\lambda & \text{-}\lambda & \text{+}\lambda & \text{-}\lambda & \text{-}\lambda & \text{-}\lambda & \text{-}\lambda & \text{-}\lambda & \text{-}\lambda & \text{-}\lambda \\ \text{-}\lambda & \text{+}\lambda & \text{-}\lambda & \text{+}\lambda & \text{-}\lambda & \text{-}\lambda & \text{-}\lambda & \text{-}\lambda & \text{-}\lambda & \text{-}\lambda \\ \text{-}\lambda & \text{-}\lambda & \text{+}\lambda & \text{-}\lambda & \text{+}\lambda & \text{-}\lambda & \text{-}\lambda & \text{-}\lambda & \text{-}\lambda & \text{-}\lambda \\ \text{-}\lambda & \text{-}\lambda & \text{-}\lambda & \text{+}\lambda & \text{-}\lambda & \text{+}\lambda & \text{-}\lambda & \text{-}\lambda & \text{-}\lambda & \text{-}\lambda \\ \text{-}\lambda & \text{-}\lambda & \text{-}\lambda & \text{-}\lambda & \text{+}\lambda & \text{-}\lambda & \text{+}\lambda & \text{-}\lambda & \text{-}\lambda & \text{-}\lambda \\ \text{-}\lambda & \text{-}\lambda & \text{-}\lambda & \text{-}\lambda & \text{-}\lambda & \text{+}\lambda & \text{-}\lambda & \text{+}\lambda & \text{-}\lambda & \text{-}\lambda \\ \text{-}\lambda & \text{-}\lambda & \text{-}\lambda & \text{-}\lambda & \text{-}\lambda & \text{-}\lambda & \text{+}\lambda & \text{-}\lambda & \text{+}\lambda & \text{-}\lambda \\ \text{-}\lambda & \text{-}\lambda & \text{-}\lambda & \text{-}\lambda & \text{-}\lambda & \text{-}\lambda & \text{-}\lambda & \text{+}\lambda & \text{-}\lambda & \text{+}\lambda \end{pmatrix} \tag{20}
$$

The matrix $B^{(3)}$ is responsible for the sum of the normalized transitions; each block operates on the output of a corresponding head in the second layer. To understand how, consider the following tokens in output of the first head in the second layer:

$$
\begin{aligned}
\hat{h}_{10}^{(2)} &= \tfrac{1}{4} \cdot \Big( \sum_{i=5,6,9,10} e_{s_i} \;\; 0\; 0\; 0\; 0\; 1\; 1\; 0\; 0\; 1\; 1 \;\; \sum_{i=5,6,9,10} \tilde{s}_i \;\; 0 \;\; \tilde{p}_{5,3}\; \tilde{p}_{6,3}\; \tilde{p}_{5,1}\; \tilde{p}_{6,1}\; \tilde{p}_{9,3}\; \tilde{p}_{10,3}\; \tilde{p}_{9,1}\; \tilde{p}_{10,1}\; 0 \Big) \\
\hat{h}_{9}^{(2)} &= \tfrac{1}{4} \cdot \Big( \sum_{i=4,5,8,9} e_{s_i} \;\; 0\; 0\; 0\; 1\; 1\; 0\; 0\; 1\; 1\; 0 \;\; \sum_{i=4,5,8,9} \tilde{s}_i \;\; \tilde{p}_{4,3}\; \tilde{p}_{5,3}\; \tilde{p}_{4,1}\; \tilde{p}_{5,1}\; \tilde{p}_{8,3}\; \tilde{p}_{9,3}\; \tilde{p}_{8,1}\; \tilde{p}_{9,1}\; 0\; 0 \Big) \\
\hat{h}_{8}^{(2)} &= \tfrac{1}{3} \cdot \Big( \sum_{i=4,7,8} e_{s_i} \;\; 0\; 0\; 0\; 1\; 0\; 0\; 1\; 1\; 0\; 0 \;\; \sum_{i=4,7,8} \tilde{s}_i \;\; \tilde{p}_{4,3}\; 0\; \tilde{p}_{4,1}\; \tilde{p}_{7,3}\; \tilde{p}_{8,3}\; \tilde{p}_{7,1}\; \tilde{p}_{8,1}\; 0\; 0\; 0 \Big)
\end{aligned}
$$

$$\underbrace{\phantom{\hspace{6cm}}}_{\hat{p}^{(2)}_8}$$

$$\tag{21}$$

By the structure of $\tilde{A}^{(3)}$ we can see how, when computing the attention, the matrices $B^{(3,h)}$ are applied on the positional encoding $e_j$ and the result is multiplied by $\hat{p}_i^{(2,h)}$:

$$
h_i^{(2)\top} \tilde{A}^{(3)} h_j^{(2)\top} = \sum_{h=1}^{K} p_i^{(2,h)\top} B^{(3,h)} e_j + e_i A^{(3)} e_j \tag{22}
$$

where $B^{(3,h)}$ is selectively summing the transition probabilities from the correspondent head. As for the simpler case of contiguous lags, for the sum to be selective it must hold that $h_i^{(2)\top} \tilde{A}^{(3)} h_{i-k+1}^{(2)} \propto \sum_{j \leq i} \tilde{p}_{j,k}$, where $i - k + 1$ are the only non-zero entries due to $A^{(3)}$ after applying softmax. As before, $B^{(3,h)}$ are boolean matrices such that each column sums only the entries containing the transitions for one of the lags. To achieve this, the matrix need to learn the same patter as in the attention of the second layer $\mathcal{A}^{(2)}$ which was used to sum the vectors and create the current inputs. Each column is shifted by one position and they cyclically repeat with frequency $K$. In the following example, we consider $\hat{p}_{10}^{(2,1)} B^{(3,1)} e_8$ in Eq. (23), which sums the transitions stored in $\hat{h}^{(2,1)}$ in the entry $\mathcal{A}_{10,8}^{(3)}$:

$$\hat{p}_{10}^{(2,1)\top} B^{(3,1)} e_8 = \frac{\beta}{4} \begin{pmatrix} 0 \\ \tilde{p}_{5,3} \\ \tilde{p}_{6,3} \\ \tilde{p}_{5,1} \\ \tilde{p}_{6,1} \\ \tilde{p}_{9,3} \\ \tilde{p}_{10,3} \\ \tilde{p}_{9,1} \\ \tilde{p}_{10,1} \\ 0 \end{pmatrix}^\top \begin{pmatrix} 0 & 1 & 1 & 0 & 0 & 1 & 1 & 0 & 0 & 1 \\ 0 & 0 & 1 & 1 & 0 & 0 & 1 & 1 & 0 & 0 \\ 0 & 0 & 0 & 0 & 1 & 1 & 0 & 0 & 1 & 1 \\ 0 & 0 & 0 & 0 & 0 & 1 & 1 & 0 & 0 & 1 \\ 0 & 0 & 0 & 0 & 0 & 0 & 1 & 1 & 0 & 0 \\ 0 & 0 & 0 & 0 & 0 & 0 & 0 & 1 & 1 & 0 \\ 0 & 0 & 0 & 0 & 0 & 0 & 0 & 0 & 1 & 1 \\ 0 & 0 & 0 & 0 & 0 & 0 & 0 & 0 & 0 & 1 \\ 0 & 0 & 0 & 0 & 0 & 0 & 0 & 0 & 0 & 0 \end{pmatrix} \begin{pmatrix} 0 \\ 0 \\ 0 \\ 0 \\ 0 \\ 0 \\ 1 \\ 0 \\ 0 \end{pmatrix} = \frac{\beta}{4} \begin{pmatrix} 0 \\ \tilde{p}_{5,3} \\ \tilde{p}_{6,3} \\ \tilde{p}_{5,1} \\ \tilde{p}_{6,1} \\ \tilde{p}_{9,3} \\ \tilde{p}_{10,3} \\ \tilde{p}_{9,1} \\ \tilde{p}_{10,1} \\ 0 \end{pmatrix}^\top \begin{pmatrix} 1 \\ 1 \\ 0 \\ 0 \\ 1 \\ 1 \\ 0 \\ 0 \\ 0 \end{pmatrix} \tag{23}$$

$$= \frac{\beta}{4} \left( \boxed{\tilde{p}_{10,3}} + \boxed{\tilde{p}_{9,3}} + \boxed{\tilde{p}_{6,3}} + \boxed{\tilde{p}_{5,3}} \right)$$

Notice how the matrix $B^{(3,1)}$ has the same pattern as $A^{(2,1)}$ in Eq. (19) but along the columns instead of the rows. Intuitively it makes sense since we need to sum the same entries resulting from the sum in the previous attention. The matrix $B^{(3,2)}$ acting on second head will have the same pattern but shifted by two positions in order to have the same pattern as $A^{(2,2)}$.

## I.2 EXAMPLE WITH $\mathcal{K} = \{1, 3, 4\}$

The case of two lags $\mathcal{K} = \{1, 3\}$ despite not being contiguous does not adequately represent the general case. Indeed due to the structure, we could always sum two consecutive tokens and therefore recover optimal performance using 2 heads. It is helpful to also consider a case where the lags do not form a structure that allows for fewer heads in the construction. For example the case of three lags $\mathcal{K} = \{1, 3, 4\}$ and $T = 12$:

**First layer:** The main structure of the first layer remains unchanged, the diagonals in the matrix $A^{(1)}$ with positive entries are 2$^{\text{nd}}$, 3$^{\text{rd}}$ and 4$^{\text{th}}$:

$$\tilde{A}^{(1)} = \begin{pmatrix} \log P^\top & 0 \\ 0 & A^{(1)} \end{pmatrix}$$

$$A_{ij}^{(1)} = \begin{cases} +\lambda & \text{if } j - i \in \mathcal{K} \\ -\lambda & \text{if } j - i \notin \mathcal{K}. \end{cases}$$

$$A^{(1)} = \begin{pmatrix} -\lambda & -\lambda & -\lambda & -\lambda & -\lambda & -\lambda & -\lambda & -\lambda & -\lambda & -\lambda & -\lambda \\ -\lambda & -\lambda & -\lambda & -\lambda & -\lambda & -\lambda & -\lambda & -\lambda & -\lambda & -\lambda & -\lambda \\ +\lambda & -\lambda & +\lambda & -\lambda & -\lambda & -\lambda & -\lambda & -\lambda & -\lambda & -\lambda & -\lambda \\ +\lambda & +\lambda & -\lambda & +\lambda & -\lambda & -\lambda & -\lambda & -\lambda & -\lambda & -\lambda & -\lambda \\ -\lambda & +\lambda & +\lambda & -\lambda & +\lambda & -\lambda & -\lambda & -\lambda & -\lambda & -\lambda & -\lambda \\ -\lambda & -\lambda & +\lambda & +\lambda & -\lambda & +\lambda & -\lambda & -\lambda & -\lambda & -\lambda & -\lambda \\ -\lambda & -\lambda & -\lambda & +\lambda & +\lambda & -\lambda & +\lambda & -\lambda & -\lambda & -\lambda & -\lambda \\ -\lambda & -\lambda & -\lambda & -\lambda & +\lambda & +\lambda & -\lambda & +\lambda & -\lambda & -\lambda & -\lambda \\ -\lambda & -\lambda & -\lambda & -\lambda & -\lambda & +\lambda & +\lambda & -\lambda & +\lambda & -\lambda & -\lambda \\ -\lambda & -\lambda & -\lambda & -\lambda & -\lambda & -\lambda & +\lambda & +\lambda & -\lambda & +\lambda & -\lambda \\ -\lambda & -\lambda & -\lambda & -\lambda & -\lambda & -\lambda & -\lambda & +\lambda & +\lambda & -\lambda & +\lambda \end{pmatrix} \tag{24}$$

The output token at index $i$ after the first layer still corresponds to a weighted average of the past tokens $h_{i-k}^{(0)}$ for $k \in \mathcal{K}$ where the weights are given by the normalized probabilities $\tilde{p}_{i,k}$:

$$\mathcal{A}^{(1)} = \begin{pmatrix} 1 & 0 & 0 & 0 & 0 & 0 & 0 & 0 & 0 & 0 & 0 & 0 \\ 1/2 & 1/2 & 0 & 0 & 0 & 0 & 0 & 0 & 0 & 0 & 0 & 0 \\ 1/3 & 1/3 & 1/3 & 0 & 0 & 0 & 0 & 0 & 0 & 0 & 0 & 0 \\ 1/4 & 1/4 & 1/4 & 1/4 & 0 & 0 & 0 & 0 & 0 & 0 & 0 & 0 \\ \tilde{p}_{5,4} & \tilde{p}_{5,3} & 0 & \tilde{p}_{5,1} & 0 & 0 & 0 & 0 & 0 & 0 & 0 & 0 \\ 0 & \tilde{p}_{6,4} & \tilde{p}_{6,3} & 0 & \tilde{p}_{6,1} & 0 & 0 & 0 & 0 & 0 & 0 & 0 \\ 0 & 0 & \tilde{p}_{7,4} & \tilde{p}_{7,3} & 0 & \tilde{p}_{7,1} & 0 & 0 & 0 & 0 & 0 & 0 \\ 0 & 0 & 0 & \tilde{p}_{8,4} & \tilde{p}_{8,3} & 0 & \tilde{p}_{8,1} & 0 & 0 & 0 & 0 & 0 \\ 0 & 0 & 0 & 0 & \tilde{p}_{9,4} & \tilde{p}_{9,3} & 0 & \tilde{p}_{9,1} & 0 & 0 & 0 & 0 \\ 0 & 0 & 0 & 0 & 0 & \tilde{p}_{10,4} & \tilde{p}_{10,3} & 0 & \tilde{p}_{10,1} & 0 & 0 & 0 \\ 0 & 0 & 0 & 0 & 0 & 0 & \tilde{p}_{11,4} & \tilde{p}_{11,3} & 0 & \tilde{p}_{11,1} & 0 & 0 \\ 0 & 0 & 0 & 0 & 0 & 0 & 0 & \tilde{p}_{12,4} & \tilde{p}_{12,3} & 0 & \tilde{p}_{12,1} & 0 \end{pmatrix}$$

$$\hat{h}^{(1)} = \begin{pmatrix} \tilde{s}_1 & \tilde{s}_2 & \tilde{s}_3 & \tilde{s}_4 & \tilde{s}_5 & \tilde{s}_6 & \tilde{s}_7 & \tilde{s}_8 & \tilde{s}_9 & \tilde{s}_{10} & \tilde{s}_{11} & \tilde{s}_{12} \\ 1 & 1/2 & 1/3 & 1/4 & \tilde{p}_{5,4} & 0 & 0 & 0 & 0 & 0 & & \\ 0 & 1/2 & 1/3 & 1/4 & \tilde{p}_{5,3} & \tilde{p}_{6,4} & 0 & 0 & 0 & 0 & 0 & 0 \\ 0 & 0 & 1/3 & 1/4 & 0 & \tilde{p}_{6,3} & \tilde{p}_{7,4} & 0 & 0 & 0 & 0 & 0 \\ 0 & 0 & 0 & 1/4 & \tilde{p}_{5,1} & 0 & \tilde{p}_{7,3} & \tilde{p}_{8,4} & 0 & 0 & 0 & 0 \\ 0 & 0 & 0 & 0 & 0 & \tilde{p}_{6,1} & 0 & \tilde{p}_{8,3} & \tilde{p}_{9,4} & 0 & 0 & 0 \\ 0 & 0 & 0 & 0 & 0 & 0 & \tilde{p}_{7,1} & 0 & \tilde{p}_{9,3} & \tilde{p}_{10,4} & 0 & 0 \\ 0 & 0 & 0 & 0 & 0 & 0 & 0 & \tilde{p}_{8,1} & 0 & \tilde{p}_{10,3} & \tilde{p}_{11,4} & 0 \\ 0 & 0 & 0 & 0 & 0 & 0 & 0 & 0 & \tilde{p}_{9,1} & 0 & \tilde{p}_{11,3} & \tilde{p}_{12,4} \\ 0 & 0 & 0 & 0 & 0 & 0 & 0 & 0 & 0 & \tilde{p}_{10,1} & 0 & \tilde{p}_{12,3} \\ 0 & 0 & 0 & 0 & 0 & 0 & 0 & 0 & 0 & 0 & \tilde{p}_{11,1} & 0 \\ 0 & 0 & 0 & 0 & 0 & 0 & 0 & 0 & 0 & 0 & 0 & \tilde{p}_{12,1} \\ 0 & 0 & 0 & 0 & 0 & 0 & 0 & 0 & 0 & 0 & 0 & 0 \end{pmatrix} \tag{25}$$

**Second layer, aggregation of transition probabilities:** In this case we can't use anymore the fact that 2 consecutive tokens can be summed without mixing the information. In fact, summing the last tow tokens $\hat{h}_{11}^{(1)}$ and $\hat{h}_{12}^{(2)}$ would now result in the mixing of $\tilde{p}_{11,3}$ and $\tilde{p}_{12,4}$. In order to avoid this, the only possibility is to sum one token every 4 similar to the case where we would have 4 contiguous lags. This solution is less efficient because summing each 4 tokens while having the missing transition corresponding to lag 2 leaves an empty element in the embedding of the token and adds an additional head, increasing both the dimension and the number of parameters. This means that even if we only have 3 lags, in order to not have any overlap we still need 4 attention heads for our construction to not mix the information. Each head has the pattern $(0,0,0,1)$ shifted by one position as if the lags would be 1,2,3,4:

**Third layer** For the third layer we use again the construction with the positional encoding that was introduced in App H:

$$A_{ij}^{(3)} = \lambda_1 \begin{cases} +1 & \text{if } j-i+1 \in \mathcal{K} \\ -1 & \text{if } j-i+1 \notin \mathcal{K} \end{cases} \qquad A^{(3)} =$$

For the selective sum, the matrices $B^{(3,1)}, \dots, B^{(3,4)}$ have the same form as before but considering now the fact that even if we only have 3 lags in the set, we still need 4 heads:

$$B_{ij}^{(3,h)} = \beta \begin{cases} +1, & \text{if } \left( (i-j-h+1) \mod \hat{k} - \min(\mathcal{K}) + 1 = 0 \right) \\ 0, & \text{otherwise} \end{cases}$$

The computation related to the matrix $B^{(3,1)}$ in the attention are reported in the following:

$$= \frac{\beta}{2} \left( \tilde{p}_{12,3} + \tilde{p}_{8,3} \right)$$

## J  CONSTRUCTION FOR TWO LAGS AND SINGLE HEAD

In the constructions illustrated so far, in order to store all the transitions in the history of the current token and not lose any information, we had to scale the number of heads at least as the number of lags in the task $K$. This allows to achieve optimal sample complexity. However, driven by experimental evidence, we observed that scaling the number of heads as the number of lags is not necessary in the special case of $|\mathcal{K}| = 2$. In this case indeed, there exists a solution, which transformers can learn, that achieves optimal sample complexity using only one head in the second layer. In the following we will report the construction that proves the previous statement while illustrating it for the case of $\mathcal{K} = \{1, 3\}$ analogous to Section I.

**First layer:**  The structure of the first layer remains unchanged from Section 4. The important difference is that now the diagonals in the matrix $A^{(1)}$ with positive entries are only $2^{\text{nd}}$ and $4^{\text{th}}$:

$$\widetilde{A}^{(1)} = \begin{pmatrix} \log P^\top & 0 \\ 0 & A^{(1)} \end{pmatrix}$$

$$A^{(1)}_{ij} = \begin{cases} +\lambda & \text{if } j - i \in \mathcal{K} \\ -\lambda & \text{if } j - i \notin \mathcal{K}. \end{cases}$$

$$A^{(1)} = \begin{pmatrix} \text{-}\lambda & \text{-}\lambda & \text{-}\lambda & \text{-}\lambda & \text{-}\lambda & \text{-}\lambda & \text{-}\lambda & \text{-}\lambda & \text{-}\lambda & \text{-}\lambda & \text{-}\lambda & \text{-}\lambda \\ \text{-}\lambda & \text{-}\lambda & \text{-}\lambda & \text{-}\lambda & \text{-}\lambda & \text{-}\lambda & \text{-}\lambda & \text{-}\lambda & \text{-}\lambda & \text{-}\lambda & \text{-}\lambda & \text{-}\lambda \\ +\lambda & \text{-}\lambda & +\lambda & \text{-}\lambda & \text{-}\lambda & \text{-}\lambda & \text{-}\lambda & \text{-}\lambda & \text{-}\lambda & \text{-}\lambda & \text{-}\lambda & \text{-}\lambda \\ \text{-}\lambda & +\lambda & \text{-}\lambda & +\lambda & \text{-}\lambda & \text{-}\lambda & \text{-}\lambda & \text{-}\lambda & \text{-}\lambda & \text{-}\lambda & \text{-}\lambda & \text{-}\lambda \\ \text{-}\lambda & \text{-}\lambda & +\lambda & \text{-}\lambda & +\lambda & \text{-}\lambda & \text{-}\lambda & \text{-}\lambda & \text{-}\lambda & \text{-}\lambda & \text{-}\lambda & \text{-}\lambda \\ \text{-}\lambda & \text{-}\lambda & \text{-}\lambda & +\lambda & \text{-}\lambda & +\lambda & \text{-}\lambda & \text{-}\lambda & \text{-}\lambda & \text{-}\lambda & \text{-}\lambda & \text{-}\lambda \\ \text{-}\lambda & \text{-}\lambda & \text{-}\lambda & \text{-}\lambda & +\lambda & \text{-}\lambda & +\lambda & \text{-}\lambda & \text{-}\lambda & \text{-}\lambda & \text{-}\lambda & \text{-}\lambda \\ \text{-}\lambda & \text{-}\lambda & \text{-}\lambda & \text{-}\lambda & \text{-}\lambda & +\lambda & \text{-}\lambda & +\lambda & \text{-}\lambda & \text{-}\lambda & \text{-}\lambda & \text{-}\lambda \\ \text{-}\lambda & \text{-}\lambda & \text{-}\lambda & \text{-}\lambda & \text{-}\lambda & \text{-}\lambda & +\lambda & \text{-}\lambda & +\lambda & \text{-}\lambda & \text{-}\lambda & \text{-}\lambda \\ \text{-}\lambda & \text{-}\lambda & \text{-}\lambda & \text{-}\lambda & \text{-}\lambda & \text{-}\lambda & \text{-}\lambda & +\lambda & \text{-}\lambda & +\lambda & \text{-}\lambda \end{pmatrix}$$

Remarking that each input element $s_i$ is encoded as $h_i^{(0)} = [e_{s_i}, e_i] \in \{0,1\}^{|\mathcal{S}|+T}$, the output token at index $i$ after the first layer still corresponds to a weighted average of the past tokens $h_{i-k}^{(0)}$ for $k \in \mathcal{K}$ where the weights are given by the normalized probabilities $\tilde{p}_{i,k}$:

$$\hat{h}_i^{(1)} = \text{Attn}(h_{1:T}^{(0)}; \tilde{A}^{(1)})_i = \begin{cases} \sum_{j=1}^{i} \mathbb{1}\left[i - j \in \mathcal{K}\right] \dfrac{P_{s_j, s_i}}{\sum_{r \in \mathcal{K}} P_{s_r, s_i}} h_j^{(0)} & \text{if } i > 1 \\ h_1^{(0)} & \text{if } i = 1 \end{cases}$$

$$= \begin{cases} \sum_{k \in \mathcal{K}, k < i} \tilde{p}_{i,k} h_{i-k}^{(0)} & \text{if } i > 1 \\ h_1^{(0)} & \text{if } i = 1 \end{cases}$$

Due to the lack of the entries on the $3^{\text{rd}}$ diagonal, both the attention and the output token will change accordingly:

$$\mathcal{A}^{(1)} = \begin{pmatrix} 1 & 0 & 0 & 0 & 0 & 0 & 0 & 0 & 0 & 0 \\ 1/2 & 1/2 & 0 & 0 & 0 & 0 & 0 & 0 & 0 & 0 \\ 1/3 & 1/3 & 1/3 & 0 & 0 & 0 & 0 & 0 & 0 & 0 \\ \tilde{p}_{4,3} & 0 & \tilde{p}_{4,1} & 0 & 0 & 0 & 0 & 0 & 0 & 0 \\ 0 & \tilde{p}_{5,3} & 0 & \tilde{p}_{5,1} & 0 & 0 & 0 & 0 & 0 & 0 \\ 0 & 0 & \tilde{p}_{6,3} & 0 & \tilde{p}_{6,1} & 0 & 0 & 0 & 0 & 0 \\ 0 & 0 & 0 & \tilde{p}_{7,3} & 0 & \tilde{p}_{7,1} & 0 & 0 & 0 & 0 \\ 0 & 0 & 0 & 0 & \tilde{p}_{8,3} & 0 & \tilde{p}_{8,1} & 0 & 0 & 0 \\ 0 & 0 & 0 & 0 & 0 & \tilde{p}_{9,3} & 0 & \tilde{p}_{9,1} & 0 & 0 \\ 0 & 0 & 0 & 0 & 0 & 0 & \tilde{p}_{1,3} & 0 & \tilde{p}_{10,1} & 0 \end{pmatrix} \quad \hat{h}^{(1)} = \begin{pmatrix} \tilde{s}_1 & \tilde{s}_2 & \tilde{s}_3 & \tilde{s}_4 & \tilde{s}_5 & \tilde{s}_6 & \tilde{s}_7 & \tilde{s}_8 & \tilde{s}_9 & \tilde{s}_{10} \\ 1 & 1/2 & 1/3 & \tilde{p}_{4,3} & 0 & 0 & 0 & 0 & 0 & 0 \\ 0 & 1/2 & 1/3 & 0 & \tilde{p}_{5,3} & 0 & 0 & 0 & 0 & 0 \\ 0 & 0 & 1/3 & \tilde{p}_{4,1} & 0 & \tilde{p}_{6,3} & 0 & 0 & 0 & 0 \\ 0 & 0 & 0 & 0 & \tilde{p}_{5,1} & 0 & \tilde{p}_{7,3} & 0 & 0 & 0 \\ 0 & 0 & 0 & 0 & 0 & \tilde{p}_{6,1} & 0 & \tilde{p}_{8,3} & 0 & 0 \\ 0 & 0 & 0 & 0 & 0 & 0 & \tilde{p}_{7,1} & 0 & \tilde{p}_{9,3} & 0 \\ 0 & 0 & 0 & 0 & 0 & 0 & 0 & \tilde{p}_{8,1} & 0 & \tilde{p}_{10,3} \\ 0 & 0 & 0 & 0 & 0 & 0 & 0 & 0 & \tilde{p}_{9,1} & 0 \\ 0 & 0 & 0 & 0 & 0 & 0 & 0 & 0 & 0 & \tilde{p}_{10,1} \\ 0 & 0 & 0 & 0 & 0 & 0 & 0 & 0 & 0 & 0 \end{pmatrix} \quad (26)$$

**Second Layer:**  the second layer uses only the first head $\tilde{A}^{(2)} = \tilde{A}^{(2,1)}$ compared to the construction illustrated in Section I and the matrix $A^{(2)} = A^{(2,1)}$ remains identical.

$$\tilde{A}^{(2)} = \begin{pmatrix} 0 & 0 & \vdots & 0 \\ 0 & A^{(2)} & \vdots & \\ \cdots & \cdots & \vdots & \cdots \\ 0 & \vdots & 0 \end{pmatrix} \qquad A^{(2)} = \begin{pmatrix} \text{-}\lambda & \text{-}\lambda & \text{-}\lambda & \text{-}\lambda & \text{-}\lambda & \text{-}\lambda & \text{-}\lambda & \text{-}\lambda & \text{-}\lambda & \text{-}\lambda \\ \text{-}\lambda & \text{-}\lambda & \text{-}\lambda & \text{-}\lambda & \text{-}\lambda & \text{-}\lambda & \text{-}\lambda & \text{-}\lambda & \text{-}\lambda & \text{-}\lambda \\ \text{-}\lambda & \text{-}\lambda & \text{-}\lambda & \text{-}\lambda & \text{-}\lambda & \text{-}\lambda & \text{-}\lambda & \text{-}\lambda & \text{-}\lambda & \text{-}\lambda \\ \text{-}\lambda & \text{-}\lambda & \text{-}\lambda & +\lambda & \text{-}\lambda & \text{-}\lambda & \text{-}\lambda & \text{-}\lambda & \text{-}\lambda & \text{-}\lambda \\ \text{-}\lambda & \text{-}\lambda & \text{-}\lambda & +\lambda & +\lambda & \text{-}\lambda & \text{-}\lambda & \text{-}\lambda & \text{-}\lambda & \text{-}\lambda \\ \text{-}\lambda & \text{-}\lambda & \text{-}\lambda & \text{-}\lambda & +\lambda & +\lambda & \text{-}\lambda & \text{-}\lambda & \text{-}\lambda & \text{-}\lambda \\ \text{-}\lambda & \text{-}\lambda & \text{-}\lambda & \text{-}\lambda & \text{-}\lambda & +\lambda & +\lambda & \text{-}\lambda & \text{-}\lambda & \text{-}\lambda \\ \text{-}\lambda & \text{-}\lambda & \text{-}\lambda & +\lambda & \text{-}\lambda & \text{-}\lambda & +\lambda & +\lambda & \text{-}\lambda & \text{-}\lambda \\ \text{-}\lambda & \text{-}\lambda & \text{-}\lambda & +\lambda & +\lambda & \text{-}\lambda & \text{-}\lambda & +\lambda & +\lambda & \text{-}\lambda \\ \text{-}\lambda & \text{-}\lambda & \text{-}\lambda & \text{-}\lambda & +\lambda & +\lambda & \text{-}\lambda & \text{-}\lambda & +\lambda & +\lambda \end{pmatrix}$$

For the case of two lags examined here, we can derive a mathematical expression for the matrix $A^{(2)}$ which is valid for any set of lags. For convenience we introduce $\bar{k} = \min \mathcal{K}$:

$$A^{(2)}_{i,j} = \begin{cases} 0, & \text{if } i < \hat{k} \text{ or } j < \hat{k}, \\ \lambda, & \text{if } j \leq i \text{ and } \left(|i - j| \mod 2(\hat{k} - \bar{k})\right) < (\hat{k} - \bar{k}), \\ 0, & \text{otherwise} \end{cases} \quad (27)$$

where the first condition ensures that all elements in the first $\hat{k}$ rows and the first $\hat{k}$ columns of the matrix are zero. The condition $j \leq i$ ensures that only the lower triangular part of the matrix (including the diagonal). Finally, the condition $(|i - j| \mod 2d) < d$ introduces a periodic pattern within the lower triangular part of the matrix. The modulo operation creates a repeating cycle of length $2(\hat{k} - \bar{k})$, and the condition $< (\hat{k} - \bar{k})$ determines whether to place a one or a zero within each cycle segment.

**Third layer:** the third layer instead has a different structure. As before, there are only two non-zero blocks $A^{(3)}$ and $B^{(3)}$ but the latter appears in the transpose position compared to the previous constructions:

$$
\tilde{A}^{(3)} = \begin{pmatrix}
\begin{smallmatrix} 0 & 0 \\ 0 & A^{(3)} \end{smallmatrix} & 0 & 0 & \begin{smallmatrix} 0 & 0 \\ 0 & B^{(3)} \end{smallmatrix} \\
0 & 0 & 0 & 0 \\
0 & 0 & 0 & 0 \\
0 & 0 & 0 & 0
\end{pmatrix}
\quad
A^{(3)} = \begin{pmatrix} \cdots \end{pmatrix}
\quad
B^{(3)} = \beta \begin{pmatrix} \cdots \end{pmatrix}
\tag{28}
$$

The matrix $A^{(3)}$ remains unchanged and has positive entries along the diagonals, corresponding to the lags shifted by one position. The main difference lies in the matrix $B^{(3)}$, which now includes negative entries in positions that previously contained zeros. The general formulation of this matrix is the following:

$$
A_{ij}^{(3)} = \begin{cases} +\lambda & \text{if } j - i + 1 \in \mathcal{K} \\ -\lambda & \text{if } j - i + 1 \notin \mathcal{K} \end{cases}
\qquad
B_{i,j}^{(3)} = \begin{cases} 0, & \text{if } j \geq i, \\ +\beta, & \text{if } \left( (i - j - 1) \mod 2(\hat{k} - \bar{k}) \right) < (\hat{k} - \bar{k}), \\ -\beta, & \text{otherwise.} \end{cases}
$$

So far the matrix $B^{(3)}$ has been structured such that it would compute the selective sum of the normalized transition of the lag of the corresponding entry in the attention: $\tilde{A}_{ij}^{(3)} \propto h_i^{(2)\top} \tilde{A}^{(3)} h_{i-k+1}^{(2)} \propto \sum_{j \leq i} \tilde{p}_{j,k}$, where $i - k + 1$ are the only non-zero entries due to $A^{(3)}$ after applying softmax. To understand the impact of having negative entries, let us consider the previous example for the case of $\mathcal{K} = \{1, 3\}$ and the output of the second attention for the 8th, 9th and 10th token:

$$
\hat{h}_{10}^{(2)} = 1/4 \cdot \left( \sum_{i=5,6,9,10} e_{s_i} \; 0 \; 0 \; 0 \; 0 \; 1 \; 1 \; 0 \; 0 \; 1 \; 1 \; \sum_{i=5,6,9,10} \tilde{s}_i \; 0 \; \tilde{p}_{5,3} \; \tilde{p}_{6,3} \; \tilde{p}_{5,1} \; \tilde{p}_{6,1} \; \tilde{p}_{9,3} \; \tilde{p}_{10,3} \; \tilde{p}_{9,1} \; \tilde{p}_{10,1} \; 0 \right)
$$

$$
\hat{h}_{9}^{(2)} = 1/4 \cdot \left( \sum_{i=4,5,8,9} e_{s_i} \; 0 \; 0 \; 0 \; 1 \; 1 \; 0 \; 0 \; 1 \; 1 \; 0 \; \sum_{i=4,5,8,9} \tilde{s}_i \; \tilde{p}_{4,3} \; \tilde{p}_{5,3} \; \tilde{p}_{4,1} \; \tilde{p}_{5,1} \; \tilde{p}_{8,3} \; \tilde{p}_{9,3} \; \tilde{p}_{8,1} \; \tilde{p}_{9,1} \; 0 \; 0 \right)
$$

$$
\hat{h}_{8}^{(2)} = 1/3 \cdot \left( \sum_{i=4,7,8} e_{s_i} \; 0 \; 0 \; 0 \; 1 \; 0 \; 0 \; 1 \; 1 \; 0 \; 0 \; \sum_{i=4,7,8} \tilde{s}_i \; \tilde{p}_{4,3} \; 0 \; \tilde{p}_{4,1} \; \tilde{p}_{7,3} \; \tilde{p}_{8,3} \; \tilde{p}_{7,1} \; \tilde{p}_{8,1} \; 0 \; 0 \; 0 \right)
$$

$$
\underbrace{\hphantom{\tilde{p}_{4,3} \; 0 \; \tilde{p}_{4,1} \; \tilde{p}_{7,3} \; \tilde{p}_{8,3} \; \tilde{p}_{7,1} \; \tilde{p}_{8,1}}}_{\hat{p}_8^{(2)}}
$$

and define $\hat{p}_i^{(2)} \in \mathbb{R}^T$ as the block of $\hat{h}_i^{(2)}$ which contains the normalized transition probabilities such that $\hat{h}_i^{(2)} = [\sum_{j \in N_i} e_{s_j}, \hat{m}_i^{(2)}, \sum_{j \in N_i} \tilde{s}_j, \hat{p}_i^{(2)}]$. By the different structure in Eq. (28) we can see how, when computing the attention for the concatenated tokens $h_i^{(2)} = [[e_{s_i}, e_i], \hat{h}_i^{(1)}, \hat{h}_i^{(2)}]$, the order of the multiplication has been reversed ($e_i$ is now on the left) and the matrix $B^{(3)}$ is applied to $\hat{p}_j^{(2)}$:

$$
h_i^{(2)\top} \tilde{A}^{(3)} h_j^{(2)} = e_i^\top B^{(3)} \hat{p}_j^{(2)} + e_i A^{(3)} e_j .
\tag{29}
$$

To better understand the implications of the reverse order in the multiplication and the presence of negative entries, consider the product $e_{10}^\top B^{(3,1)} \hat{p}_8^{(2)}$ in Eq. (31), which sums the transitions stored in

$\hat{h}^{(2,1)}$ which, after softmax, will correspond to $\mathcal{A}_{10,8}^{(3)}$:

$$
e_{10}^\top B^{(3)} \hat{p}_8^{(2)\top} = \frac{\beta}{3}
\begin{pmatrix} 0 \\ 0 \\ 0 \\ 0 \\ 0 \\ 0 \\ 0 \\ 0 \\ 0 \\ 1 \end{pmatrix}^\top
\begin{pmatrix}
0 & 0 & 0 & 0 & 0 & 0 & 0 & 0 & 0 & 0 \\
1 & 0 & 0 & 0 & 0 & 0 & 0 & 0 & 0 & 0 \\
1 & 1 & 0 & 0 & 0 & 0 & 0 & 0 & 0 & 0 \\
-1 & 1 & 1 & 0 & 0 & 0 & 0 & 0 & 0 & 0 \\
-1 & -1 & 1 & 1 & 0 & 0 & 0 & 0 & 0 & 0 \\
1 & -1 & -1 & 1 & 1 & 0 & 0 & 0 & 0 & 0 \\
1 & 1 & -1 & -1 & 1 & 1 & 0 & 0 & 0 & 0 \\
-1 & 1 & 1 & -1 & -1 & 1 & 1 & 0 & 0 & 0 \\
-1 & -1 & 1 & 1 & -1 & -1 & 1 & 1 & 0 & 0 \\
1 & -1 & -1 & 1 & 1 & -1 & -1 & 1 & 1 & 0
\end{pmatrix}
\begin{pmatrix} \tilde{p}_{4,3} \\ 0 \\ \tilde{p}_{4,1} \\ \tilde{p}_{7,3} \\ \tilde{p}_{8,3} \\ \tilde{p}_{7,1} \\ \tilde{p}_{8,1} \\ 0 \\ 0 \\ 0 \end{pmatrix}
= \frac{\beta}{3}
\begin{pmatrix} 1 \\ -1 \\ -1 \\ 1 \\ 1 \\ -1 \\ -1 \\ 1 \\ 1 \\ 0 \end{pmatrix}^\top
\begin{pmatrix} \tilde{p}_{4,3} \\ 0 \\ \tilde{p}_{4,1} \\ \tilde{p}_{7,3} \\ \tilde{p}_{8,3} \\ \tilde{p}_{7,1} \\ \tilde{p}_{8,1} \\ 0 \\ 0 \\ 0 \end{pmatrix}
$$

$$
= \frac{\beta}{3} \left( \tilde{p}_{8,3} + \tilde{p}_{7,3} + \tilde{p}_{4,3} - \tilde{p}_{8,1} - \tilde{p}_{7,1} - \tilde{p}_{4,1} \right)
\tag{30}
$$

where we observe how, the product involving $B^{(3)}$, is now not only computing the sum of transitions for the lag 3 as for the previous constructions to copy the $8^{\text{th}}$ to predict the $11^{\text{th}}$, it is also subtracting all the transitions of lag 3. To fully understand the implications, we also consider the entry of the attention correspondent to the other lag in the set, 1 and the relative product $e_{10}^\top B^{(3,1)} \hat{p}_{10}^{(2)}$:

$$
e_{10}^\top B^{(4)} \hat{p}_{10}^{(2)\top} = \frac{\beta}{3}
\begin{pmatrix} 0 \\ 0 \\ 0 \\ 0 \\ 0 \\ 0 \\ 0 \\ 0 \\ 0 \\ 1 \end{pmatrix}^\top
\begin{pmatrix}
0 & 0 & 0 & 0 & 0 & 0 & 0 & 0 & 0 & 0 \\
1 & 0 & 0 & 0 & 0 & 0 & 0 & 0 & 0 & 0 \\
1 & 1 & 0 & 0 & 0 & 0 & 0 & 0 & 0 & 0 \\
-1 & 1 & 1 & 0 & 0 & 0 & 0 & 0 & 0 & 0 \\
-1 & -1 & 1 & 1 & 0 & 0 & 0 & 0 & 0 & 0 \\
1 & -1 & -1 & 1 & 1 & 0 & 0 & 0 & 0 & 0 \\
1 & 1 & -1 & -1 & 1 & 1 & 0 & 0 & 0 & 0 \\
-1 & 1 & 1 & -1 & -1 & 1 & 1 & 0 & 0 & 0 \\
-1 & -1 & 1 & 1 & -1 & -1 & 1 & 1 & 0 & 0 \\
1 & -1 & -1 & 1 & 1 & -1 & -1 & 1 & 1 & 0
\end{pmatrix}
\begin{pmatrix} 0 \\ \tilde{p}_{5,3} \\ \tilde{p}_{6,3} \\ \tilde{p}_{5,1} \\ \tilde{p}_{6,1} \\ \tilde{p}_{9,3} \\ \tilde{p}_{10,3} \\ \tilde{p}_{9,1} \\ \tilde{p}_{10,1} \\ 0 \end{pmatrix}
= \frac{\beta}{4}
\begin{pmatrix} 1 \\ -1 \\ -1 \\ 1 \\ 1 \\ -1 \\ -1 \\ 1 \\ 1 \\ 0 \end{pmatrix}^\top
\begin{pmatrix} 0 \\ \tilde{p}_{5,3} \\ \tilde{p}_{6,3} \\ \tilde{p}_{5,1} \\ \tilde{p}_{6,1} \\ \tilde{p}_{9,3} \\ \tilde{p}_{10,3} \\ \tilde{p}_{9,1} \\ \tilde{p}_{10,1} \\ 0 \end{pmatrix}
$$

$$
= \frac{\beta}{4} \left( \tilde{p}_{10,1} + \tilde{p}_{9,1} + \tilde{p}_{6,1} + \tilde{p}_{5,1} - \tilde{p}_{10,3} - \tilde{p}_{9,3} - \tilde{p}_{6,3} - \tilde{p}_{5,3} \right)
\tag{31}
$$

therefore both products contains the sum of the transitions for the respective lags and the negative sum of the other lag and notice how they are all computed on different elements of the past. The first one contains the transitions for the tokens 8,7,4 whereas the second one contains the remaining ones 10,9,6,5. If we now compute the softmax:

$$
\mathcal{A}_{10,10} = \frac{\exp\left( e_{10}^\top B^{(3)} \hat{p}_{10}^{(2)} + \lambda \right)}{\sum_{\substack{i=1 \\ i \neq 8}}^{9} \exp\left( e_{10}^\top B^{(3)} \hat{p}_j^{(2)} - \lambda \right) + \exp\left( e_{10}^\top B^{(3)} \hat{p}_8^{(2)} + \lambda \right) + \exp\left( e_{10}^\top B^{(3)} \hat{p}_{10}^{(2)} + \lambda \right)}
$$

$$
= \frac{\exp\left( e_{10}^\top B^{(3)} \hat{p}_{10}^{(2)} \right)}{\sum_{\substack{i=1 \\ i \neq 8}}^{9} \exp\left( e_{10}^\top B^{(3)} \hat{p}_j^{(2)} - 2\lambda \right) + \exp\left( e_{10}^\top B^{(3)} \hat{p}_8^{(2)} \right) + \exp\left( e_{10}^\top B^{(3)} \hat{p}_{10}^{(2)} \right)}
$$

Considering the limit of $\lambda \to \infty$:

$$
\lim_{\lambda \to \infty} \mathcal{A}_{10,10} = \frac{\exp\left( e_{10}^\top B^{(3)} \hat{p}_{10}^{(2)} \right)}{\exp\left( e_{10}^\top B^{(3)} \hat{p}_8^{(2)} \right) + \exp\left( e_{10}^\top B^{(3)} \hat{p}_{10}^{(2)} \right)}
$$

$$
= \frac{1}{\exp\left( e_{10}^\top B^{(3)} \hat{p}_8^{(2)} - e_{10}^\top B^{(3)} \hat{p}_{10}^{(2)} \right) + 1}
$$

$$
= \frac{1}{\exp\left( + \frac{\beta}{3} \sum_{i \in \{8,7,4\}} \tilde{p}_{i,3} - \frac{\beta}{3} \sum_{i \in \{8,7,4\}} \tilde{p}_{i,1} - \frac{\beta}{4} \sum_{i \in \{10,9,6,5\}} \tilde{p}_{i,1} + \sum_{i \in \{10,9,6,5\}} \tilde{p}_{i,3} \right) + 1}
$$

which is considering all the possible transitions as for the case of two heads therefore achieving optimal sample complexity.

