# OpenReview forum: "Selective Induction Heads: How Transformers Select Causal Structures in Context"
_ICLR.cc/2025/Conference — ICLR 2025 Poster_

### Official Review · Reviewer_Lmxm · 2024-10-26

**Soundness:** 3
**Presentation:** 3
**Contribution:** 2
**Rating:** 5
**Confidence:** 3

**Summary:**

This paper studies the ability of transformers to identify the causal dependency in Interleaved Markov Chains. The authors start by introducing Interleaved Markov Chains as a Markov Chain with a random lag. Then, the authors give an explicit construction to show a transformer can implement a predictor for the in-context selection of interleaved Markov chains. Finally, the authors train a transformer on interleaved Markov chains and show that the transformer not only solves the detection task but also exhibits similar attention maps to the theoretical construction given by the authors.

**Strengths:**

* The paper provides strong evidence that transformers can identify causal structures in the context.
* It is nice to see that the theoretical construction can match the experimental findings in a controlled experiment.
* Some slightly more general cases are discussed: Single-head transformers & Non-contiguous lags.

**Weaknesses:**

* The scope of this study is a bit narrow. The authors only discussed the detection of interleaved Markov chains
* It is unclear whether an LLM can detect interleaved Markov chains. Maybe the authors can do some inferences on open-source models to verify this.
* It remains uncertain whether an LLM can retain its ability to detect interleaved Markov chains when trained on data that is unrelated or only loosely related to such structures. Given the artificial nature of this task, it's more effective to assess detection ability in an out-of-distribution setting rather than through direct training on the same task.

**Questions:**

* Do you have any recommendations for an LLM to improve its understanding and learning of causal structures?

---

> ### Author Response · Authors · 2024-11-23
>
> We thank the Reviewer for their feedback and questions. We provide our responses below:
>
> **Weaknesses:**
> > The scope of this study is a bit narrow. The authors only discussed the detection of interleaved Markov chains
>
> We clarify that **our paper focuses on next-token prediction in interleaved Markov chains, not detection**. This framework serves as a tool to investigate and demonstrate how transformers dynamically select causal structures in context. The task enables us to derive an interpretable solution, providing insights into transformers' capabilities beyond this specific setup. This effort is part of the general line of research on mechanistic interpretability, which investigates how transformers learn to solve next-token prediction tasks in simplified setups such as Markov Chains, linear regression, and other dynamical systems.
>
>
> > It is unclear whether an LLM can detect interleaved Markov chains. Maybe the authors can do some inferences on open-source models to verify this.
>
> We clarify that this study does not aim to assess LLMs’ performance on interleaved Markov chains. Instead, it uses this controlled setting to investigate the underlying capabilities of transformers in sequence modelling. While applying this framework to open-source LLMs is an interesting direction for future research, it is beyond the scope of our work.
>
> > It remains uncertain whether an LLM can retain its ability to detect interleaved Markov chains when trained on data that is unrelated or only loosely related to such structures. Given the artificial nature of this task, it's more effective to assess detection ability in an out-of-distribution setting rather than through direct training on the same task.
>
> Our preliminary experiments show that transformers do not generalize to causal structures unseen during training, indicating limited generalization to out-of-distribution data. However, a full evaluation of this interesting behaviour in LLMs is beyond the scope of our work, as mentioned above.
>
>
> **Questions:**
>
> > Do you have any recommendations for an LLM to improve its understanding and learning of causal structures?
>
> In our experiments, we observed that setting the softmax temperature in the first attention layer to $1$—instead of using the standard scaling factor $\sqrt{d_{QK}}$—appears to accelerate the learning of causal structures in the data. This finding suggests that exploring similar adjustments could potentially enhance large language models’ ability to learn and represent causal structures.

---

> ### Comment · Reviewer_Lmxm · 2024-11-26
> **Post-rebuttal response**
>
> I appreciate the authors' detailed response during the rebuttal phase. Upon reviewing it, I believe the authors would benefit from adopting a broader perspective to more effectively convey why this problem/setting is significant. While there is a distinction between large language models (LLMs) and transformers based on model size and behavior, the attempt to simplify the analysis using a few transformer blocks does not necessarily provide conclusions that generalize to larger models.
>
> For example, in response to the question, "Do you have any recommendations for an LLM to improve its understanding and learning of causal structures?" the authors suggested, "setting the softmax temperature in the first attention layer to 1 instead of $\sqrt{d_K}$ using the standard scaling factor appears to accelerate the learning of causal structures in the data." While interesting, this ad-hoc adjustment is unlikely to translate effectively to larger models trained on extensive datasets and optimized with fine-tuned hyperparameters.
>
> Given my continued skepticism regarding the practical implications for real-world large model scenarios, my score remains unchanged.

---

> > ### Author Response · Authors · 2024-11-28
> > **Further clarification**
> >
> > Dear Reviewer,
> >
> > Thank you for your feedback, we would like to offer further clarification.
> >
> > First, in our previous response, our intention was not to suggest that the ad-hoc adjustment of setting the softmax temperature in the first attention layer to 1 should be directly implemented in large language models as-is. Instead, our findings indicate that adjusting the temperature parameter within the attention mechanism can influence a model’s ability to learn causal structures, which may be a promising direction for enhancing these abilities also in large language models.
> >
> > Second, we understand your concern regarding the practical implications of theoretical work for real-world large language model scenarios. However, we emphasize that a direct application to LLMs is not the scope of our work, we focus on the general capabilities of transformer models.
> > Moreover, we respectfully believe that ICLR has a strong tradition of valuing theoretical contributions that deepen our understanding of machine learning, even when their practical applications may not be immediate.
> >
> > Best regards,
> >
> > The Authors

---

### Official Review · Reviewer_DFwY · 2024-11-02

**Soundness:** 3
**Presentation:** 3
**Contribution:** 3
**Rating:** 8
**Confidence:** 2

**Summary:**

The paper mainly extends the work of Nichani et al. (2024). Nichani et al. show that Transformers can learn causal structures of Markov chains and estimate transitional probabilities via In-Context Learning (ICL). The current works investigates whether Transformers can learn causal structures from a more complicated variant of that task - particularly from interleaved Markov chain, where a prediction at position t, is not dependent on the state a t-1 necessarily, but some state at t-k. So in this context, Transformers have to not only learn the transition probabilities and Markov chain dependencies, but also have to learn to select the right Markov chain for prediction at t ignoring other interleaved ones that are not relevant for prediction at t.

The paper makes a theoretical construction based on Disentangled Transformers to show how Transformers, in principle, can utilize in-context examples to solve the task. The paper then empirically investigates and show that Transformers can learn variants of the task through gradient descent.

**Strengths:**

1. A reasonable extension of a prior publication (Nichani et al. (2024)).
1. Provides deeper insight into ICL capabilities of Transformers.
1. The task could be another sanity check for alternative architectures to test whether they can do what Transformers can or not.

**Weaknesses:**

1. The scope of impact may be a bit limited. While the experiments and construction do deepen our insights to ICL, they don't seem to provide a fundamental shift in our understanding, and may not translate into being particularly informative for specific practical decisions. But this is a more subjective point and I am not factoring this too strongly for the evaluation.

**Questions:**

n/a

---

> ### Author Response · Authors · 2024-11-23
>
> Thank you for your positive evaluation of our work. Regarding the noted weakness, we agree that the scope of the synthetic task is limited compared to real-world complexities. However, as outlined in our general response, our primary goal is to use this controlled framework to isolate and examine the mechanisms transformers employ for causal structure selection. We consider this a valuable step towards addressing more complex tasks in future research.

---

### Official Review · Reviewer_7pgv · 2024-11-03

**Soundness:** 3
**Presentation:** 4
**Contribution:** 3
**Rating:** 6
**Confidence:** 4

**Summary:**

This work introduces a new circuit in Transformers, “selective induction heads” that enables transformers to determine the lag in causally-structured data as per the context (“in-context”) and apply the induction head copying mechanism. A 3-layer disentangled Transformer model is constructed to achieve this, with specifically designed purposes for each layer towards identifying the correct lag: capturing the transition probabilities, aggregating the probabilities, and summing the probabilities to select the lag. Empirical validation on synthetic data shows that the constructed transformer can achieve performance close to the theoretical maximum likelihood estimator across settings (contiguous, non-contiguous, many lags, single head), and the attention maps verify that the layers are performing the functions they were designed for.

**Strengths:**

* The general setting is defined in a clear manner, and intuitively demonstrated through Figure 1.
* The 3-layer disentangled transformer perspective is an interesting construction that is well motivated by designing specific purposes for each layer; the preliminaries are also explained quite well.
* The paper is written clearly despite it being admittedly challenging to follow some of the notation — the visualizations are immensely helpful to follow the construction in Section 4, as well as the various special cases examined in Appendices E, G, and H. These appendix sections were greatly appreciated, as they thoroughly flesh out the nuances in construction for each such setting.
* The attention maps serve to reinforce the model construction; each layer serves their respective intended purposes.

**Weaknesses:**

* More discussion in the empirical validation would be useful, to explain the findings and walk through the plots in Figure 4. For instance, it would be beneficial to explain the difference between the 1 head (1H) and 3 heads (3H) cases, as the proposed construction scales the heads with the number of lags, and detail the implications of these differences.
* The scalability of the method remains unexplored; that is, what are the empirically-realized sets of lags that appear in real-world tasks, and how do they vary across tasks? Many natural language reasoning tasks have a non-linear and more complex causal structure, and it would be good to see how / whether this method extends to such tasks, beyond synthetic ones.
* While Proposition 1 is defended by construction (and the empirical findings), it is unclear how Claim 1 is supported empirically (although it is acknowledged that this claim has yet to be formally proven) through the results in Section 5.

**Questions:**

* Continuing from the note on scalability in the “weaknesses” section: is it necessarily true that there is such an idea as a fixed “correct lag” that ever appears in real data? This is partially addressed in the many lags and non-contiguous lags setting, but appears to require further generalization for such a transformer model to be truly useful (for example, in natural language tasks).
* Are there any core (architectural) considerations that might need to be made for scaling to larger data, from an efficiency standpoint? How many heads would be needed, and can this be reduced? This is addressed to some extent in the many lags setting in an Appendices G and H.

---

> ### Author Response · Authors · 2024-11-23
>
> We appreciate the Reviewer’s comments and questions. We have considered each point and outlined our responses below:
>
> **Weaknesses:**
> >More discussion in the empirical validation would be useful, to explain the findings and walk through the plots in Figure 4. For instance, it would be beneficial to explain the difference between the 1 head (1H) and 3 heads (3H) cases, as the proposed construction scales the heads with the number of lags, and detail the implications of these differences.
>
> We refer the reviewer to Section 4.4 where the paragraph Single-head Transformers discusses the implications of using fewer heads than the number of lags. Furthermore, we conducted new experiments on this topic, reported in **Appendix C**. Further details are provided in the general response **Additional experiments for scalability to more heads and layers**.
>
> >The scalability of the method remains unexplored; that is, what are the empirically-realized sets of lags that appear in real-world tasks, and how do they vary across tasks? Many natural language reasoning tasks have a non-linear and more complex causal structure, and it would be good to see how / whether this method extends to such tasks, beyond synthetic ones.
>
> We acknowledge that the causal structures in our synthetic setting are simplified compared to the complexities of real-world tasks, where the number and nature of lags can vary significantly depending on the data. However, evaluating the applicability of lag selection to real-world tasks is beyond the scope of this work. Our primary goal is to provide a clear and interpretable analysis of the mechanisms transformers use to dynamically select causal structures in controlled environments.
>
> >While Proposition 1 is defended by construction (and the empirical findings), it is unclear how Claim 1 is supported empirically (although it is acknowledged that this claim has yet to be formally proven) through the results in Section 5.
>
> We performed additional experiments validating our Claim 1 and added them in **Appendix E**. More details are also included in the general response under **Additional experiments validating Claim 1**.
>
>
>
> **Questions:**
> >Continuing from the note on scalability in the “weaknesses” section: is it necessarily true that there is such an idea as a fixed “correct lag” that ever appears in real data? This is partially addressed in the many lags and non-contiguous lags setting, but appears to require further generalization for such a transformer model to be truly useful (for example, in natural language tasks).
>
> We do not believe there is a single "correct lag" in real data. Instead, each sentence has its own causal structure. Predicting the next token requires inferring the correct causal structure from the context. Our interleaved Markov chain framework allows us to model this phenomenon, where each source generates sentences according to a lag, and the transformer needs to learn from the context which lag to use to predict the next token in the sequence.
>
> >Are there any core (architectural) considerations that might need to be made for scaling to larger data, from an efficiency standpoint? How many heads would be needed, and can this be reduced? This is addressed to some extent in the many lags setting in Appendices G and H.
>
> As illustrated by our construction, the second layer stores single transitions for each lag within the embedding of the tokens. In the worst case, this would require the embedding dimension to scale with the sequence length. Nevertheless, in practice, we observe that transformers can find more efficient ways to store this information. For example, when training standard transformers, we fixed the embedding dimension to $64$ while having sequences of length $128$. Regarding the number of heads, we conducted additional experiments with varying the number of heads and reported them in **Appendix C, Figure 5 (left)**. The results show that for lags $ 1,2,3,4,5 $, transformers with less than 5 heads can't achieve the optimal performance.

---

> > ### Comment · Reviewer_7pgv · 2024-11-27
> > **Official Comment by Reviewer 7pgv**
> >
> > Thank you for the detailed responses and revisions. I appreciate the new experiments in Appendix C and E, which address my concerns on those respective points. I do still believe that the applicability to real-world tasks is an important aspect for understanding both the scalability of this approach and practical impact of these findings. As such, I would like to keep my score; nonetheless, I find this paper to be interesting and think that it presents meaningful findings for inclusion at ICLR.

---

### Official Review · Reviewer_o5dv · 2024-11-03

**Soundness:** 3
**Presentation:** 3
**Contribution:** 2
**Rating:** 6
**Confidence:** 3

**Summary:**

This paper presents an in context learning task that requires a model to infer what type of markov chain was used to generate the input tokens in order to predict the next token. Then it constructs a solution to this task with a three layer transformer that involves a particular type of attention head they call selective induction heads. In the final two pages an experiment is reported on where a three layer model is trained on this task and then the attention head patterns are visualized. The attention pattern in the constructed model and the trained model is highly similar, suggesting that the very same solution is achieved in both cases.

The paper is well written, and great care is taken in the presentation of the technical details to the theoretical solution. However, the task here is pretty idiosyncratic and a specialized transformer architecture is used to make the analysis easier to perform. The connection to actual in-context learning scenarios seems like it might be a bit fraught, I’m not sure how much closer a lagging markov chain is to natural language than a normal markov chain. I also get the feeling that moving to a standard transformer architecture would make things a lot less pretty...

Generally, I am amenable to technical papers that deal with toy settings, so I think the core content of this paper is solid. However, I don’t think that the title and framing of the paper is accurate. The process of selecting causal structures in context is incredibly general and complex, and selective induction heads are just a step in the direction of understanding how in context learning occurs. I think the title and framing needs more transparently communicate that this work is largely theoretical. How you do this is totally up to you and I’m willing to raise my score if you address this concern.

The attention pattern visualization is remarkably similar between the constructed model and the trained model, so while this analysis is correlational, I would expect the causal claims to follow as well. That being said, you might wonder whether a standard transformer model trained on this task implements basically the same solution (I assume that the attention head pattern is different, otherwise you would have included results). I think the right way to understand this involves causality, and all the work that you reference comes from an AI safety mechanistic interpretability background, but there has been loads of work done in causal analysis of neural networks. Here are some review/history papers to check out and get some citations from:

https://arxiv.org/abs/2408.01416

https://arxiv.org/abs/2301.04709

https://arxiv.org/abs/2410.09087

**Strengths:**

See review

**Weaknesses:**

See review

**Questions:**

See review

---

> ### Author Response · Authors · 2024-11-23
>
> We thank the Reviewer for their detailed comments and questions, which helped improve the quality of our manuscript. We addressed each point and provided our responses in the following:
>
>
> >The paper is well written, and great care is taken in the presentation of the technical details to the theoretical solution. However, the task here is pretty idiosyncratic and a specialized transformer architecture is used to make the analysis easier to perform. The connection to actual in-context learning scenarios seems like it might be a bit fraught, I’m not sure how much closer a lagging markov chain is to natural language than a normal markov chain.
>
> We would like to clarify that we do not claim interleaved Markov chains are inherently closer to natural language. Instead, they are a specific type of order-$\hat{k}$ Markov chains that provide a useful framework for analyzing how transformers learn to select causal structures.
>
> These Markov chains have the unique feature of being learnable through two distinct methods, both implementable by transformers:
> 1. **In-context learning of the $\hat{k}$-gram (studied in [1])**: This approach involves learning the Markov chain of order $\hat{k}$, corresponding to the largest lag in the sequence. However, it is sample-inefficient, with sample complexity growing exponentially with $\hat{k}$
> 2. **In-context selection of the correct lag**: In this more efficient method, the transformer selects the correct lag from the context to predict the next token, achieving optimal performance.
>
> Our theoretical construction shows that a 3-layer attention-only transformer can achieve optimal sample complexity using the second method. Empirically, we observe that trained transformers (both standard and disentangled) naturally learn this strategy, which confirms that they implement mechanisms for lag selection rather than relying on inefficient high-order Markov chain learning.
>
> By having transformers focus on in-context selection rather than in-context learning, our task offers an ideal framework for studying how transformers infer causal structures.
>
> > I also get the feeling that moving to a standard transformer architecture would make things a lot less pretty...
>
> To show that our findings are not a byproduct of the specialized architecture of the disentangled transformers, we added some additional experiments to validate the compatibility of our construction and solution with standard transformers. See general response **Disentangled vs standard transformers, additional experiments:** for more details.
>
> >Generally, I am amenable to technical papers that deal with toy settings, so I think the core content of this paper is solid. However, I don’t think that the title and framing of the paper is accurate. The process of selecting causal structures in context is incredibly general and complex, and selective induction heads are just a step in the direction of understanding how in context learning occurs. I think the title and framing needs more transparently communicate that this work is largely theoretical. How you do this is totally up to you and I’m willing to raise my score if you address this concern.
>
> We agree with the reviewer that selective induction heads are just the first step towards understanding the process of selecting causal structures in context. In the revision, we clarified in both the abstract and introduction that the focus is on a synthetic task, that we analyze 3-layer attention-only transformer, and the theoretical nature of the work (see modifications in red in the revised manuscript).
>
> We are open to modifying the title if other reviewers agree. We propose to add Markov chain as "Selective induction heads: transformers and in-context causal structure selection in Markov Chains" to highlight the theoretical nature of the work.
>
>
> [1] Nichani, Eshaan, Alex Damian, and Jason D. Lee. "How transformers learn causal structure with gradient descent." arXiv preprint arXiv:2402.14735 (2024).

---

> > ### Author Response · Authors · 2024-11-23
> >
> > >The attention pattern visualization is remarkably similar between the constructed model and the trained model, so while this analysis is correlational, I would expect the causal claims to follow as well. That being said, you might wonder whether a standard transformer model trained on this task implements basically the same solution (I assume that the attention head pattern is different, otherwise you would have included results).
> >
> > To validate that standard transformers implement the same solutions as our construction and that our findings extend to it, we repeated all experiments in Section 5 using standard transformers and included additional experiments in **Figure 3,4** and **Appendices C and D**.
> >
> > In these experiments, the attention was parameterized as usual using the three matrices Q, K and V and both the semantic and positional embeddings are learned during training. The results report a remarkable similarity to those obtained with disentangled transformers, both in terms of performance (see the yellow line in Figure 3) and attention maps (Figure 4 and Appendix C and D).
> >
> > Specifically:
> > - The first layer in standard transformers learns to extract the likelihood of individual transitions, as the disentangled transformer and our theoretical construction (in Figure 4 we can observe how the values with lighter colours are in the same positions for all 3 models)
> > - The second layer shares similar patterns for aggregating transitions.
> > - The third layer effectively implements the selective induction head, copying the token corresponding to the correct lag.
> >
> > These results strengthen our contributions by illustrating that standard transformers naturally discover the same solutions as disentangled transformers. Furthermore, they validate that disentangled transformers provide a good proxy for studying and understanding the behaviour of standard transformers.
> >
> > >I think the right way to understand this involves causality, and all the work that you reference comes from an AI safety mechanistic interpretability background, but there has been loads of work done in causal analysis of neural networks. Here are some review/history papers to check out and get some citations from:
> >
> > We thank the reviewers for these references which we discussed and integrated in our manuscript.

---

> > > ### Comment · Reviewer_o5dv · 2024-11-26
> > >
> > > I appreciate the responsiveness to my concerns, and I'm going to raise my score. I think the content of this paper is pretty interesting and worth having at ICLR.

---

### Official Review · Reviewer_acym · 2024-11-03

**Soundness:** 3
**Presentation:** 4
**Contribution:** 2
**Rating:** 6
**Confidence:** 3

**Summary:**

This paper proposes a new synthetic task based on interleaved Markov Chains in order to better mimic the complex token interactions in natural language. The authors come up with an algorithm to solve this task with a 3 layer attention-only (disentangled) transfomer. Notably, the authors also show that when they train a similar model with gradient descent, it learns similar mechanisms (apparent by comparing the attention maps).

**Strengths:**

* The paper is well written, I learned something.
* Introduces a new synthetic task and discusses how transformers can successfully solve it with an interpretable mechanism. The authors also show that the same mechanism is learned when training with gradient descent, which I found most interesting.

**Weaknesses:**

* I have no major concerns apart from the utility of this insight in the broader language (or sequence) modeling tasks that we actually care about. See questions below.

**Questions:**

* The synthetic task of modeling interleaved Markov Chains is interesting and definitely a step in the right direction. But do you think it is yet a good enough proxy to model the complex dynamics of token interactions? In language modeling, often these interactions are informed by the semantics of the tokens and the context. Whereas I don't think here the token meanings are important in any way.

* In my opinion, one of the things that made the induction head work interesting is that they found heads behaving in similar ways in actual large language models which were trained on real textual data. I am not really sure if the same can be said about the selective induction head circuit you found here. If this is something that can only be observed in a synthetic task, do we really care about it? Or, do you have any hypotheses about how selective induction heads might manifest in models trained on natural language texts? And, how do you plan to investigate this hypothesis (maybe in future work)?

* In Section 5 you mentioned that a model when trained with less than k attention heads per layer was also able to successfully solve this task. This either suggests a completely different algorithm or is indicative of head-level superposition of information. Can you get more insights into this by looking at the attention maps?
    * Also, have you tried with more than K heads per layer? I wonder if it would further break the mechanism. Or, you might start seeing duplicate heads behaving in similar ways.
    * A similar question about the number of layers --- You need at least 2 layers to form the usual induction head behavior. Do you really need at least 3 layers for "selective" induction heads? Also, what happens with more layers? Do more layers introduce more "steps"? Or, some of the layers essentially become identity functions?

---

> ### Author Response · Authors · 2024-11-23
>
> We thank the reviewer for the useful comments, which will help improve the quality of our manuscript. Below, we provide responses to their questions:
> >The synthetic task of modelling interleaved Markov Chains is interesting and definitely a step in the right direction. But do you think it is yet a good enough proxy to model the complex dynamics of token interactions? In language modelling, often these interactions are informed by the semantics of the tokens and the context. Whereas I don't think here the token meanings are important in any way.
>
> Our synthetic task is indeed a simplification of natural language, but it captures a critical aspect: context-dependent causal relationships.
>
> Modelling such relationships, where causality depends on both context and token semantics, is inherently complex. Our interleaved Markov chain setup is a simplification where the causal structure is fixed within a sequence but is allowed to change across sequences. Token semantics remain important as they directly determine transition probabilities and, thus, the interactions between tokens. Transformers rely on the semantics of the tokens in a sequence to learn both the transition matrix P and to identify the correct causal relationship between tokens (by computing normalized likehoods based on the learned P).
>
> To further approximate real-world language, where causal relationships can vary within a single context, a natural extension for future works would involve dynamically sampling the lag for each context, such that $p(k_t|x_{t-1},\dots,x_{t-c})$ for some fixed context length $c$.
>
> >In my opinion, one of the things that made the induction head work interesting is that they found heads behaving in similar ways in actual large language models which were trained on real textual data. I am not really sure if the same can be said about the selective induction head circuit you found here. If this is something that can only be observed in a synthetic task, do we really care about it? Or, do you have any hypotheses about how selective induction heads might manifest in models trained on natural language texts? And, how do you plan to investigate this hypothesis (maybe in future work)?
>
> The reviewer raised an important point about the relevance of selective induction heads for real-world language tasks. The induction head mechanism, while insightful, is relatively simple: it detects repeated patterns and can copy tokens by focusing on previous occurrences of the same tokens. This mechanism is well-suited for specific types of positional dependencies but does not extend well to more nuanced, context-dependent relationships.
>
> In contrast, our selective induction head mechanism is more complex. It involves not only identifying a relevant position in the past by simply comparing the semantics of single tokens but extracting and accumulating some statistics (likelihood of single transitions in our case) from the entire context and using it to select the position from which to copy. This allows the model to make decisions about which prior tokens carry the most relevant information based on the entire history, rather than relying solely on pattern repetition.
>
> We believe that a similar mechanism may occur in real-world language models, where different layers specialize in distinct roles: some layers focus on extracting and encoding relevant information therefore having attention concentrating on fewer tokens, while others accumulate and integrate this information to determine which prior tokens are most significant therefore having more uniform attention scores. Similar findings have been shown in [1] in the context of attention sinks.
>
> However, given the higher complexity than in standard induction heads, we anticipate that detecting selective induction heads in LLMs might require designing ad-hoc NLP tasks: if the causal structures in such tasks are sufficiently clear, we will expect to see similar attention maps to those observed in our synthetic task. We leave these investigations to future work.

---

> > ### Author Response · Authors · 2024-11-23
> >
> > >In Section 5 you mentioned that a model when trained with less than k attention heads per layer was also able to successfully solve this task. This either suggests a completely different algorithm or is indicative of head-level superposition of information. Can you get more insights into this by looking at the attention maps?
> >
> > With fewer than $K$ attention heads with $K$ the number of lags, the model can still solve the task but with worse sample complexity. In **appendix C**, we added experiments with different numbers of heads on the task $\mathcal{K}= \{1,2,3,4,5\}$. **Figure 5 (right)** reports the performance across these configurations:  transformers with less than $5$ heads perform worse compared to those with $5$ heads, confirming the intuition from our theoretical construction. Moreover, Figures 6, 7, and 8 illustrate the attention maps for a 3-layer transformer with only 1, 2, and 3 heads respectively in the second layer, despite the task having 5 lags. Remarkably, even with fewer than  5  heads, the layers remain consistent with our construction: the first layer extracts transition probabilities, the second aggregates them, and the third implements the selective head. However, with fewer heads, the second layer appears to find an efficient way to superpose information—a mechanism we could not theoretically describe. Understanding this behaviour in the second layer remains an open question for future work.
> >
> >
> > >Also, have you tried with more than K heads per layer? I wonder if it would further break the mechanism. Or, you might start seeing duplicate heads behaving in similar ways.
> >
> > **Figure 5 (right)** also reports the performance with more heads than lags, i.e. $K>5$. The performance remains consistent with that observed for $K=5$, where the number of heads matches the number of lags in the tasks. This suggests that the additional heads do not contribute to the algorithm but instead introduce redundancy. As hypothesized by the reviewer, we do observe duplicate attention heads behaving in similar ways, effectively replicating the same transition probabilities and storing redundant information in the embeddings.
> >
> > >A similar question about the number of layers --- You need at least 2 layers to form the usual induction head behaviour. Do you really need at least 3 layers for "selective" induction heads? Also, what happens with more layers? Do more layers introduce more "steps"? Or, some of the layers essentially become identity functions?
> >
> > We added additional experiments in **Figure 5 (left)** to investigate the impact of using more or less than 3 layers. The results show that with only 2 layers, no combination of heads is able to solve the task, confirming that at least 3 layers are necessary for implementing selective induction heads. Adding more than 3 layers does not affect the performance, as the model achieves the same performance as with 3 layers. However, with additional layers, we observe the emergence of structures in the attention maps that deviate from our theoretical construction. These structures remain uninterpreted, suggesting that the extra layers might encode redundant or alternative representations whose role requires further investigation.
> >
> > [1] Yu, Zhongzhi, et al. "Unveiling and harnessing hidden attention sinks: Enhancing large language models without training through attention calibration." arXiv preprint arXiv:2406.15765 (2024).

---

> > > ### Comment · Reviewer_acym · 2024-11-24
> > >
> > > I am satisfied with the authors' response and would like to keep my score. I still remain skeptical if selective induction heads and other insights from this paper can cleanly transfer to actual language models (or other sequence models) trained on real data. But I do recognize the value of the insights and the potential of the synthetic task to better understand the mechanisms of transformers.
> > >
> > > I would like to thank the authors for their response and for the interesting work. I wish them good luck with the future work.

---

### Author Response · Authors · 2024-11-23
**General response**

We thank the Reviewers for their valuable feedback, which helped improve our work. We made several revisions and additions, highlighted in red in the updated manuscript and summarized below:


1. **Interleaved Markov chains and natural language:** We would like to clarify that we do not claim interleaved Markov Chains to be a comprehensive or accurate model of natural languages. Rather, our goal is to design a synthetic task that captures some key properties relevant to next-token prediction, enabling the study of transformers and their capabilities. In natural language, different sentences have different causal structures, requiring a model to infer which words in the context are most relevant for accurately predicting the next ones. Our interleaved Markov Chain framework mirrors this task by generating sequences from multiple sources, each with its own causal structure (lag), requiring the model to identify and leverage the appropriate one from the context to predict the next token. The analogy with natural language stems from the fact that one could consider the different sources each generating sentences with a given structure and at inference time the model needs to infer which one generated the current sentence to continue it.
This controlled synthetic setting allows us to isolate and examine in detail the mechanisms through which transformers dynamically select causal structures and the specific operations learned by attention layers to achieve this ability. Our aim is not to replicate the full complexity of natural language but to advance the understanding of how transformers predict tokens within a sequence using the context.

&nbsp;

2. **Disentangled vs standard transformers, additional experiments:** Following Reviewer **o5dv**’s suggestion, we have strengthened and expanded our discussion on how the results for the disentangled transformer generalize to the standard transformer. We repeated all our experiments with standard transformers with the usual parameterization of the attention layer given by Q,K,V (compared to the single matrix in the disentangled transformer) and learned positional and semantic embeddings (compared to one-hot encodings). Moreover, the output is not concatenated but summed as per standard practice. The results are reported in Figure 3 and Figure 4 (yellow line) in the revised version of the paper. We show that **the standard transformer matches the performance of the construction** and, remarkably, the attention maps (Figure 5) also match our construction. This new experiment provides compelling evidence that our construction is not merely a byproduct of the disentangled transformer’s architecture but it is also implemented and learned by standard transformers. To provide further evidence we also added the attention maps for $\mathcal{K} = \{1,2\}$, $\mathcal{K} = \{1,2,3\}$ and $\mathcal{K} = \{1,3,4\}$ for constructed, disentangled and standard transformer in Appendix D. We believe that this new evidence substantially strengthens our contribution. We would like to remark that the disentangled transformers merely facilitate the analysis by keeping the residual stream explicitly disentangled, but as shown in this new set of experiments, the observed construction is also implemented and learned by standard transformers.

&nbsp;

3. **Additional experiments validating Claim 1 (Appendix E):** As suggested by Reviewer **7pgv**, to complement Figure 4c, we provide direct empirical evidence to support Claim 1 in Appendix E. We sample a set of $12$ lags uniformly between $1$ and $30$; we then sample $1000$ different transition matrices and for each matrix and each lag $1000$ sequences of length $1000$ according to the respective Interleaved Markov chain. For each lag and each set of sequences we then compute the expectation in Claim1 by Monte Carlo. The results are reported in the histogram in Fig 16; all values are positive therefore confirming our claim. Moreover, the results in Figures 17 and 18 show the quantities in the claim for each lag. Furthermore, in Figures 19 and 20 we compute the average normalized transition probabilities along the sequence and show that it quickly becomes larger for true lag.

---

> ### Author Response · Authors · 2024-11-23
>
> 4. **Additional experiments for scalability to more heads and layers (Appendix C):** As suggested by Reviewers **acym** and **7pgv** we included a discussion of the cases with more layers or more heads in the second layer and provided additional experiments in Appendix C. We train standard transformers with learned positional and semantic embeddings in the same setup as for the experiments in the main text.
>     - **Varying number of heads:** Figure 5 (right) includes scenarios with fewer, equal to, or more than  $K$  heads. As predicted by our construction increasing the number of heads leads to performances that get closer to the maximum likelihood up to having the number of heads equal to the number of lags in the set $K$. Beyond this point adding more heads does not change the performance, as expected since ML is optimal.
>     - Figures 6, 7, and 8 illustrate the attention maps for a 3-layer transformer with only 1, 2, and 3 heads respectively in the second layer, despite the task having 5 lags. Remarkably, even with fewer than  $5$  heads, the attention maps remain consistent with our theoretical construction, displaying analogous patterns.
>     - **Varying number of layers:** In Figure 5 (left) for lags in $\mathcal{K} = \{1,2,3\}$, we show the behaviour of the model with 2 layers and different combinations of heads $[1,1],[3,1],[1,3],[3,3]$. The results clearly show that transformers with $2$ layers can't solve the task. Moreover, increasing the number of layers beyond $3$ does not change the performance.

---

### Meta-Review · Area_Chair_p2ai · 2024-12-22

**Metareview:**

Summary: The paper attempts to improve our understanding of transformer architectures and attention mechanism in particular. In this regards, the authors introduced a synthetic task based on interleaved Markov chains with varying lags. A 3-layer transformer was shown to solve this task by a constructive proof and empirically verified alignment between this construction and transformers trained with Adam.

Strengths:
- Designed an interesting synthetic task of interleaved Markov chains to simulate dynamic causal structures for understanding induction heads
- Well-motivated theoretical analysis with clear mathematical formulation and proofs
- Empirical validation showing alignment between theoretical construction and learned solutions in both disentangled and standard transformers implementing similar mechanisms
- Clear presentation with helpful visualizations of attention patterns

Weakness:
- Some reviewers had a lukewarm response to the paper
- Limited scope:
    a. focuses on synthetic Markov chain setting rather than real-world tasks.
    b. synthetic task lacks the complexity of real-world token interactions, which involve semantic and contextual nuances.
    c. while the 3-layer transformer is effective for the proposed task, not clear if results extend to deeper or more complex models used in practical NLP.
- Could benefit from more discussion of generalization beyond training distribution

Decision:
Most of the reviewers were leaning positive and agreed that while the practical implications are limited, the paper represents an important step forward in understanding transformer working, providing both theoretical insights and paths for future work. It merits publication at ICLR 2025 after a few minor edits: 1. acknowledging the practical applicability, and 2. either remove the claim around optimal sample complexity or show it precisely (e.g. optimal against what, proof of optimality, etc.).

**Additional Comments On Reviewer Discussion:**

We thank the authors and reviewers for engaging during the discussion phase towards improving the paper. Below are some of the highlights:

1. Applicability to real-world tasks/LLMs (Reviewers Lmxm, acym)
- Authors clarified focus is on understanding transformer capabilities, not direct LLM applications
- Reasonable response though practical relevance remains limited

2. Experimental validation of claims (Reviewer 7pgv)
- Authors added new experiments in Appendix E validating Claim 1
- Satisfactorily addressed through additional results

3. Architectural variations (Reviewers acym, 7pgv)
- Authors added comprehensive experiments on varying heads/layers
- Well-addressed through new results in Appendix C

4. Standard vs disentangled transformers (Reviewer o5dv)
- Authors added experiments showing similar behaviour
- Strong experimental validation provided

The authors have done an excellent job addressing reviewer concerns through additional experiments and analysis during the rebuttal period.

---

### Decision · Program_Chairs · 2025-01-22

Accept (Poster)